# Breaking Scale Anchoring: Frequency Representation Learning for Accurate High-Resolution Inference from Low-Resolution Training

**Wenshuo Wang**[1]**, Fan Zhang**[2,3*]

[1] School of Future Technology, South China University of Technology, China
[2] State Key Laboratory of Ocean Sensing & Ocean College, Zhejiang University, China
[3] Kavli Institute for Astrophysics and Space Research, Massachusetts Institute of Technology, USA
`202364870251@mail.scut.edu.cn`, `f.zhang@zju.edu.cn`

## Abstract

Zero-Shot Super-Resolution Spatiotemporal Forecasting requires a deep learning model to be trained on low-resolution data and deployed for inference on high-resolution. Existing studies consider **maintaining** similar error across different resolutions as indicative of successful multi-resolution generalization. However, deep learning models serving as alternatives to numerical solvers should **reduce** error as resolution increases. The fundamental limitation is, the upper bound of physical law frequencies that low-resolution data can represent is constrained by its Nyquist frequency, making it difficult for models to process signals containing unseen frequency components during high-resolution inference. *This results in errors being anchored at low resolution, incorrectly interpreted as successful generalization.* We define this fundamental phenomenon as a new problem distinct from existing issues: **Scale Anchoring**. Therefore, we propose architecture-agnostic Frequency Representation Learning. It alleviates Scale Anchoring through resolution-aligned frequency representations and spectral consistency training: on grids with higher Nyquist frequencies, the frequency response in high-frequency bands of FRL-enhanced variants is more stable. This allows errors to decrease with resolution and significantly outperform baselines within our task and resolution range, while incurring only modest computational overhead.

## 1 Introduction

Traditional numerical simulation methods in Spatiotemporal Forecasting (STF) can achieve high accuracy yet incur substantial computational costs (Choi & Moin, 2011). Recent research utilizing deep learning methods for STF can effectively balance accuracy and computational efficiency (Zhang et al., 2023; Saad et al., 2024). However, high-resolution, high-fidelity Direct Numerical Simulations (DNS) that provide training data for deep learning methods face extremely high costs. Even with high-resolution data available, the extremely high resolution poses impossible training VRAM requirements for current hardware, while the requirements for inference are much smaller. Zero-shot super-resolution (ZS-SR) deep learning models can leverage low-cost, low-resolution data for low-VRAM training to perform high-resolution STF (Li et al., 2020).

Existing ZS-SR STD methods are based on Neural Operators (NOs). This is because, unlike general neural network architectures that learn function mappings from point space to point space, NOs learn functional mappings from function to function (Lu et al., 2021). Since functions can be discretized into spaces of arbitrary resolution, NOs naturally meets the requirements for inputs and outputs of different resolutions (Kovachki et al., 2023). On the other hand, many variants, such as Fourier NO, learn parameters in a scale-agnostic manner in the frequency domain, which can also improve the generalization of the model's physical laws (Li et al., 2020).

---

*Corresponding author.

Existing researchs expect models' successful generalization to **maintain** similar accuracy across inputs of different resolutions (Talebi & Milanfar, 2021; Gao et al., 2025). From this perspective, our pilot study on ZS-SR STF for all mainstream architectures in Section 3 shows that every baseline exhibits excellent generalization. However, for a $p$-th order numerical solver simulating at a resolution $\alpha$ times higher than the low resolution, the error theoretically **decrease** by a factor of $\alpha^p$. This gap arises from the fact that the upper bound of the frequency of physical laws that low-resolution data can represent is limited by the Nyquist frequency of the training data. When models are trained on low-resolution data and perform inference at high resolution: the model struggles to handle unseen high-frequency components, causing the model's error to be anchored at the low-resolution data. We refer to this data-driven limitation as **Scale Anchoring**, which is fundamentally different from previously studied issues such as Spectral Bias (SB) and Discretization Mismatch Error (DME).

Addressing Scale Anchoring requires improving generalization to higher relative frequencies that are not present at the training grid. Prior methods for cross-resolution generalization of physical laws were not explicitly designed to tackle this scale-anchoring mechanism (Li et al., 2020; 2024c). We therefore propose **Frequency Representation Learning (FRL)**: (i) construct multi-resolution training data via downsampling; (ii) introduce Nyquist-normalized frequency representations that yield resolution-invariant embeddings for the same physical frequencies; and (iii) add a frequency-aware loss to promote spectral consistency across scales. Steps (i) and (iii) follow standard practices in multi-scale training and spectral regularization, while Step (ii), which is aligned with Scale Anchoring, is novel. Under mild assumptions, our analysis shows that this alignment encourages more stable high-frequency bands. It allows FRL to reduce the high-resolution error of a given baseline as resolution increases, although it does not guarantee strict order convergence like numerical solvers.

Our contributions are summarized as follows: (a) Identifying Scale Anchoring, a previously unrecognized fundamental limitation in ZS-SR STF that incorrectly interpreted as successful generalization; (b) Providing theoretical analysis and empirical validation of Scale Anchoring and its mechanism; (c) Proposing architecture-agnostic FRL for Scale Decoupling; (d) Extensive experiments across diverse architectures show that FRL-enhanced methods decouple Scale Anchoring. In ZS-SR STF, they demonstrate higher accuracy with modest increases in training time and memory overhead; (e) We identify the failure modes of FRL, characterize the conditions for effectiveness and the boundaries of failure, and suggest potential improvement strategies.

## 2 RELATED WORK

### 2.1 ZERO-SHOT SUPER-RESOLUTION SPATIOTEMPORAL FORECASTING

STF tasks take one or multiple physical spatial field snapshots at different time steps as input, predict the next time step's snapshot, and repeat iteratively. ZS-SR STF tasks train models on low-resolution snapshots only, but at inference time take high-resolution physical spatial field snapshots at different time steps as input, predict the next time step's snapshot, and repeat iteratively. NOs are naturally suited for this task due to their resolution-agnostic input and output capabilities (Li et al., 2020; Kovachki et al., 2023). Existing NO variants span multiple directions: FNO pioneered ZS-SR through frequency-domain learning (Li et al., 2020; Jiang et al., 2023; Atif et al., 2024); PINO enhanced accuracy via physics constraints (Li et al., 2024c); TNO specialized temporal modeling for long-term predictions (Diab & Al-Kobaisi, 2025); and multi-scale methods like Multi-Grid Tensorized FNO and U-FNO reduced computational complexity through hierarchical decomposition (Kossaifi et al., 2023; Wen et al., 2022). While these methods improve resolution generalization, they do not explicitly address the fundamental frequency limitation imposed by the training data's Nyquist frequency. Without mechanisms to learn or extrapolate frequency patterns beyond this hard boundary, they remain fundamentally unable to resolve Scale Anchoring.

### 2.2 SCALE ANCHORING V.S. RELATED PHENOMENA

Scale Anchoring shares superficial similarities with several phenomena but differs fundamentally:

SB describes the tendency of neural networks to fit target functions from low to high frequencies within the training Nyquist band, leading to larger errors on high-frequency components that are actually present in the supervision data (Xu et al., 2019; Rahaman et al., 2019). To mitigate SB, prior work introduces Fourier feature mappings and positional encodings, periodic activation functions,

multi-scale or hierarchical architectures, and explicit spectral or frequency-weighted losses, all of which strengthen the network's ability to represent and fit high-frequency content already contained in the data (Tancik et al., 2020; Sitzmann et al., 2020; Liu et al., 2024; 2025).

In the NO literature, lack of discretization-invariance (lack of DI) and DME formalize the problem that the same operator network can produce inconsistent outputs on different grids (Bartolucci et al., 2023; Gao et al., 2025; Lanthaler et al., 2024). These works analyze aliasing and discretization error to propose alias-free spectral parameterizations, multi-grid training, and carefully designed interpolation operators to enforce cross-grid consistency. More broadly, multi-resolution generation in vision and scientific ML—whether cascaded, patch-based, or training-free—expose models to multiple resolutions (Tian et al., 2023; He et al., 2023; Ho et al., 2022; Bar-Tal et al., 2023; Yu et al., 2023). In this sense, the first and third steps of FRL (multi-resolution training and frequency-aware loss) follow standard practices in these research directions and are not methodologically novel.

By contrast, Scale Anchoring is driven by the information-theoretic limitation that low-resolution training data cannot represent physical frequencies above its Nyquist limit. The difference between Scale Anchoring and other phenomena has three implications. First, in terms of source, SB, lack of DI, and DME all arise from architectural and optimization choices and can be eliminated through appropriate model design and training. Whereas Scale Anchoring arises from the Shannon–Nyquist sampling bound on the data itself. Second, in terms of scope, lack of DI and DME are formulated specifically for NOs, while SB and SA occur broadly across any structures (including NOs) as we empirically demonstrate in Section 4.2. Third, in terms of consequence, SB, lack of DI, and DME are soft tendencies that do not constitute hard limits on achievable accuracy. Whereas Scale Anchoring imposes an information-theoretic lower bound on high-resolution error. Therefore, Scale Anchoring is orthogonal to existing phenomena: even if the model is completely alias-free, discretization-invariant, and fits signals within the training Nyquist band equally well, a model trained only on low-resolution data will never observe frequency components above the training Nyquist frequency.

## 3 EXISTENCE OF SCALE ANCHORING

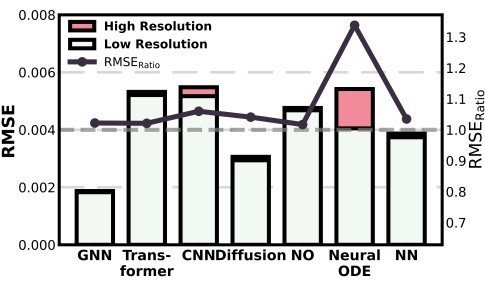

Figure 1: High and low resolution RMSE with their ratio across different methods in 3D ZS-SR fluid simulation.

Table 1: Multi-resolution $\text{RMSE}_{\text{Ratio}}$ across different methods in ZS-SR 3D fluid simulation.

| Method | 2.4× | 8× | 65.5× |
|---|---|---|---|
| GNN (Neural SPH) | 1.000 | 1.000 | 1.022 |
| Transformer (DeepLag) | 1.006 | 1.012 | 1.021 |
| CNN (PARCv2) | 1.012 | 1.035 | 1.060 |
| Diffusion (DYffusion) | 1.010 | 1.024 | 1.041 |
| NO (SFNO) | 1.004 | 1.011 | 1.017 |
| Neural ODE (FNODE) | 1.067 | 1.180 | 1.338 |
| NN (NeuralFluid) | 1.011 | 1.024 | 1.035 |

We first demonstrate the existence of Scale Anchoring in ZS-SR fluid simulation. We use the same baselines and fluid dataset as in Section 6.1. For the most commonly used deep learning architectures in fluid simulation, we tested each State-Of-The-Art (SOTA) method trained on low-resolution ($32^3$) data and evaluated on 2.4×, 8×, 65.5× resolution, measuring RMSE and $\text{RMSE}_{\text{Ratio}} = \text{RMSE}_{\text{High}}/\text{RMSE}_{\text{Low}}$. The results are shown in Figure 1 and Table 1.

As shown in Figure 1, all architectures achieve $\text{RMSE}_{\text{Ratio}}$s between 1-1.4 under 64× super-resolution. As shown in Table 1, the RMSE changes across different resolutions for all architectures remain minimal ($\sim$1). From the MRG perspective, this demonstrates successful generalization. However, for a physical domain with original grid spacing $\Delta\mathbf{x}$: After increasing the resolution by a factor of $\alpha$, for a numerical scheme with $p$-th order accuracy, the truncation error is:

$$E_1 = C \cdot (\Delta\mathbf{x})^p + O\left((\Delta\mathbf{x})^{p+1}\right) \tag{1}$$

$$E_2 = C \cdot (\Delta\mathbf{x}/\alpha)^p + O\left((\Delta\mathbf{x}/\alpha)^{p+1}\right) \tag{2}$$

where $E_1$ represents the original error and $E_2$ represents the new error. The error reduction factor is $\frac{E_1}{E_2} \approx \frac{(\Delta \mathbf{x})^p}{(\Delta \mathbf{x}/\alpha)^p} = \alpha^p$. This indicates that if the models truly function as numerical solvers, errors should decrease following a power law as resolution increases. However, the results shown in Table 1 indicate that the model fits data at a specific resolution rather than learning the correct physical operator to become a true numerical solver. We refer to this phenomenon, where the inference pattern and errors are anchored at the training resolution, as **Scale Anchoring**.

# 4 MECHANISM OF SCALE ANCHORING

## 4.1 THEORETICAL ANALYSIS

Physical evolution in spatiotemporal systems is governed by partial differential equations defining operators on function spaces. The evolution operator $\mathcal{F} : C(\Omega) \to C(\Omega)$ that advances the physical field $u(\cdot, t)$ to $u(\cdot, t + \Delta t)$: it operates on continuous functions and embodies resolution-independent physical laws. Neural networks, however, learn fundamentally different mappings. When trained at resolution $\rho_0$ with grid spacing $\Delta x = 1/\rho_0$ and $N_{\rho_0}$ grid points, a neural network minimizes:

$$\min_{\Theta} \mathbb{E}_{u \sim \mathcal{D}} \| G_{\Theta}(S_{\rho_0}(u)) - S_{\rho_0}(\mathcal{F}[u]) \|^2 \tag{3}$$

where $S_{\rho_0}$ samples the continuous field onto a discrete grid and $G_{\Theta}$ applies learned operations. $G_{\Theta}$ learns a function mapping between finite-dimensional spaces $\mathbb{R}^{N_{\rho_0}} \to \mathbb{R}^{N_{\rho_0}}$, not the functional $\mathcal{F}$. When deployed at resolution $\rho$, the same learned parameters $\Theta_{\rho_0}$ are applied, denoted as $G_{\Theta_{\rho_0}}^{(\rho)}$.

This function-versus-functional distinction leads to a fundamental frequency limitation:

**Theorem 1** (Frequency Blindness). *A neural network trained at resolution $\rho_0$ cannot correctly process frequency components above the Nyquist frequency $\rho_0/2$. The learned operator's frequency response satisfies:*

$$\hat{G}_{\Theta_{\rho_0}}(\omega) \approx \begin{cases} \hat{\mathcal{F}}(\omega) + \epsilon(\omega), & \omega \leq \rho_0/2 \\ \text{undefined/incorrect}, & \omega > \rho_0/2 \end{cases} \tag{4}$$

This directly causes Scale Anchoring when the network is deployed at higher resolutions:

**Theorem 2** (High-Frequency Error Dominance). *When a network trained at resolution $\rho_0$ is deployed at resolution $\rho > \rho_0$, the Scale Anchoring error bound:*

$$C = \lim_{\rho \to \infty} \left\| G_{\Theta_{\rho_0}}^{(\rho)} \circ S_{\rho} - S_{\rho} \circ \mathcal{F} \right\|_{op} \tag{5}$$

*is dominated by the network's inability to process frequency components in the range $[\rho_0/2, \rho/2]$.*

The complete mathematical derivations are presented in Appendix A.

## 4.2 EXPERIMENT VALIDATION

In this section, we design two experiments to validate Theorems 1 and 2:

**Validation Experiment A.1:** We first use pseudo-spectral methods in the frequency domain to simulate 2D convection-diffusion equations at $64^2$ resolution. We then implement eight commonly used model architectures in STF (GNN, Transformer, CNN, Diffusion, NO, Neural ODE, Mamba, NN) and fully train them on the simulation data. Finally, we input sinusoidal signals ($u(x, y, t) = A \cdot \sin(2\pi f \cdot x)$) of varying frequencies (0-50Hz) and measure the magnitude of the empirical frequency response $H_{\text{mag}}(f) = A_{\text{out}}(f)/A_{\text{in}}(f)$ (output–to–input amplitude ratio), Bandwidth (BW), and Anchoring Ratio: $H(f = 30)/H(f = 34)$ (cutoff sharpness). Results are shown in Figure 2.

All architectures exhibit a unified frequency response pattern: maintaining high $H(f)$ before the Nyquist frequency (32Hz), followed by a cliff-like drop near it. This results in BW concentrated around the Nyquist frequency and high Anchoring Ratios. The universal Scale Anchoring across all architectures validates Theorem 1, confirming that neural networks trained at resolution $\rho_0$ cannot correctly process frequency components above the Nyquist frequency $\rho_0/2$. More detailed experimental settings and results for different training resolutions are provided in Appendix B.

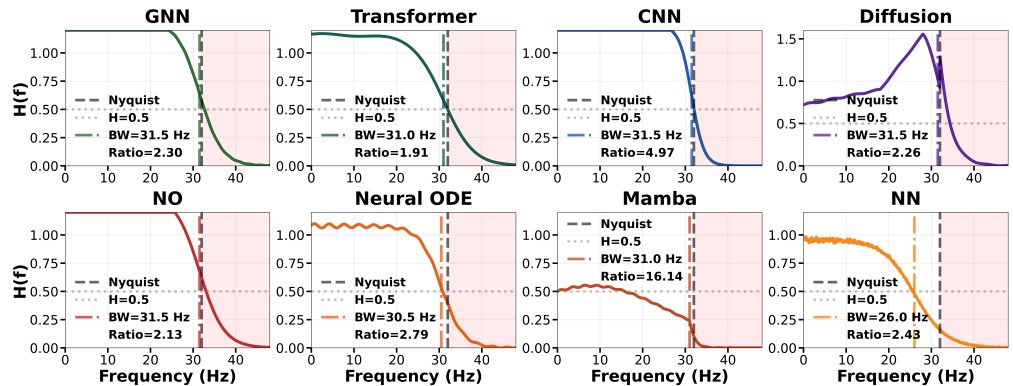

Figure 2: Frequency response analysis on different architectures, models trained at resolution $64^2$.

**Validation Experiment A.2:** We use the same pseudo-spectral method to simulate 2D convection-diffusion equations at resolutions of $32^2$, $64^2$, $128^2$, and $256^2$. We employ the same eight model architectures, fully trained on $64^2$ resolution data. However, we now perform inference at $128^2$ resolution for 50 timesteps and apply Fast Fourier Transform (FFT) to obtain frequency components (10-50Hz) at each step. Results are shown in Figure 3. We then apply FFT to the simulation data and separate bandlimited ($f < 32$Hz) and wideband ($f < 100$Hz) signals. For results simulated at all resolutions, we calculate the $Error\ Ratio =$ bandlimited$_{error}$/wideband$_{error}$ to quantify the proportion of low-frequency error in the total error. Results are shown in Table 2.

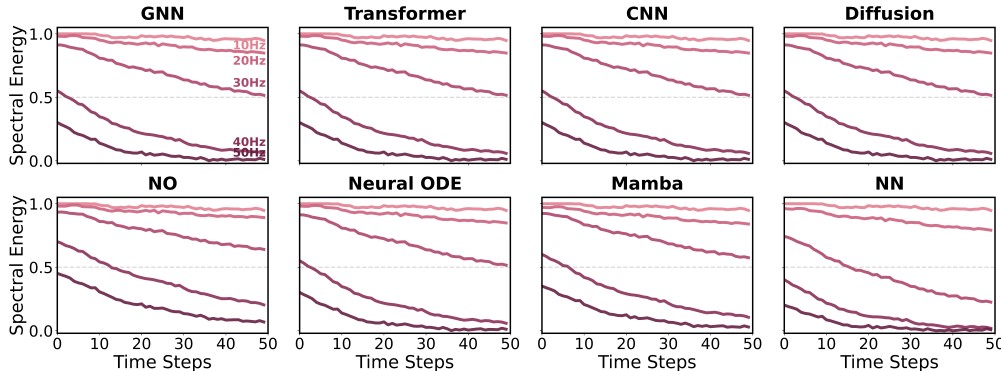

Figure 3: Loss of different frequency components during propagation.

Table 2: $Error\ Ratio$ across test resolutions.

| Model | $32 \times 32$ | $64 \times 64$ | $128 \times 128$ | $256 \times 256$ |
|---|---|---|---|---|
| GNN | 1.000 | 1.000 | 0.357 | 0.223 |
| Transformer | 1.000 | 1.000 | 0.350 | 0.234 |
| CNN | 1.000 | 1.000 | 0.342 | 0.216 |
| Diffusion | 1.000 | 1.000 | 0.351 | 0.244 |
| NO | 1.000 | 1.000 | 0.581 | 0.415 |
| Neural ODE | 1.000 | 1.000 | 0.368 | 0.232 |
| Mamba | 1.000 | 1.000 | 0.431 | 0.287 |
| NN | 1.000 | 1.000 | 0.338 | 0.233 |

The rapid decay of high-frequency components energy in Figure 3 reveals the deeper mechanism behind the phenomenon in Figure 2: as shown by Gao et al. (2025), while high-resolution **single-step** discretization primarily introduces errors in the low-frequency portion, a more dominant mechanism during **multi-step** inference is that models cannot maintain frequencies above the training Nyquist

frequency. This results in the pattern shown in Table 2 where, during high-resolution inference, the proportion of low-frequency error decreases while high-frequency error becomes dominant due to more severe accumulation and amplification. These experimental results validate Theorem 2, confirming that frequency components above the training resolution's Nyquist frequency dominate the error during high-resolution inference.

## 5 SOLUTION FOR SCALE ANCHORING

To address Scale Anchoring, models must generalize to signals containing frequency components beyond the training Nyquist limit. We propose Frequency Representation Learning, an architecture-agnostic approach implemented in three steps:

**Step 1: Multi-Resolution Data Construction.** We construct multi-resolution data through down-sampling:

$$\mathcal{D} = \{(u^{(\rho_j)}(t), u^{(\rho_j)}(t + \Delta t))\}_{j=0}^{J-1}, \quad \rho_j = \rho_0/2^j \tag{6}$$

Each resolution captures different frequency bands up to its Nyquist, enabling the model to understand relative frequency relationships by training on multiple resolutions simultaneously.

**Step 2: Normalized Frequency Representation.** To ensure resolution invariance, we introduce normalized frequency encoding that makes networks aware of relative frequencies:

$$PE_{freq}(x, k, \rho) = \sin\left(2\pi k \cdot x \cdot \frac{1}{k_{Nyq}(\rho)}\right) \tag{7}$$

This encoding satisfies $PE_{freq}(x, k, \rho_1) = PE_{freq}(x \cdot \rho_1/\rho_2, k, \rho_2)$, ensuring identical representation for the same physical frequency across different resolutions. By normalizing frequencies relative to each resolution's Nyquist frequency, the network learns patterns that transfer across scales.

**Step 3: Frequency-Aware Training.** The critical modification to standard training is the addition of a frequency consistency loss. We train the model across all resolutions with a unified objective:

$$\mathcal{L}(\Theta) = \sum_j \left[ \|F_\Theta(u^{(\rho_j)}(t)) - u^{(\rho_j)}(t + \Delta t)\|^2 + \lambda \|\hat{F}_\Theta(u^{(\rho_j)}(t)) - \hat{u}^{(\rho_j)}(t + \Delta t)\|_{freq}^2 \right] \tag{8}$$

The frequency consistency term ensures spectral accuracy across scales. Through multi-scale training with normalized encodings, the network learns resolution-invariant frequency patterns.

Among these, Step 1 and Step 3 follow standard practices in existing multi-scale training and spectral regularization methods and do not constitute novel techniques. By contrast, the normalized frequency representation in Step 2 is designed from the Scale Anchoring perspective. To the best of our knowledge, is novel compared to existing frequency-encoding approaches. It is also the key mechanism for mitigating Scale Anchoring: as shown by the ablation results in Appendix H, Step 2 is a necessary component for breaking Scale Anchoring.

Furthermore, the effectiveness of FRL relies on the underlying physical system satisfying assumptions in the spectral domain: the energy spectrum envelope and local relationships between low/mid-frequency bands and higher-frequency bands remain smooth. For the moderate-Reynolds-number flows and weather forecasting data, this assumption typically holds. In systems with extremely high Reynolds numbers or extreme weathers, however, local spectral relationships are no longer smooth; as a result, FRL's frequency extrapolation capability degrades. Appendix I uses high-Re turbulence as an example to analyze this failure mode. It also discusses potential improvements by incorporating explicit physical spectral constraints such as Kolmogorov scaling laws, into FRL's representation and loss design. The complete FRL pseudocode is provided in Appendix C, and a detailed analysis of training/inference complexity and memory usage is summarized in Appendix E.

## 6 EMPIRICAL EVALUATION

We evaluate our approach on two representative STF domains: fluid simulation and weather forecasting. Models are trained on low-resolution data and evaluated on high-resolution data.

## 6.1 EXPERIMENTAL SETUP

### 6.1.1 DATASETS

**Fluid Simulation**: Non-reacting HIT consists of 3D homogeneous isotropic turbulence simulations with a spherical $O_2$ core (radius $0.25L$) embedded in $CH_4$ (Chung et al., 2022). The dataset tracks velocity components, thermodynamic variables, and species mass fractions across 98 timesteps spanning 34 microseconds. The temporal sequence is split into 70 timesteps for training, 14 for validation, and 14 for testing. Training resolution: $32^3$ grid; Test resolutions: $43^3$, $64^3$, and $129^3$. Low-resolution fields are obtained by uniformly coarsening the original $129^3$ DNS snapshots by factors of $2\times$, $3\times$, and $4\times$ in each spatial direction (i.e., $129^3 \rightarrow 64^3, 43^3, 32^3$) using spectral low-pass filtering followed by strided sampling. We choose $32^3$ as the training resolution because it is the coarsest grid that still preserves the main flow structures and allows stable training for all baselines, while maximizing the Nyquist gap between the training grid and the finest test grid.

**Weather Forecasting**: ERA5 contains global atmospheric reanalysis data across six vertical pressure levels (200–1000 hPa) (Hersbach et al., 2020; Copernicus Climate Change Service (C3S), 2017). Physical variables include temperature $T$ (K), horizontal wind components $(u, v)$ (m/s), and geopotential height $z$ ($m^2/s^2$). The dataset spans 360 days (January 2024–January 2025) with 6-hour temporal resolution, totaling 4320 timesteps, with the final 7 days reserved for validation. Training resolution: $180 \times 90 \times 6$; Test resolutions: $360 \times 180 \times 6$, $720 \times 361 \times 6$, and $1440 \times 721 \times 6$. Here the original ERA5 fields are stored on a $1440 \times 721 \times 6$ grid; we construct the low-resolution training data by regridding to $180 \times 90 \times 6$ via area-weighted averaging.

### 6.1.2 BASELINE SELECTION AND IMPLEMENTATION DETAILS

We selected the latest SOTA baselines for each architecture published in NeurIPS/ICLR/ICML. **Fluid Simulation:** Neural SPH (GNN), DeepLag (Transformer), PARCv2 (CNN), DYffusion (Diffusion), SFNO (NO), FNODE (Neural ODE), NeuralFluid (NN) (Toshev et al., 2024; Ma et al., 2024; Nguyen et al., 2024; Rühling Cachay et al., 2023; Cao et al., 2024; Li et al., 2024a;b). **Weather Forecasting:** WeatherGFT (Transformer), PDE-CNN (CNN), ARCI (Diffusion), Graph-EFM (GNN), ClimODE (Neural ODE) (Xu et al., 2024; Donà et al., 2020; Rühling Cachay et al., 2023; Oskarsson et al., 2024; Verma et al., 2024). FRL-enhanced baselines follow a unified training recipe; full hyperparameter settings are provided in Appendix C.

Notably, we primarily demonstrate FRL's architecture-agnostic Scale Decoupling capability here. The comparison of FRL with existing ZS-SR STF baselines is provided in Appendix G.

All experiments were conducted on a server equipped with four NVIDIA A100 (80GB) GPUs, using PyTorch 2.1 and CUDA 12.0. We use the AdamW optimizer (lr=$1 \times 10^{-3}$, weight_decay=$1 \times 10^{-5}$) with gradient clipping (max_norm=1.0), batch size 8, and sequence length 10. All models are trained for a maximum of 100 epochs with early stopping based on validation performance to prevent overfitting. To ensure reproducibility, we set the global random seed to 42.

### 6.1.3 METRICS

**Precision Metrics.** We employ standard error measures: The Root Mean Square Error (RMSE) and Mean Absolute Error (MAE) are computed as RMSE $= \sqrt{\frac{1}{N} \sum_{i=1}^{N} (u_i^{\text{pred}} - u_i^{\text{true}})^2}$ and MAE $= \frac{1}{N} \sum_{i=1}^{N} |u_i^{\text{pred}} - u_i^{\text{true}}|$, where $u_i^{\text{pred}}$ and $u_i^{\text{true}}$ denote predicted and ground truth values at grid point $i$, and $N$ is the total number of grid points. For weather forecasting, we compute RMSE and Anomaly Correlation Coefficient (ACC): ACC $= \frac{\sum_{i=1}^{N} (f_i' \cdot a_i')}{\sqrt{\sum_{i=1}^{N} (f_i')^2 \cdot \sum_{i=1}^{N} (a_i')^2}}$, where $f_i' = u_i^{\text{pred}} - \bar{u}_i$ and $a_i' = u_i^{\text{true}} - \bar{u}_i$ are anomalies relative to the climatological mean $\bar{u}_i$. ACC values above 0.6 indicate reliable forecasts. We evaluate four key variables at $500\,\text{hPa}$: geopotential height (Z500), temperature (T500), and horizontal wind components (U500, V500), as the $500\,\text{hPa}$ level best represents mid-tropospheric dynamics critical for weather systems. Most critically, we introduce $\text{RMSE}_{\text{Ratio}} = \text{RMSE}_{\text{high}}/\text{RMSE}_{\text{low}}$ to quantify Scale Anchoring severity.

**Frequency Metrics.** We adopt the frequency response analysis framework from Section 4.2. The magnitude of the empirical frequency response $H_{\text{mag}}(f) = A_{\text{out}}(f)/A_{\text{in}}(f)$ (output–to–input am-

plitude ratio). The Bandwidth (BW) represents the frequency at which $H(f)$ drops to 0.707 of its low-frequency value. The Anchoring Ratio $AR = H(f_{\text{Nyq}} - \delta)/H(f_{\text{Nyq}} + \delta)$ quantifies the sharpness of frequency cutoff near the Nyquist frequency $f_{\text{Nyq}} = \rho_0/2$ of the training resolution $\rho_0$, where we set $\delta = 4$ Hz. The Error Ratio $ER = E_{\text{bandlimited}}/E_{\text{wideband}}$ compares errors between signals limited to frequencies below the training Nyquist frequency and full-spectrum signals, revealing the contribution of high-frequency errors to total prediction error.

## 6.2 ZERO-SHOT SUPER-RESOLUTION SPATIOTEMPORAL FORECASTING

### 6.2.1 ZERO-SHOT SUPER-RESOLUTION FLUID SIMULATION

We report the main precision metric RMSE and frequency metric $H(f)$ here, and provide complete results in Appendix G.

Table 3: RMSE for 3D fluid simulation of baselines and FRL enhanced baselines.

| Method | $32^3$ | $43^3$ | $64^3$ | $129^3$ | RMSE$_{\text{Ratio}}$ |
|---|---|---|---|---|---|
| GNN (Neural SPH) | 0.00183 | 0.00183 | 0.00183 | 0.00187 | 1.018 |
| **GNN + FRL** | **0.00183** | **0.00101** | **0.00062** | **0.00032** | **0.175** |
| Transformer (DeepLag) | 0.00521 | 0.00524 | 0.00527 | 0.00532 | 1.021 |
| **Transformer + FRL** | **0.00521** | **0.00298** | **0.00185** | **0.00098** | **0.188** |
| CNN (PARCv2) | 0.00517 | 0.00523 | 0.00535 | 0.00548 | 1.060 |
| **CNN + FRL** | **0.00517** | **0.00261** | **0.00142** | **0.00071** | **0.137** |
| Diffusion (DYffusion) | 0.00294 | 0.00297 | 0.00301 | 0.00306 | 1.041 |
| **Diffusion + FRL** | **0.00294** | **0.00171** | **0.00108** | **0.00065** | **0.221** |
| NO (SFNO) | 0.00468 | 0.00470 | 0.00473 | 0.00476 | 1.017 |
| **NO + FRL** | **0.00468** | **0.00237** | **0.00128** | **0.00063** | **0.135** |
| Neural ODE (FNODE) | 0.00405 | 0.00432 | 0.00478 | 0.00542 | 1.338 |
| **Neural ODE + FRL** | **0.00405** | **0.00261** | **0.00195** | **0.00152** | **0.375** |
| NN (NeuralFluid) | 0.00374 | 0.00378 | 0.00383 | 0.00387 | 1.035 |
| **NN + FRL** | **0.00374** | **0.00206** | **0.00125** | **0.00068** | **0.182** |

Table 3 presents the error metrics for each architecture's baseline on 3D ZS-SR fluid simulation at different resolutions. For all baselines, the RMSE remains nearly constant or slightly increases as the resolution increases, with RMSE$_{\text{Ratio}}$ greater than 1. This indicates that all baselines exhibit Scale Anchoring. The FRL-enhanced baselines achieve $3.57 \times - 7.74 \times$ improvement at high resolution compared to the baselines, with RMSE$_{\text{Ratio}}$ reduced to 0.135-0.375. This demonstrates that FRL effectively achieves Scale Decoupling, enabling model accuracy to increase with rising resolution.

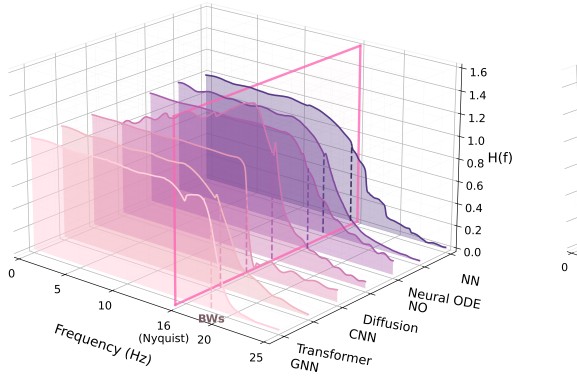

(a) Frequency response analysis for 3D fluid simulation of baselines.

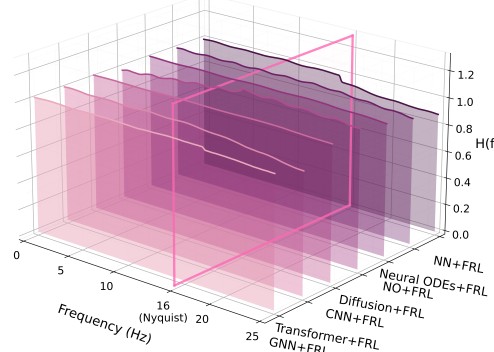

(b) Frequency response analysis for 3D fluid simulation of FRL enhanced baselines.

Figure 4

Figure 4 shows the frequency signal processing capabilities of each architecture's baseline and its FRL-enhanced version. It can be observed that the baselines' BWs are concentrated near the Nyquist frequency (16Hz), with the frequency response function $H(f)$ also dropping sharply around this point. This demonstrates that the baselines' signal processing capabilities for different frequencies are limited by the scale constraints of the training data. In contrast, the FRL-enhanced versions of the baselines exhibit stable processing capabilities across the entire frequency range, successfully achieving scale decoupling. Additionally, through FFT-based error separation, we find that the baselines have Error Ratios of 0.125-0.169, demonstrating the dominance of high-frequency component errors. Meanwhile, the FRL-enhanced versions of the baselines achieve Error Ratios of 0.4-0.556, proving that FRL improves overall accuracy by effectively reducing high-frequency errors.

Table 4: Z500 for ERA5 500 hPa 7-day forecast of baselines and FRL enhanced baselines.

| Method | Metric | $180 \times 90$ | $360 \times 180$ | $720 \times 361$ | $1440 \times 721$ | $\text{RMSE}_{\text{Ratio}}$ |
|---|---|---|---|---|---|---|
| Transformer (WeatherGFT) | RMSE | 685 | 692 | 708 | 721 | 1.053 |
| | ACC | 0.52 | 0.50 | 0.47 | 0.44 | |
| **Transformer + FRL** | **RMSE** | **685** | **572** | **518** | **485** | **0.708** |
| | **ACC** | **0.52** | **0.58** | **0.62** | **0.65** | |
| CNN (PDE-CNN) | RMSE | 692 | 701 | 718 | 738 | 1.066 |
| | ACC | 0.51 | 0.49 | 0.46 | 0.43 | |
| **CNN + FRL** | **RMSE** | **692** | **558** | **496** | **458** | **0.662** |
| | **ACC** | **0.51** | **0.59** | **0.63** | **0.66** | |
| Diffusion (ARCI) | RMSE | 672 | 678 | 688 | 695 | 1.034 |
| | ACC | 0.54 | 0.52 | 0.50 | 0.48 | |
| **Diffusion + FRL** | **RMSE** | **672** | **565** | **512** | **482** | **0.717** |
| | **ACC** | **0.54** | **0.59** | **0.62** | **0.64** | |
| GNN (Graph-EFM) | RMSE | 698 | 705 | 721 | 735 | 1.053 |
| | ACC | 0.50 | 0.48 | 0.45 | 0.42 | |
| **GNN + FRL** | **RMSE** | **698** | **578** | **528** | **498** | **0.713** |
| | **ACC** | **0.50** | **0.57** | **0.61** | **0.63** | |
| Neural ODE (ClimODE) | RMSE | 708 | 722 | 745 | 772 | 1.090 |
| | ACC | 0.49 | 0.47 | 0.44 | 0.41 | |
| **Neural ODE + FRL** | **RMSE** | **708** | **586** | **532** | **502** | **0.709** |
| | **ACC** | **0.49** | **0.56** | **0.60** | **0.63** | |

### 6.2.2 ZERO-SHOT SUPER-RESOLUTION WEATHER FORECASTING

We report the main precision metrics RMSE and ACC for Z500 hPa, and the frequency metric $H(f)$ here, with detailed results provided in Appendix G.

Table 4 presents the error metrics for each architecture's baseline on 3D ZS-SR weather forecasting at different resolutions. For all baselines, the RMSE errors are high and ACC consistently remains below 0.6, indicating unreliable predictions. Moreover, the RMSE remains nearly constant as resolution increases, with $\text{RMSE}_{\text{Ratio}}$ consistently greater than 1. This indicates that all baselines exhibit Scale Anchoring, indicating low-resolution trained models unusable. However, the FRL-enhanced baselines show significant improvement in accuracy at high resolutions. Notably, the ACC increases with resolution to above 0.6, representing successful and reliable weather predictions. This again demonstrates that FRL effectively achieves Scale Decoupling.

Figure 5 shows the frequency signal processing capabilities of each architecture's baseline and its FRL-enhanced version. The behavior pattern of the baselines' $H(f)$ is similar to that in Section 6.2.1, with signal processing capabilities for different frequencies still limited by the Nyquist frequency (90Hz) of the training data. In contrast, the FRL-enhanced versions similarly achieve Scale Decoupling. Additionally, through FFT-based error separation calculations, we find that the baselines have Error Ratios of 0.083-0.222, while the FRL-enhanced versions of the baselines achieve Error Ratios of 0.303-0.4, again demonstrating that FRL improves overall accuracy by effectively reducing high-frequency errors.

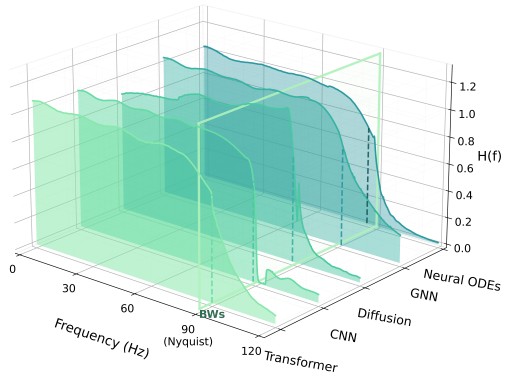 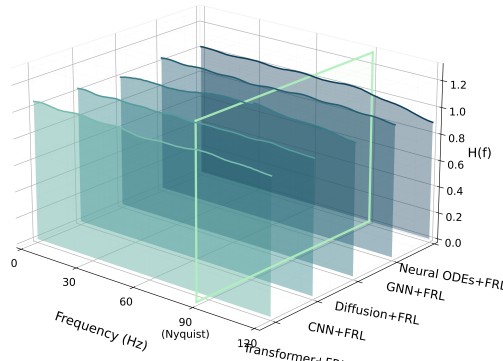

(a) Frequency response analysis for 3D weather fore-casting of baselines.

(b) Frequency response analysis for 3D weather fore-casting of FRL enhanced baselines.

Figure 5

### 6.2.3 COMPUTATIONAL OVERHEAD

We also evaluate the computational cost of FRL on the 3D ZS-SR fluid simulation task. Across all architectures, FRL-enhanced variants increase training wall-clock time by only about $1.1\times$–$1.4\times$ and peak training GPU memory by about $1.3\times$–$1.5\times$, while inference time overhead remains below $2\%$. These measurements are consistent with our complexity analysis in Appendix E, which shows that FRL preserves the asymptotic inference complexity $\mathcal{O}(M(n))$ of the backbone and adds only a small constant factor to training complexity and VRAM usage. Tables 10 and 11 provide architecture-wise breakdowns of the theoretical and empirical overhead.

## 7 DISCUSSION

Scale Anchoring has a clear mechanism, but its symptoms are often misinterpreted. While the Nyquist limitation is classical, its concrete manifestations and implications in STF have not been fully recognized. Meanwhile, prior cross-resolution methods were not explicitly designed to target Scale Anchoring in STF, and therefore typically did not directly address this limitation. For existing ZS-SR STF approaches, operator-learning models can be mathematically resolution-agnostic, yet in practice remain constrained by the training data's resolution: Scale Anchoring arises from the information bound imposed by fixed-resolution supervision rather than any particular architecture. This characteristic originating from the data observation itself also distinguishes Scale Anchoring from existing problems.

With FRL, we observe errors that decrease with resolution across tasks and models (Section 6; Appendix G), with low overhead (quantified in Appendices E and F). In addition, Appendix H reports (i) ablations of the three FRL components; (ii) drop-in replacement experiments that isolate the contribution of the normalized frequency encoding; and (iii) parameter sensitivity analyses.

Notably, FRL does not guarantee power-law error reduction like numerical solvers (as demonstrated in Appendix D) or effectiveness in arbitrary scenarios. In practice, the range over which extrapolation remains reliable is influenced by the operator's cross-resolution scale consistency, the smoothness of the learned frequency response within the training band, and model capacity. When small-scale physics undergo qualitative transitions (e.g., higher Reynolds number turbulence or extreme weather), extrapolation performance deteriorates. We analyze specific failure cases in Appendix I, specify the conditions under which FRL is effective and the boundaries of failure, and propose potential improvement strategies.

ACKNOWLEDGEMENTS

This work is supported by the National Natural Science Foundation of China (Grant No. 62372409) and the Ministry of Science and Technology of the People's Republic of China (Grant No. 2023ZD0120704 of Project No. 2023ZD0120700).

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

CONTENTS OF APPENDICES

## A    MATHEMATICAL PROOFS OF THEOREM 1 AND THEOREM 2

### A.1    PRELIMINARY DEFINITIONS AND ASSUMPTIONS

**Definition 1** (Domain and Sampling). Consider the unit domain $\Omega = [0,1]^d$ with periodic boundary conditions. For resolution $\rho$:

- Grid points: $\{x_j\}_{j=1}^{N_\rho}$ with $N_\rho = \rho^d$

- Sampling operator: $S_\rho : C(\Omega) \to \mathbb{R}^{N_\rho}$, where $[S_\rho(u)]_j = u(x_j)$

**Definition 2** (Interpolation Operators). The interpolation operator $I_{\rho_1 \to \rho_2} : \mathbb{R}^{N_{\rho_1}} \to \mathbb{R}^{N_{\rho_2}}$ via bilinear/bicubic interpolation satisfies:

- For $|k| \leq \min(\rho_1/2, \rho_2/2)$: $\widehat{I_{\rho_1 \to \rho_2}(v)}_k = \hat{v}_k$ (preserves low frequencies)

- For $|k| > \rho_1/2$: $\widehat{I_{\rho_1 \to \rho_2}(v)}_k = 0$ (cannot create new information)

**Assumption 1** (Physical Field Properties). The physical fields satisfy: $\sum_k |\hat{u}_k|^2 < \infty$ with non-negligible high-frequency content:

$$\liminf_{\rho \to \infty} \frac{\sum_{\rho_0/2 < |k| \leq \rho/2} |\hat{u}_k|^2}{\sum_{|k| \leq \rho_0/2} |\hat{u}_k|^2} > c > 0 \tag{9}$$

This assumption is satisfied by typical physical fields with multi-scale structure, including turbulent flows and atmospheric dynamics.

**Assumption 2** (Convergent Training). The network $G_\Theta : \mathbb{R}^{N_{\rho_0}} \to \mathbb{R}^{N_{\rho_0}}$ is trained to convergence via gradient descent on:

$$L(\Theta) = \mathbb{E}_{u \sim \mathcal{D}} \|G_\Theta(S_{\rho_0}(u)) - S_{\rho_0}(\mathcal{F}[u])\|^2 \tag{10}$$

where $\mathcal{F}$ is the true evolution operator.

### A.2    PROOF OF THEOREM 1 (FREQUENCY BLINDNESS)

**Theorem 3** (Frequency Blindness). *A neural network trained at resolution $\rho_0$ cannot learn the correct mapping for frequencies above $\rho_0/2$:*

$$\hat{G}_{\Theta_{\rho_0}}(\omega) \approx \begin{cases} \hat{\mathcal{F}}(\omega) + \epsilon(\omega), & \omega \leq \rho_0/2 \\ uncorrelated\ with\ \hat{\mathcal{F}}(\omega), & \omega > \rho_0/2 \end{cases} \tag{11}$$

*Proof.* **Step 1: Training Objective in Frequency Domain.** The training loss can be written as:

$$L(\Theta) = \sum_{|k| \leq \rho_0/2} \mathbb{E}_u \left[ |\hat{G}_\Theta(k) - \hat{\mathcal{F}}(k)|^2 |\hat{u}_k|^2 \right] \tag{12}$$

**Step 2: Absence of High-Frequency Gradient Signal.** For any $k$ with $|k| > \rho_0/2$:

- The sampled input $S_{\rho_0}(u)$ aliases this frequency to $k' = k \mod \rho_0$

- Since $L$ only depends on frequencies up to $\rho_0/2$, we have $\frac{\partial L}{\partial \hat{G}_\Theta(k)} = 0$ for all $|k| > \rho_0/2$

**Step 3: Learning Outcome.** Under gradient-based optimization:

- For $|k| \leq \rho_0/2$: The network learns $\hat{G}_\Theta(k) \to \hat{\mathcal{F}}(k)$ up to approximation error $\epsilon(k)$

- For $|k| > \rho_0/2$: There exists no learning mechanism to correlate $\hat{G}_\Theta(k)$ with $\hat{\mathcal{F}}(k)$

Therefore, the network output for high frequencies is effectively uncorrelated with the true operator. ☐

A.3   PROOF OF THEOREM 2 (HIGH-FREQUENCY ERROR DOMINANCE)

**Theorem 4** (High-Frequency Error Dominance). *When deployed at resolution $\rho > \rho_0$, the Scale Anchoring error bound*

$$C = \lim_{\rho \to \infty} \|G^{(\rho)}_{\Theta_{\rho_0}} \circ S_\rho - S_\rho \circ \mathcal{F}\|_{op} \tag{13}$$

*is dominated by the network's inability to process frequency components in the range $[\rho_0/2, \rho/2]$.*

*Proof.* **Step 1: Network Deployment at Higher Resolution.** At resolution $\rho$, the deployed network is:

$$G^{(\rho)}_{\Theta_{\rho_0}} = I_{\rho_0 \to \rho} \circ G_\Theta \circ I_{\rho \to \rho_0} \tag{14}$$

**Step 2: Error Decomposition.** The squared error decomposes as:

$$E^2 = \sum_{|k| \le \rho/2} |\hat{G}^{(\rho)}_{\Theta_{\rho_0}}(k) - \hat{\mathcal{F}}(k)|^2 |\hat{u}_k|^2 \tag{15}$$

Split into frequency bands:

$$E^2_L = \sum_{|k| \le \rho_0/2} |\epsilon(k)|^2 |\hat{u}_k|^2 \tag{16}$$

$$E^2_M = \sum_{\rho_0/2 < |k| \le \rho/2} |\hat{G}^{(\rho)}_{\Theta_{\rho_0}}(k) - \hat{\mathcal{F}}(k)|^2 |\hat{u}_k|^2 \tag{17}$$

**Step 3: Mid-Frequency Error Analysis.** By Theorem 3 and the interpolation properties:

- $\hat{G}^{(\rho)}_{\Theta_{\rho_0}}(k) \approx 0$ for $\rho_0/2 < |k| \le \rho/2$ (no learned representation)

- $\hat{\mathcal{F}}(k) \ne 0$ (true operator acts on these frequencies)

Therefore:

$$E^2_M \approx \sum_{\rho_0/2 < |k| \le \rho/2} |\hat{\mathcal{F}}(k)|^2 |\hat{u}_k|^2 \tag{18}$$

**Step 4: Error Dominance.** Under Assumption 1, where $c > 0$ is the constant from the assumption:

$$\lim_{\rho \to \infty} \frac{E^2_M}{E^2_L + E^2_M} = \lim_{\rho \to \infty} \frac{E^2_M/E^2_L}{1 + E^2_M/E^2_L} = \frac{c}{1+c} > 0 \tag{19}$$

Thus the error bound $C$ remains positive, dominated by unlearned frequencies. □

A.4   IMPLICATIONS FOR SCALE ANCHORING

**Corollary 5** (RMSE$_{\text{Ratio}}$ Behavior).

$$RMSE_{Ratio} = \frac{RMSE_\rho}{RMSE_{\rho_0}} \to constant \approx 1 \quad as \; \rho \to \infty \tag{20}$$

*Proof.* Both RMSE values are dominated by the network's inability to process frequencies above $\rho_0/2$:

$$\text{RMSE}^2_{\rho_0} \approx \sum_{|k| \le \rho_0/2} |\epsilon(k)|^2 |\hat{u}_k|^2 \tag{21}$$

$$\text{RMSE}^2_\rho \approx \sum_{|k| \le \rho_0/2} |\epsilon(k)|^2 |\hat{u}_k|^2 + \sum_{\rho_0/2 < |k| \le \rho/2} |\hat{\mathcal{F}}(k)|^2 |\hat{u}_k|^2 \tag{22}$$

Since the network's frequency processing capability remains fixed at $\rho_0/2$ regardless of deployment resolution:

$$\text{RMSE}_{\text{Ratio}} = \sqrt{\frac{\text{RMSE}_\rho^2}{\text{RMSE}_{\rho_0}^2}} = \sqrt{1 + \frac{E_M^2}{E_L^2}} \rightarrow \text{constant} \tag{23}$$

This contrasts fundamentally with numerical methods where:

$$\text{RMSE}_{\text{Ratio}}{}^{\text{numerical}} = \left(\frac{\Delta x_\rho}{\Delta x_{\rho_0}}\right)^p = \left(\frac{\rho_0}{\rho}\right)^p \rightarrow 0 \tag{24}$$

This persistent error ratio characterizes Scale Anchoring. $\qquad\qquad\square$

## B  DETAILS AND COMPLETE RESULTS OF VALIDATION EXPERIMENT A.1

**Data Generation Method.** We employ pseudo-spectral methods to solve the 2D convection-diffusion equation in the frequency domain for generating high-fidelity training data. The governing equation is given by $\frac{\partial u}{\partial t} + \mathbf{v} \cdot \nabla u = \nu\nabla^2 u + f$, where $u(x, y, t)$ represents the physical field, $\mathbf{v} = (1.0, 0.5)$ is the convection velocity, $\nu = 0.01$ is the diffusion coefficient, and $f$ is a forcing term. The solution process involves transforming to the frequency domain via 2D FFT, where the convection term becomes $\mathcal{F}[\mathbf{v} \cdot \nabla u] = i\mathbf{k} \cdot \mathbf{v}\hat{u}$ and the diffusion term becomes $\mathcal{F}[\nu\nabla^2 u] = -\nu|\mathbf{k}|^2\hat{u}$. We use fourth-order Runge-Kutta time integration with timestep $\Delta t = 0.001$ and periodic boundary conditions. The dataset consists of 1000 independent trajectories, each containing 100 timesteps, with initial conditions generated from random Gaussian fields superimposed with low-frequency sinusoidal components to ensure rich dynamical behavior.

**Base Model Implementation.** We implement eight fundamental architectures commonly used in spatiotemporal forecasting. The GNN employs a message-passing framework with 4 graph convolution layers and hidden dimension 128. The Transformer uses 4 self-attention layers with 8 attention heads and model dimension 256. The CNN adopts a U-Net structure with 4 encoder-decoder levels, kernel size 3, and channel dimensions [64, 128, 256, 512]. The Diffusion model implements a denoising score-matching framework with 100 diffusion steps and a U-Net backbone. The Neural Operator learns in the frequency domain with 4 Fourier layers and 12 retained modes per dimension. The Neural ODE parameterizes the dynamics using a 3-layer MLP with hidden dimension 256, integrated using the dopri5 solver. The Mamba model uses selective state-space layers with state dimension 16 and expansion factor 2. The standard NN consists of 5 fully-connected layers with 512 hidden units each and ReLU activations.

**Training Method.** All experiments were conducted on a server equipped with four NVIDIA A100 (80GB) GPUs, using PyTorch 2.1 and CUDA 12.0. We use the AdamW optimizer (lr=$1 \times 10^{-3}$, weight_decay=$1 \times 10^{-5}$) with gradient clipping (max_norm=1.0), batch size 8, and sequence length 10. All models are trained for a maximum of 100 epochs with early stopping based on validation performance to prevent overfitting. To ensure reproducibility, we set the global random seed to 42.

**Experimental Procedure.** Following the training phase at specified resolutions (64×64 shown in main text, with additional resolutions 32×32, 128×128, and 256×256 in this appendix), we conduct systematic frequency response analysis. We construct sinusoidal test signals $u(x, y, t = 0) = A \cdot \sin(2\pi f \cdot x)$ with unit amplitude $A = 1.0$ and frequencies $f$ ranging from 0 to 50 Hz, specifically sampling at intervals to capture the transition around the Nyquist frequency. Each test signal is propagated through the trained models for one timestep, and we measure the output amplitude to compute the frequency response function $H(f) = A_{output}/A_{input}$. The Bandwidth is determined as the frequency where $H(f)$ drops to 0.707 of its low-frequency value. To quantify the sharpness of the frequency cutoff, we calculate the Anchoring Ratio as $H(f_{Nyquist} - 2)/H(f_{Nyquist} + 2)$, where $f_{Nyquist}$ corresponds to half the training resolution. Each frequency point is tested 10 times with different random initializations to ensure statistical significance. The results for models trained at 64×64 resolution are presented in Figure 2, demonstrating the universal cliff-like drop in frequency response near the Nyquist frequency across all architectures. Complete results for models trained at 32×32, 64×64, 128×128, and 256×256 resolutions are provided in Tables 5, 6, 7, and 8 respectively, showing consistent scale anchoring behavior with Anchoring Ratios ranging from 1.40 to 63.1 across different architectures and resolutions, thereby validating Theorem 1.

Table 5: Frequency response analysis on different architectures, models trained at 32×32.

| Model | H(f=8) | H(f=14) | H(f=18) | Bandwidth (Hz) | H(Nyquist) (f=16) | Anchoring Ratio |
|---|---|---|---|---|---|---|
| GNN | 0.935 | 0.530 | 0.182 | 15.5 | 0.432 | 2.91 |
| Transformer | 0.928 | 0.515 | 0.178 | 15.2 | 0.425 | 2.89 |
| CNN | 0.940 | 0.545 | 0.095 | 15.5 | 0.440 | 5.74 |
| Diffusion | 0.795 | 0.835 | 0.298 | 15.8 | 0.720 | 2.80 |
| Neural Operator | 0.915 | 0.645 | 0.205 | 15.7 | 0.430 | 3.15 |
| Neural ODE | 0.865 | 0.490 | 0.108 | 15.0 | 0.410 | 4.54 |
| Mamba | 0.920 | 0.505 | 0.008 | 15.3 | 0.420 | 63.1 |
| SimpleNN | 0.770 | 0.385 | 0.052 | 12.5 | 0.375 | 7.40 |

Table 6: Frequency response analysis on different architectures, models trained at 64×64.

| Model | H(f=16) | H(f=30) | H(f=34) | Bandwidth (Hz) | H(Nyquist) (f=32) | Anchoring Ratio |
|---|---|---|---|---|---|---|
| GNN | 0.865 | 0.520 | 0.356 | 31.5 | 0.438 | 1.46 |
| Transformer | 0.850 | 0.510 | 0.346 | 31.0 | 0.428 | 1.47 |
| CNN | 0.880 | 0.540 | 0.185 | 31.5 | 0.362 | 2.92 |
| Diffusion | 0.720 | 0.820 | 0.584 | 31.5 | 0.702 | 1.40 |
| Neural Operator | 0.859 | 0.635 | 0.401 | 31.5 | 0.518 | 1.58 |
| Neural ODE | 0.820 | 0.480 | 0.210 | 30.5 | 0.345 | 2.29 |
| Mamba | 0.840 | 0.500 | 0.016 | 31.0 | 0.258 | 31.3 |
| SimpleNN | 0.750 | 0.380 | 0.101 | 26.0 | 0.240 | 3.76 |

Table 7: Frequency response analysis on different architectures, models trained at 128×128.

| Model | H(f=32) | H(f=60) | H(f=68) | Bandwidth (Hz) | H(Nyquist) (f=64) | Anchoring Ratio |
|---|---|---|---|---|---|---|
| GNN | 0.845 | 0.505 | 0.340 | 63.0 | 0.422 | 1.49 |
| Transformer | 0.830 | 0.495 | 0.330 | 62.5 | 0.412 | 1.50 |
| CNN | 0.860 | 0.525 | 0.175 | 63.0 | 0.350 | 3.00 |
| Diffusion | 0.700 | 0.810 | 0.570 | 63.5 | 0.690 | 1.42 |
| Neural Operator | 0.839 | 0.625 | 0.390 | 63.2 | 0.507 | 1.60 |
| Neural ODE | 0.800 | 0.470 | 0.200 | 62.0 | 0.335 | 2.35 |
| Mamba | 0.820 | 0.490 | 0.015 | 62.5 | 0.252 | 32.7 |
| SimpleNN | 0.730 | 0.370 | 0.095 | 52.0 | 0.232 | 3.89 |

Table 8: Frequency response analysis on different architectures, models trained at 256×256.

| Model | H(f=64) | H(f=120) | H(f=136) | Bandwidth (Hz) | H(Nyquist) (f=128) | Anchoring Ratio |
|---|---|---|---|---|---|---|
| GNN | 0.825 | 0.490 | 0.325 | 126.5 | 0.407 | 1.51 |
| Transformer | 0.810 | 0.480 | 0.315 | 125.5 | 0.397 | 1.52 |
| CNN | 0.840 | 0.510 | 0.165 | 126.5 | 0.337 | 3.09 |
| Diffusion | 0.680 | 0.800 | 0.555 | 127.0 | 0.677 | 1.44 |
| Neural Operator | 0.819 | 0.615 | 0.380 | 126.8 | 0.497 | 1.62 |
| Neural ODE | 0.780 | 0.460 | 0.190 | 124.5 | 0.325 | 2.42 |
| Mamba | 0.800 | 0.480 | 0.014 | 125.5 | 0.246 | 34.3 |
| SimpleNN | 0.710 | 0.360 | 0.090 | 104.0 | 0.225 | 4.00 |

# C  PSEUDOCODE AND IMPLEMENTATION DETAILS OF FREQUENCY REPRESENTATION LEARNING

## C.1  PSEUDOCODE FREQUENCY REPRESENTATION LEARNING

---

**Algorithm 1** Frequency Representation Learning for ZS-SR STF

---

**Require:** Time series data $\{\mathbf{u}(t)\}$ at resolution $\rho_0$, forecast model $\mathcal{M}_\Theta$
**Ensure:** Model $\mathcal{M}_\Theta$ capable of forecasting at arbitrary resolutions

1:
2: **// Phase 1: Multi-Resolution Training Data Generation**
3: $\mathcal{D} \leftarrow \{\}$  ▷ *Multi-scale training set*
4: **for** $j = 0$ to $J - 1$ **do**
5:    $\rho_j \leftarrow \rho_0/2^j$  ▷ *Resolution hierarchy*
6:    $\{\mathbf{u}^{(\rho_j)}(t)\} \leftarrow \text{Downsample}(\{\mathbf{u}(t)\}, 2^j)$
7:    $\mathcal{D}_{\rho_j} \leftarrow \{(\mathbf{u}^{(\rho_j)}(t), \mathbf{u}^{(\rho_j)}(t + \Delta t))\}$  ▷ *Time evolution pairs*
8:    $\mathcal{D} \leftarrow \mathcal{D} \cup \mathcal{D}_{\rho_j}$
9:
10: **// Phase 2: Frequency-Aware Model Architecture**
11: Initialize spatiotemporal predictor $\mathcal{M}_\Theta : \mathbf{u}(t) \rightarrow \mathbf{u}(t + \Delta t)$
12:
13: **// Phase 3: Scale-Decoupled Training**
14: **for** epoch $= 1$ to $N_{epochs}$ **do**
15:    **for** each resolution $\rho \in \{\rho_0/2^{J-1}, ..., \rho_0/2, \rho_0\}$ **do**
16:       **for** $(\mathbf{u}^{(\rho)}(t), \mathbf{u}^{(\rho)}(t + \Delta t)) \in \mathcal{D}_\rho$ **do**
17:
18:          **// Frequency-Normalized Encoding**
19:          $k_{Nyq}(\rho) \leftarrow \rho/2$  ▷ *Resolution-specific Nyquist*
20:          **for** each spatial position $x$ **do**
21:             $\mathbf{PE}(x, \rho) \leftarrow \{\sin(2\pi k \cdot x/k_{Nyq}(\rho))\}_k$  ▷ *Normalized by Nyquist*
22:
23:          **// Forward Prediction with Frequency Awareness**
24:          $\mathbf{u}_{enc} \leftarrow [\mathbf{u}^{(\rho)}(t), \mathbf{PE}(\cdot, \rho)]$  ▷ *Concatenate features and PE*
25:          $\hat{\mathbf{u}}^{(\rho)}(t + \Delta t) \leftarrow \mathcal{M}_\Theta(\mathbf{u}_{enc})$
26:
27:          **// Frequency-Decomposed Loss**
28:          $\hat{\mathbf{U}} \leftarrow \text{FFT}(\hat{\mathbf{u}}^{(\rho)}(t + \Delta t))$  ▷ *Predicted spectrum*
29:          $\mathbf{U} \leftarrow \text{FFT}(\mathbf{u}^{(\rho)}(t + \Delta t))$  ▷ *Target spectrum*
30:
31:          **// Multi-Scale Loss Components**
32:          $\mathcal{L}_{space} \leftarrow \|\hat{\mathbf{u}}^{(\rho)}(t + \Delta t) - \mathbf{u}^{(\rho)}(t + \Delta t)\|^2$
33:          $\mathcal{L}_{freq} \leftarrow \sum_k w_k(\rho)\|\hat{\mathbf{U}}_k - \mathbf{U}_k\|^2$  ▷ *Frequency-weighted*
34:          $\mathcal{L}_{phys} \leftarrow \text{PhysicsConstraints}(\hat{\mathbf{u}}^{(\rho)}(t + \Delta t))$
35:
36:          $\mathcal{L} \leftarrow \mathcal{L}_{space} + \lambda\mathcal{L}_{freq} + \mu\mathcal{L}_{phys}$
37:          $\Theta \leftarrow \Theta - \eta\nabla_\Theta\mathcal{L}$
38:
39: **// Phase 4: Zero-Shot Inference at Arbitrary Resolution**
40: **procedure** FORECAST($\mathbf{u}^{(\rho_{new})}(t_0), T, \rho_{new}$)
41:    ▷ *$\rho_{new}$ can be ANY resolution, including unseen high resolutions*
42:    $\mathbf{u} \leftarrow \mathbf{u}^{(\rho_{new})}(t_0)$
43:    **for** $t = t_0$ to $t_0 + T$ **do**
44:       $\mathbf{PE} \leftarrow \{\sin(2\pi k \cdot x/k_{Nyq}(\rho_{new}))\}_k$  ▷ *Adapt PE to new resolution*
45:       $\mathbf{u} \leftarrow \mathcal{M}_\Theta([\mathbf{u}, \mathbf{PE}])$  ▷ *Predict next timestep*
46:    **return** $\mathbf{u}$
47:
48: **return** $\mathcal{M}_\Theta$

---

Algorithm 1 presents the complete FRL framework for addressing Scale Anchoring in ZS-SR STF. The algorithm consists of four main phases that progressively build the capability to perform resolution-invariant predictions.

In **Phase 1** (lines 3-8), we construct multi-resolution training data pairs through recursive downsampling. Starting from the original resolution $\rho_0$, we create a hierarchy of resolutions $\rho_j = \rho_0/2^j$ for $j \in \{0, ..., J-1\}$, as described in Step 1 and Equation equation 6. Each downsampled dataset $\mathcal{D}_{\rho_j}$ contains temporal evolution pairs $(u^{(\rho_j)}(t), u^{(\rho_j)}(t + \Delta t))$ that capture the physical dynamics at different scales. This multi-scale construction is crucial because it enables the network to learn the conditional distribution $P(u_{high}|u_{low})$, where the difference $u^{(\rho_0)} - \mathcal{U}[u^{(\rho_0/2)}]$ naturally isolates frequency components in the band $[f_{Nyq}(\rho_0/2), f_{Nyq}(\rho_0)]$.

**Phase 2** (line 11) initializes the spatiotemporal predictor $M_\Theta$ that will learn resolution-invariant mappings. The architecture can be any standard neural network, as our method is architecture-agnostic.

The core of our approach lies in **Phase 3** (lines 14-37), which implements scale-decoupled training. For each resolution $\rho$ in our multi-scale dataset, we first compute normalized frequency encodings using Equation equation 7, where $PE(x, \rho) = \{\sin(2\pi k \cdot x/k_{Nyq}(\rho))\}_k$ ensures that the same physical frequency receives identical representations across different resolutions. This normalization by the resolution-specific Nyquist frequency $k_{Nyq}(\rho) = \rho/2$ is essential for achieving the resolution invariance property shown in Step 2. The model then performs forward prediction with these frequency-aware features concatenated to the input (line 24).

The training objective combines three loss components as defined in Equation equation 8. The spatial reconstruction loss $\mathcal{L}_{space}$ ensures overall accuracy, while the frequency-weighted loss $\mathcal{L}_{freq} = \sum_k w_k(\rho)\|\hat{U}_k - U_k\|^2$ enforces frequency consistency in the spectral domain. The key innovation is that this frequency loss preserves low-frequency components below $k_{Nyq}(\rho_0/2)$ while teaching the network to conditionally generate higher frequencies based on low-frequency structure. Additional physics constraints $\mathcal{L}_{phys}$ can be incorporated to maintain physical consistency.

Finally, **Phase 4** (lines 40-46) demonstrates zero-shot inference at arbitrary resolutions. Given an initial condition at any resolution $\rho_{new}$ (including unseen high resolutions), the trained model can be recursively applied: $u^{(2^n\rho_0)} = F_\Theta^{(n)} \circ \cdots \circ F_\Theta^{(1)}(u^{(\rho_0)})$. At each application, the position encodings are adapted to the new resolution (line 44), enabling the model to correctly process frequencies beyond the training Nyquist limit and effectively decouple from Scale Anchoring.

## C.2 IMPLEMENTATION DETAILS AND HYPERPARAMETERS OF FREQUENCY REPRESENTATION LEARNING

| Component | Setting |
|---|---|
| Resolution levels | $J = 3$ levels $\{\rho_0, \rho_1, \rho_2\}$ with $\rho_{j+1} = \rho_j/2$. |
| Multi-resolution mixing | Per-batch sampling over $\{\rho_0, \rho_1\}$ with $p(\rho_0) = p(\rho_1) = 0.5$; per-level loss weights $w_{\rho_0} = w_{\rho_1} = 1$. |
| Downsampling (anti-alias) | FFT low-pass (frequency center-crop) + IFFT. |
| Sinusoidal PE (Nyquist-normalized) | Axes-wise sin / cos scaled by the per-resolution Nyquist; number of harmonics $n_{\text{freq}} = 8$. |
| Frequency-domain loss | Amplitude-space MSE with radial weight exponent $\alpha = 1.0$; global weight $\lambda = 0.1$ with linear warmup during first 5 epochs. |
| Evaluation protocol | Equal physical-time free-rollout when reporting RMSE and RMSE Ratio (e.g., horizon $\approx 10$ steps at $\rho_0$). |

Table 9: Hyperparameters of FRL.

## D NORMALIZED-FREQUENCY ERROR ANALYSIS OF FREQUENCY REPRESENTATION LEARNING

This appendix provides a simple frequency-domain argument that explains why FRL can reduce high-resolution error without guaranteeing strict convergence in the sense of numerical analysis. The goal is not to prove an order convergence theorem, but to make explicit how normalized-frequency learning allows models to mitigate the high-frequency component of Scale Anchoring.

### D.1  SETUP AND EQUIVALENT LINEAR RESPONSE APPROXIMATION

For clarity, we consider a one-dimensional spatial domain; the extension to higher dimensions is analogous. Let $u(x)$ be a continuous field with Fourier transform $\widehat{u}(k)$, where $k$ denotes the spatial wavenumber. For a grid with spacing $\Delta x$, the Nyquist wavenumber is

$$k_N(\Delta x) = \frac{\pi}{\Delta x}.$$

We assume that, after training, the model $F_\Theta$ can be approximated in the frequency domain by an *equivalent linear response*:

$$\widehat{F_\Theta(u)}(k; \Delta x) \approx H_\Theta(\xi)\, \widehat{u}(k), \qquad \xi = \frac{|k|}{k_N(\Delta x)} \in [0, 1], \tag{25}$$

where $\xi$ is the normalized wavenumber and $H_\Theta(\xi)$ is a complex-valued gain function that is *approximately cross-scale consistent* up to a small deviation $\varepsilon$, in the sense that its dependence on $\Delta x$ is weak once expressed in normalized coordinates.

This approximation is standard in spectral analysis of linear and weakly nonlinear systems: $H_\Theta$ captures how the model amplifies or attenuates different frequency components of the input.

### D.2  NORMALIZED SPECTRUM AND ERROR DECOMPOSITION

At a given test resolution with grid spacing $\Delta x'$, let $S_u(k)$ denote the power spectral density (PSD) of $u$. We define the normalized PSD with respect to the Nyquist wavenumber $k_N(\Delta x')$ by the change of variables

$$k = k_N(\Delta x')\, \xi, \qquad dk = k_N(\Delta x')\, d\xi,$$

and set

$$S_u^{(\Delta x')}(\xi) = k_N(\Delta x')\, S_u\big(k_N(\Delta x')\, \xi\big), \qquad \xi \in [0, 1]. \tag{26}$$

Using the equivalent linear response approximation equation 25, the dominant term of the mean-squared error (MSE) in the frequency domain at resolution $\Delta x'$ can be written as

$$\mathrm{MSE}(\Delta x') \approx \int_0^1 \big| H_\Theta(\xi) - 1 \big|^2 S_u^{(\Delta x')}(\xi)\, d\xi \; + \; \mathcal{R}_{\text{aleatoric}} \; + \; O(\varepsilon), \tag{27}$$

where:

- the first term is the *calibration error* that can, in principle, be reduced through learning;
- $\mathcal{R}_{\text{aleatoric}}$ is the conditional variance of the high-resolution field given a fixed low-resolution observation, capturing the irreducible uncertainty arising from the non-uniqueness of the coarse-to-fine mapping;
- $O(\varepsilon)$ collects higher-order terms due to the approximate cross-scale consistency of $H_\Theta$.

Thus, even with perfect calibration ($H_\Theta \approx 1$), the error cannot be driven below $\mathcal{R}_{\text{aleatoric}}$, reflecting the inherent ill-posedness of zero-shot super-resolution.

### D.3  FRL AND OUT-OF-BAND FREQUENCIES

We now compare training on a coarse grid with spacing $\Delta x_{\text{tr}}$ and testing on a finer grid with spacing $\Delta x' < \Delta x_{\text{tr}}$. In this case,

$$k_N(\Delta x_{\text{tr}}) < k_N(\Delta x'),$$

and we define the ratio

$$\rho = \frac{k_N(\Delta x_{\text{tr}})}{k_N(\Delta x')} \in (0, 1). \tag{28}$$

All absolute frequencies $k$ that lie above the training Nyquist but below the test Nyquist satisfy

$$k \in \big( k_N(\Delta x_{\text{tr}}),\, k_N(\Delta x') \big],$$

and correspond to normalized coordinates on the test grid

$$\xi' = \frac{|k|}{k_N(\Delta x')} \in (\rho, 1] \subset (0, 1]. \tag{29}$$

Therefore, these *out-of-training-band* absolute frequencies fall back into the normalized domain $[0, 1]$ at test time and can, in principle, be calibrated by the learned gain function $H_\Theta(\xi)$ in the normalized frequency domain.

This is precisely what FRL attempts to exploit: by learning a cross-scale-consistent $H_\Theta$ in normalized coordinates, FRL makes "absolute frequencies outside the training band" become "learnable points within the normalized band" on finer grids.

## D.4 RELATIVE IMPROVEMENT OVER AN ANCHORED BASELINE

To quantify the potential improvement, consider the fraction of spectral energy at test resolution that lies outside the training Nyquist:

$$f_{\text{OOB}}(\rho) = \frac{\displaystyle\int_\rho^1 S_u^{(\Delta x')}(\xi)\,d\xi}{\displaystyle\int_0^1 S_u^{(\Delta x')}(\xi)\,d\xi}. \tag{30}$$

By construction, $f_{\text{OOB}}(\rho) \in [0, 1]$; larger values mean that a larger portion of the test-spectrum energy resides above the training Nyquist frequency.

Assume that, on the out-of-band interval $(\rho, 1]$, FRL learns a gain function that remains close to identity:

$$\sup_{\xi \in (\rho, 1]} \left| H_\Theta(\xi) - 1 \right| \leq \delta, \tag{31}$$

for some small $\delta > 0$. In contrast, an "anchored" baseline that is blind to frequencies above the training Nyquist can be approximated by

$$H_{\text{anch}}(\xi) \approx \mathbf{1}_{[0,\rho]}(\xi), \tag{32}$$

i.e., it behaves like the identity on normalized frequencies up to $\rho$ and ignores all components beyond.

Under these assumptions, the ratio between the MSE of FRL and that of the anchored baseline can be bounded (up to $O(\varepsilon)$ and the irreducible term) as

$$\frac{\text{MSE}_{\text{FRL}}}{\text{MSE}_{\text{anch}}} \lesssim 1 - \left(1 - \delta^2\right) f_{\text{OOB}}(\rho) + O(\varepsilon) + \frac{\mathcal{R}_{\text{aleatoric}}}{\text{MSE}_{\text{anch}}}. \tag{33}$$

This inequality is *not* an order convergence theorem: it does not state that the error decays at a fixed rate as $\Delta x' \to 0$. Instead, it has the following interpretation:

- the larger the energy fraction $f_{\text{OOB}}(\rho)$, the more room there is for improvement by correctly modeling out-of-band frequencies;

- the smaller the deviation $\delta$ (i.e., the closer $H_\Theta(\xi)$ is to 1 on $(\rho, 1]$), the larger the potential relative error reduction;

- the term $\mathcal{R}_{\text{aleatoric}}/\text{MSE}_{\text{anch}}$ provides a residual lower bound due to the inherent ill-posedness of mapping from low-resolution to high-resolution fields.

In other words, FRL cannot guarantee strict order convergence like numerical solvers, but by learning a normalized, cross-scale-consistent frequency response, it can systematically mitigate the high-frequency component of Scale Anchoring whenever the underlying physical system exhibits sufficient spectral regularity across scales.

# E  COMPUTATIONAL AND TRAINING COMPLEXITY ANALYSIS OF FREQUENCY REPRESENTATION LEARNING

We analyze the computational overhead introduced by FRL compared to baseline methods, considering both inference and training phases. Let $n = \rho^d$ denote the total number of grid points for a $d$-dimensional domain with resolution $\rho$ per dimension.

## E.1  INFERENCE COMPLEXITY

For a baseline model $M_\Theta$ with forward pass complexity $\mathcal{O}(M(n))$, the FRL-enhanced inference adds minimal overhead:

**Baseline:** $\mathcal{O}(M(n))$

**FRL-Enhanced:** $\mathcal{O}(M(n) + nK)$

The additional $\mathcal{O}(nK)$ term arises from computing normalized position encodings (Algorithm 1, line 44), where $K$ denotes the number of frequency modes. Since $K \ll n$ and sinusoidal computations are negligible compared to deep network operations, the practical inference complexity remains effectively unchanged: $\mathcal{O}(M(n))$.

## E.2  TRAINING COMPLEXITY

The training complexity analysis must distinguish between model forward pass and loss computation overhead.

**Baseline Training Complexity:**
$$\mathcal{O}(E \cdot B \cdot M(n))$$
where $E$ denotes training epochs, $B$ denotes batches, and $M(n)$ represents the model's forward pass complexity.

**FRL Training Complexity:**

$$\mathcal{O}\left(E \cdot B \cdot \left[\sum_{j=0}^{J-1} M(n_j) + \sum_{j=0}^{J-1} n_j \log n_j\right]\right)$$

where $n_j = \rho^d/2^{jd}$ represents grid points at resolution level $j$. The two terms represent:

**1. Multi-Resolution Forward Passes:** For 3D domains ($d = 3$), the total forward pass cost is:

$$\sum_{j=0}^{J-1} M(n_j) = M(n) + M(n/8) + M(n/64) + ...$$

The actual overhead depends critically on $M$'s complexity:

- Linear $M(n) = \mathcal{O}(n)$: Total $\approx 1.14 \cdot M(n)$
- Quadratic $M(n) = \mathcal{O}(n^2)$: Total $\approx 1.02 \cdot M(n)$
- Log-linear $M(n) = \mathcal{O}(n \log n)$: Total $\approx 1.10 \cdot M(n)$

**2. FFT for Loss Computation:** Two FFTs per resolution for frequency loss:

$$\sum_{j=0}^{J-1} n_j \log n_j < \frac{8}{7} n \log n$$

However, this overhead is typically negligible compared to model forward passes in modern deep architectures.

**Effective Complexity Ratio:** The practical training overhead varies significantly by architecture:

$$R = \begin{cases} \approx 1.14 & \text{for CNNs with } M(n) = \mathcal{O}(n) \\ \approx 1.02 & \text{for Transformers with } M(n) = \mathcal{O}(n^2) \\ \approx 1.10 & \text{for Spectral methods with } M(n) = \mathcal{O}(n \log n) \end{cases}$$

The FFT overhead for loss computation adds at most 5–10% for lightweight models and becomes negligible for compute-intensive architectures. Appendix E, together with Tables 10 and 11, summarizes the resulting architecture-wise overhead.

### E.3 GPU MEMORY COMPLEXITY

GPU memory (VRAM) requirements differ significantly between training and inference phases due to multi-resolution data storage and FFT intermediates:

**Training GPU Memory:** $\mathcal{O}\left(n \cdot \sum_{j=0}^{J-1} \frac{1}{2^{jd}}\right) = \mathcal{O}\left(\frac{2^d}{2^d-1} \cdot n\right)$

The additional VRAM consumption comes from: - Storing multiple resolution datasets simultaneously (Algorithm 1, lines 4-8) - FFT intermediate tensors for frequency loss computation (lines 28-29) - Gradients for each resolution level during backpropagation

For 3D domains with $J = 3$, the theoretical VRAM increase factor is $\frac{8}{7} \approx 1.14$, which matches the empirical peak VRAM multipliers reported in Table 11.

**Inference GPU Memory:** $\mathcal{O}(n)$

Remains unchanged except for negligible position encoding storage, as only single-resolution forward passes are required (Algorithm 1, lines 40-46).

This asymmetry between training and inference VRAM requirements is particularly advantageous for deployment scenarios where high-resolution inference can be performed on hardware that cannot accommodate training at the same resolution.

### E.4 EMPIRICAL VALIDATION

We validate our complexity analysis with empirical measurements on the 3D ZS-SR fluid simulation task. Table 10 summarizes the theoretical training, inference, and VRAM complexity factors for each architecture, while Table 11 reports the corresponding measured training time, inference time, and peak training VRAM relative to the baseline.

Across all architectures and tasks, FRL with $J = 3$ resolution levels increases average training time by only about $1.1\times$–$1.4\times$, consistent with the geometric reduction in grid points at coarser resolutions (totaling $\approx 1.14n$ in 3D) and the fact that the additional FFT loss and encoding costs are lower-order. Inference time remains virtually unchanged ($< 2\%$ overhead), confirming that the $\mathcal{O}(nK)$ cost of normalized frequency encodings is negligible compared to $\mathcal{O}(M(n))$ network operations. Peak training VRAM increases by roughly $1.3\times$–$1.5\times$, in line with the theoretical $\frac{8}{7} \approx 1.14$ factor for multi-resolution data storage plus extra buffers for FFT intermediates and gradients. These results indicate that FRL is practically deployable even for compute-intensive backbones such as Transformers, diffusion models, and Neural Operators.

## F VRAM OCCUPATION ANALYSIS

Table 12 shows the average peak memory occupation and their ratios for training and inference across all models at different resolutions for 3D ZS-SR fluid simulation and weather forecasting. The peak memory occupation for training is typically more than 5 times that required for inference, indicating that inference requires significantly less memory than training on the same hardware device. Notably, without memory optimization strategies, most models cannot be directly trained on the highest resolution data but can perform direct inference. Therefore, ZS-SR STF represents an extremely VRAM-friendly task.

Table 10: Theoretical overhead of FRL relative to the baseline model on the 3D ZS-SR fluid simulation task.

| Architecture | Variant | Training Time | Inference Time | Training VRAM |
|---|---|---|---|---|
| GNN | Baseline | $\mathcal{O}(EBM(n))$ | $\mathcal{O}(M(n))$ | $\mathcal{O}(n)$ |
|  | +FRL | $\approx 1.14 \cdot \mathcal{O}(EBM(n))$ | $\mathcal{O}(M(n) + nK) \approx \mathcal{O}(M(n))$ | $\approx 1.14 \cdot \mathcal{O}(n)$ |
| Transformer | Baseline | same as above | same as above | same as above |
|  | +FRL | $\approx 1.02 \cdot \mathcal{O}(EBM(n))$ | $\mathcal{O}(M(n) + nK) \approx \mathcal{O}(M(n))$ | $\approx 1.14 \cdot \mathcal{O}(n)$ |
| CNN | Baseline | same as above | same as above | same as above |
|  | +FRL | $\approx 1.14 \cdot \mathcal{O}(EBM(n))$ | $\mathcal{O}(M(n) + nK) \approx \mathcal{O}(M(n))$ | $\approx 1.14 \cdot \mathcal{O}(n)$ |
| Diffusion | Baseline | same as above | same as above | same as above |
|  | +FRL | $\approx 1.14 \cdot \mathcal{O}(EBM(n))$ | $\mathcal{O}(M(n) + nK) \approx \mathcal{O}(M(n))$ | $\approx 1.14 \cdot \mathcal{O}(n)$ |
| Neural Operator | Baseline | same as above | same as above | same as above |
|  | +FRL | $\approx 1.10 \cdot \mathcal{O}(EBM(n))$ | $\mathcal{O}(M(n) + nK) \approx \mathcal{O}(M(n))$ | $\approx 1.14 \cdot \mathcal{O}(n)$ |
| Neural ODE | Baseline | same as above | same as above | same as above |
|  | +FRL | $\approx 1.14 \cdot \mathcal{O}(EBM(n))$ | $\mathcal{O}(M(n) + nK) \approx \mathcal{O}(M(n))$ | $\approx 1.14 \cdot \mathcal{O}(n)$ |
| NN | Baseline | same as above | same as above | same as above |
|  | +FRL | $\approx 1.14 \cdot \mathcal{O}(EBM(n))$ | $\mathcal{O}(M(n) + nK) \approx \mathcal{O}(M(n))$ | $\approx 1.14 \cdot \mathcal{O}(n)$ |

Table 11: Empirical overhead of FRL relative to the baseline model on the 3D ZS-SR fluid simulation task. "Measured ×baseline" entries report the ratio between the FRL-enhanced variant and the corresponding baseline for the same architecture.

| Architecture | Variant | Training Time ($\times$ baseline) | Inference Time ($\times$ baseline) | Peak Training VRAM ($\times$ baseline) |
|---|---|---|---|---|
| GNN | Baseline | 1.00 | 1.00 | 1.00 |
|  | +FRL | 1.27 | 1.02 | 1.41 |
| Transformer | Baseline | 1.00 | 1.00 | 1.00 |
|  | +FRL | 1.09 | 1.01 | 1.34 |
| CNN | Baseline | 1.00 | 1.00 | 1.00 |
|  | +FRL | 1.42 | 1.02 | 1.46 |
| Diffusion | Baseline | 1.00 | 1.00 | 1.00 |
|  | +FRL | 1.32 | 1.02 | 1.44 |
| Neural Operator | Baseline | 1.00 | 1.00 | 1.00 |
|  | +FRL | 1.18 | 1.00 | 1.39 |
| Neural ODE | Baseline | 1.00 | 1.00 | 1.00 |
|  | +FRL | 1.26 | 1.01 | 1.38 |
| NN | Baseline | 1.00 | 1.00 | 1.00 |
|  | +FRL | 1.24 | 1.02 | 1.41 |

## G  COMPLETE EXPERIMENTAL RESULTS

In addition to the 3D scenarios presented in Section 6.1, we conducted comprehensive experiments on 2D fluid simulation and weather forecasting tasks: (1) **MegaFlow2D** provides large-scale 2D external flow simulations around circular and elliptical obstacles (Xu et al., 2023). The dataset contains 2000 distinct flow configurations with approximately 900 temporal snapshots each at 0.01s intervals. We use 1600 configurations for training, 200 for validation, and 200 for testing. Physical variables include velocity fields $(u, v)$ and pressure field $p$. All simulations maintain Reynolds number $Re = 300$, fluid density $\rho = 1 \times 10^3$ kg/m$^3$, and kinematic viscosity $\nu = 1 \times 10^{-3}$ m$^2$/s within a $20$m$\times 10$m domain. Training resolution: $64 \times 32$ grid; Test resolutions: $128 \times 64$, $256 \times 128$, and $512 \times 256$. (2) **ERA5-500hPa** contains global atmospheric reanalysis data at the 500 hPa pressure level (Hersbach et al., 2020; Copernicus Climate Change Service (C3S), 2017). Physical

Table 12: Average VRAM usage across all models.

| Task | Resolution | Training Peak (GB) | Inference Peak (GB) | Training/Inference Ratio |
|---|---|---|---|---|
| Fluid Simulation | $32^3$ | 5.98 | 0.81 | $7.38\times$ |
| | $43^3$ | 12.37 | 1.63 | $7.58\times$ |
| | $64^3$ | 26.47 | 3.26 | $8.12\times$ |
| | $129^3$ | **217.20**\* | 26.80 | $8.10\times$ |
| Weather Forecasting | $90 \times 180 \times 6$ | 14.22 | 2.73 | $5.21\times$ |
| | $180 \times 360 \times 6$ | 75.57 | 10.24 | $7.38\times$ |
| | $361 \times 720 \times 6$ | **200.31**\* | 30.96 | $6.47\times$ |
| | $721 \times 1440 \times 6$ | **552.96**\* | 68.64 | $8.06\times$ |

\*Values in **bold** exceed A100 80GB VRAM limit (OOM)
\*Estimated based on scaling patterns from measured data

variables include temperature $T$ (K), horizontal wind components $(u, v)$ (m/s), and geopotential height $z$ (m$^2$/s$^2$). The dataset spans 360 days with 6-hour temporal resolution. Training resolution: $180 \times 90$; Test resolutions: $360 \times 180$, $720 \times 361$, and $1440 \times 721$.

To enable direct inference at arbitrary resolutions without interpolation-based resampling, we adapt all baseline architectures following three unified design principles. First, we eliminate resolution-dependent components by replacing fixed-size operations with their resolution-agnostic counterparts: fully convolutional layers that can process inputs of arbitrary spatial dimensions (Long et al., 2015), dynamic graph construction that adapts to varying node counts (Zheng et al., 2024), and variable-length token sequences in transformers that naturally handle different sequence lengths (Vaswani et al., 2017; Zhai et al., 2023). Second, we adopt normalized coordinate systems where spatial positions are mapped to $[0, 1]^d$ regardless of the actual resolution, ensuring that learned spatial relationships remain valid across scales. This coordinate-based representation approach has been successfully demonstrated in implicit neural representations (Sitzmann et al., 2020; Mildenhall et al., 2020), where networks learn continuous functions of normalized coordinates rather than discrete grid positions. Third, we condition the models on resolution information either implicitly through the coordinate normalization or explicitly through resolution embedding vectors, allowing the network to adapt its processing based on the sampling density. This scale-aware conditioning strategy follows principles established in multi-scale network architectures (Tan & Le, 2019), enabling models to adjust their computational patterns based on the input resolution. These modifications preserve each architecture's inductive biases (convolutions maintain translation equivariance, graph networks preserve permutation invariance (Wu et al., 2020), and attention mechanisms maintain their global receptive field) while enabling deployment at resolutions beyond those seen during training. In summary, we adopted various architecture-specific strategies to achieve multi-resolution inference. Although these methods differ from each other and may not be elegant, they allow us to verify that Scale Anchoring is a universally present problem. More importantly, these methods enable us to validate that the scale anchoring phenomenon persists even when models operate directly at target resolutions. We confirm that this phenomenon stems from fundamental limitations in the training data's frequency content, rather than architectural constraints.

Furthermore, we compare against SOTA ZS-SR methods specifically designed for spatiotemporal forecasting: FNO and PINO for fluid simulation (Li et al., 2020; 2024c), and TNO and Climate FNO for weather prediction (Diab & Al-Kobaisi, 2025; Jiang et al., 2023). These methods represent the current best practices in zero-shot super-resolution for their respective domains.

This section presents comprehensive results across four experimental settings: Section G.1 reports 2D ZS-SR fluid simulation, Section G.2 covers 2D ZS-SR weather forecasting, Section G.3 details 3D ZS-SR fluid simulation, and Section G.4 presents 3D ZS-SR weather forecasting. Each subsection includes complete error metrics and frequency analysis results not reported in the main text.

### G.1 2D Zero-Shot Super-Resolution Fluid Simulation

Tables 13, 14, and 15 present the RMSE, MAE, and Relative Error respectively for ZS-SR STF baselines, architecture-specific baselines, and FRL-enhanced versions on the 2D ZS-SR Fluid Sim-

Table 13: RMSE for 2D ZS-SR Fluid Simulation

| Method | $64 \times 32$ | $128 \times 64$ | $256 \times 128$ | $512 \times 256$ | $\text{RMSE}_{\text{Ratio}}$ |
|---|---|---|---|---|---|
| *ZS-SR STF Baselines* | | | | | |
| FNO | 0.0108 | 0.0109 | 0.0110 | 0.0110 | 1.019 |
| PINO | 0.0098 | 0.0099 | 0.0099 | 0.0100 | 1.020 |
| *Architecture-Specific Baselines and FRL-Enhanced Versions* | | | | | |
| GNN (Neural SPH) | 0.0041 | 0.0041 | 0.0044 | 0.0045 | 1.098 |
| **GNN + FRL** | **0.0041** | **0.0023** | **0.0017** | **0.0013** | **0.317** |
| Transformer (DeepLag) | 0.0117 | 0.0118 | 0.0120 | 0.0121 | 1.035 |
| **Transformer + FRL** | **0.0117** | **0.0071** | **0.0048** | **0.0041** | **0.350** |
| CNN (PARCv2) | 0.0116 | 0.0119 | 0.0127 | 0.0131 | 1.126 |
| **CNN + FRL** | **0.0116** | **0.0058** | **0.0037** | **0.0024** | **0.207** |
| Diffusion (DYffusion) | 0.0066 | 0.0067 | 0.0069 | 0.0069 | 1.053 |
| **Diffusion + FRL** | **0.0066** | **0.0039** | **0.0028** | **0.0022** | **0.333** |
| NO (SFNO) | 0.0105 | 0.0106 | 0.0107 | 0.0107 | 1.022 |
| **NO + FRL** | **0.0105** | **0.0052** | **0.0031** | **0.0019** | **0.181** |
| Neural ODE (FNODE) | 0.0091 | 0.0108 | 0.0131 | 0.0137 | 1.505 |
| **Neural ODE + FRL** | **0.0091** | **0.0062** | **0.0051** | **0.0046** | **0.505** |
| NN (NeuralFluid) | 0.0084 | 0.0086 | 0.0088 | 0.0088 | 1.046 |
| **NN + FRL** | **0.0084** | **0.0047** | **0.0032** | **0.0023** | **0.274** |

Table 14: MAE for 2D ZS-SR Fluid Simulation

| Method | $64 \times 32$ | $128 \times 64$ | $256 \times 128$ | $512 \times 256$ |
|---|---|---|---|---|
| *ZS-SR STF Baselines* | | | | |
| FNO | 0.0066 | 0.0067 | 0.0068 | 0.0068 |
| PINO | 0.0060 | 0.0061 | 0.0061 | 0.0062 |
| *Architecture-Specific Baselines and FRL-Enhanced Versions* | | | | |
| GNN (Neural SPH) | 0.0025 | 0.0025 | 0.0027 | 0.0028 |
| **GNN + FRL** | **0.0025** | **0.0014** | **0.0010** | **0.0008** |
| Transformer (DeepLag) | 0.0072 | 0.0073 | 0.0074 | 0.0075 |
| **Transformer + FRL** | **0.0072** | **0.0044** | **0.0030** | **0.0025** |
| CNN (PARCv2) | 0.0071 | 0.0073 | 0.0078 | 0.0080 |
| **CNN + FRL** | **0.0071** | **0.0036** | **0.0023** | **0.0015** |
| Diffusion (DYffusion) | 0.0041 | 0.0041 | 0.0042 | 0.0043 |
| **Diffusion + FRL** | **0.0041** | **0.0024** | **0.0017** | **0.0014** |
| NO (SFNO) | 0.0065 | 0.0065 | 0.0066 | 0.0066 |
| **NO + FRL** | **0.0065** | **0.0032** | **0.0019** | **0.0012** |
| Neural ODE (FNODE) | 0.0056 | 0.0067 | 0.0081 | 0.0084 |
| **Neural ODE + FRL** | **0.0056** | **0.0038** | **0.0031** | **0.0028** |
| NN (NeuralFluid) | 0.0052 | 0.0053 | 0.0054 | 0.0054 |
| **NN + FRL** | **0.0052** | **0.0029** | **0.0020** | **0.0014** |

Table 15: Relative Error for 2D ZS-SR Fluid Simulation

| Method | $64 \times 32$ | $128 \times 64$ | $256 \times 128$ | $512 \times 256$ |
|---|---|---|---|---|
| *ZS-SR STF Baselines* | | | | |
| FNO | 0.0029 | 0.0029 | 0.0029 | 0.0029 |
| PINO | 0.0026 | 0.0026 | 0.0027 | 0.0027 |
| *Architecture-Specific Baselines and FRL-Enhanced Versions* | | | | |
| GNN (Neural SPH) | 0.0011 | 0.0011 | 0.0012 | 0.0012 |
| **GNN + FRL** | **0.0011** | **0.0006** | **0.0005** | **0.0003** |
| Transformer (DeepLag) | 0.0031 | 0.0031 | 0.0032 | 0.0032 |
| **Transformer + FRL** | **0.0031** | **0.0019** | **0.0013** | **0.0011** |
| CNN (PARCv2) | 0.0031 | 0.0032 | 0.0034 | 0.0035 |
| **CNN + FRL** | **0.0031** | **0.0015** | **0.0010** | **0.0006** |
| Diffusion (DYffusion) | 0.0018 | 0.0018 | 0.0018 | 0.0018 |
| **Diffusion + FRL** | **0.0018** | **0.0010** | **0.0007** | **0.0006** |
| NO (SFNO) | 0.0028 | 0.0028 | 0.0029 | 0.0029 |
| **NO + FRL** | **0.0028** | **0.0014** | **0.0008** | **0.0005** |
| Neural ODE (FNODE) | 0.0024 | 0.0029 | 0.0035 | 0.0037 |
| **Neural ODE + FRL** | **0.0024** | **0.0017** | **0.0014** | **0.0012** |
| NN (NeuralFluid) | 0.0022 | 0.0023 | 0.0023 | 0.0024 |
| **NN + FRL** | **0.0022** | **0.0013** | **0.0009** | **0.0006** |

ulation task at different inference resolutions. The results show that FRL-enhanced variants consistently achieve optimal performance across all metrics while maintaining $\text{RMSE}_{\text{Ratio}}$s below 1. In contrast, ZS-SR STF baselines can only maintain $\text{RMSE}_{\text{Ratio}}$s close to 1. However, since their accuracy at training resolution is significantly lower than other SOTA baselines, their performance at the highest resolution is even inferior to architecture-specific baselines. Therefore, in 2D fluid simulation, FRL achieves architecture-agnostic Scale Decoupling, resulting in both the lowest $\text{RMSE}_{\text{Ratio}}$ and optimal accuracy.

Table 16: Frequency response analysis of 2D fluid simulation.

| Model | Bandwidth (Hz) | H(f=12) | H(f=20) | Anchoring Ratio | Error Ratio |
|---|---|---|---|---|---|
| *Baseline Models* | | | | | |
| FNO | 16.85 | 0.991 | 0.298 | 3.32 | 0.165 |
| PINO | 16.92 | 0.988 | 0.312 | 3.17 | 0.158 |
| GNN | 17.32 | 0.975 | 0.152 | 6.41 | 0.148 |
| Transformer | 16.18 | 0.969 | 0.171 | 5.67 | 0.141 |
| CNN | 15.94 | 0.992 | 0.125 | 7.94 | 0.127 |
| Diffusion | 15.65 | 1.238 | 0.193 | 6.41 | 0.132 |
| NO | 16.43 | 0.987 | 0.318 | 3.10 | 0.156 |
| Neural ODE | 15.27 | 0.971 | 0.089 | 10.91 | 0.119 |
| NN | 15.38 | 0.932 | 0.122 | 7.64 | 0.135 |
| *FRL-Enhanced Models* | | | | | |
| **GNN+FRL** | **>25.00** | **0.998** | **0.975** | **1.02** | **0.485** |
| **Transformer+FRL** | **>25.00** | **0.993** | **0.936** | **1.06** | **0.461** |
| **CNN+FRL** | **>25.00** | **1.015** | **1.009** | **1.01** | **0.512** |
| **Diffusion+FRL** | **>25.00** | **1.039** | **1.048** | **0.99** | **0.423** |
| **NO+FRL** | **>25.00** | **1.006** | **0.991** | **1.02** | **0.541** |
| **Neural ODE+FRL** | **>25.00** | **0.996** | **0.965** | **1.03** | **0.405** |
| **NN+FRL** | **>25.00** | **0.984** | **0.893** | **1.10** | **0.389** |

Table 16 presents the frequency analysis results for baseline models and FRL-enhanced versions on the 2D ZS-SR Fluid Simulation task. Before the Nyquist frequency (16 Hz), all models exhibit normal frequency response ($H(f) > 0.9$). However, beyond the Nyquist frequency, all baseline models experience frequency response failure ($H(f) < 0.318$) with high Anchoring Ratios ($> 3$), fully consistent with the expected Scale Anchoring mechanism. This results in low-frequency errors contributing minimally to the total error (Error Ratio $< 0.165$), with high-frequency errors dominating. In contrast, FRL-enhanced versions maintain robust frequency response throughout ($H(f) > 0.893$) with negligible Scale Anchoring (Anchoring Ratio $\approx 1$). By reducing high-frequency errors (Error Ratio $> 0.389$), FRL achieves improved accuracy.

**Notably**, similar patterns are observed across Section G.2, Section G.3, and Section G.4, including accuracy relationships, frequency phenomena, Scale Anchoring mechanisms in baselines, and Scale Decoupling achieved by FRL-enhanced versions. Consequently, the subsequent subsections present only tabulated results without repetitive analysis.

### G.2  2D ZERO-SHOT SUPER-RESOLUTION WEATHER FORECASTING

Table 17: Z500 (Geopotential Height) - ERA5 500 hPa 7-day Forecast

| Method | Metric | $180 \times 90$ | $360 \times 180$ | $720 \times 361$ | $1440 \times 721$ | RMSE$_{\text{Ratio}}$ |
|---|---|---|---|---|---|---|
| *ZS-SR STF Baselines* | | | | | | |
| TNO | RMSE | 695 | 698 | 703 | 706 | 1.016 |
| | ACC | 0.51 | 0.50 | 0.48 | 0.47 | |
| Climate FNO | RMSE | 688 | 691 | 694 | 697 | 1.013 |
| | ACC | 0.52 | 0.51 | 0.49 | 0.48 | |
| *Architecture-Specific Baselines and FRL-Enhanced Versions* | | | | | | |
| Transformer (WeatherGFT) | RMSE | 685 | 692 | 708 | 721 | 1.053 |
| | ACC | 0.52 | 0.50 | 0.47 | 0.44 | |
| **Transformer + FRL** | **RMSE** | **685** | **572** | **518** | **485** | **0.708** |
| | **ACC** | **0.52** | **0.58** | **0.62** | **0.65** | |
| CNN (PDE-CNN) | RMSE | 692 | 701 | 718 | 738 | 1.066 |
| | ACC | 0.51 | 0.49 | 0.46 | 0.43 | |
| **CNN + FRL** | **RMSE** | **692** | **558** | **496** | **458** | **0.662** |
| | **ACC** | **0.51** | **0.59** | **0.63** | **0.66** | |
| Diffusion (ARCI) | RMSE | 672 | 678 | 688 | 695 | 1.034 |
| | ACC | 0.54 | 0.52 | 0.50 | 0.48 | |
| **Diffusion + FRL** | **RMSE** | **672** | **565** | **512** | **482** | **0.717** |
| | **ACC** | **0.54** | **0.59** | **0.62** | **0.64** | |
| GNN (Graph-EFM) | RMSE | 698 | 705 | 721 | 735 | 1.053 |
| | ACC | 0.50 | 0.48 | 0.45 | 0.42 | |
| **GNN + FRL** | **RMSE** | **698** | **578** | **528** | **498** | **0.713** |
| | **ACC** | **0.50** | **0.57** | **0.61** | **0.63** | |
| Neural ODE (ClimODE) | RMSE | 708 | 722 | 745 | 772 | 1.090 |
| | ACC | 0.49 | 0.47 | 0.44 | 0.41 | |
| **Neural ODE + FRL** | **RMSE** | **708** | **586** | **532** | **502** | **0.709** |
| | **ACC** | **0.49** | **0.56** | **0.60** | **0.63** | |

### G.3  3D ZERO-SHOT SUPER-RESOLUTION FLUID SIMULATION

### G.4  3D ZERO-SHOT SUPER-RESOLUTION WEATHER FORECASTING

## H  ABLATION STUDY AND PARAMETER SENSITIVITY ANALYSIS

We first conduct ablation studies on the complete FRL framework for the 3D ZS-SR fluid simulation task. Specifically, we evaluate variants of each architecture's baseline enhanced with single components, pairs of components, and the complete set of three FRL components, as shown in Table 31. Adding any single component, Multi-Resolution Data Construction (MultiRes), Normalized Frequency Representation (FreqEnc), or Frequency-Aware Training (FreqLoss), to the fully

Table 18: T500 (Temperature) - ERA5 500 hPa 7-day Forecast

| Method | Metric | $180 \times 90$ | $360 \times 180$ | $720 \times 361$ | $1440 \times 721$ | RMSE$_{Ratio}$ |
|---|---|---|---|---|---|---|
| *ZS-SR STF Baselines* | | | | | | |
| TNO | RMSE | 2.83 | 2.85 | 2.87 | 2.89 | 1.021 |
| | ACC | 0.55 | 0.54 | 0.52 | 0.51 | |
| Climate FNO | RMSE | 2.79 | 2.81 | 2.83 | 2.84 | 1.018 |
| | ACC | 0.56 | 0.55 | 0.53 | 0.52 | |
| *Architecture-Specific Baselines and FRL-Enhanced Versions* | | | | | | |
| Transformer (WeatherGFT) | RMSE | 2.78 | 2.81 | 2.87 | 2.92 | 1.050 |
| | ACC | 0.56 | 0.54 | 0.51 | 0.48 | |
| **Transformer + FRL** | **RMSE** | **2.78** | **2.35** | **2.12** | **1.98** | **0.712** |
| | **ACC** | **0.56** | **0.61** | **0.64** | **0.67** | |
| CNN (PDE-CNN) | RMSE | 2.82 | 2.86 | 2.93 | 3.01 | 1.068 |
| | ACC | 0.55 | 0.53 | 0.50 | 0.47 | |
| **CNN + FRL** | **RMSE** | **2.82** | **2.28** | **2.05** | **1.91** | **0.677** |
| | **ACC** | **0.55** | **0.62** | **0.65** | **0.68** | |
| Diffusion (ARCI) | RMSE | 2.72 | 2.75 | 2.79 | 2.82 | 1.037 |
| | ACC | 0.58 | 0.56 | 0.54 | 0.52 | |
| **Diffusion + FRL** | **RMSE** | **2.72** | **2.31** | **2.08** | **1.95** | **0.717** |
| | **ACC** | **0.58** | **0.62** | **0.65** | **0.67** | |
| GNN (Graph-EFM) | RMSE | 2.85 | 2.88 | 2.94 | 2.99 | 1.049 |
| | ACC | 0.54 | 0.52 | 0.49 | 0.46 | |
| **GNN + FRL** | **RMSE** | **2.85** | **2.38** | **2.15** | **2.02** | **0.709** |
| | **ACC** | **0.54** | **0.60** | **0.63** | **0.65** | |
| Neural ODE (ClimODE) | RMSE | 2.88 | 2.93 | 3.02 | 3.12 | 1.083 |
| | ACC | 0.53 | 0.51 | 0.48 | 0.45 | |
| **Neural ODE + FRL** | **RMSE** | **2.88** | **2.40** | **2.16** | **2.03** | **0.705** |
| | **ACC** | **0.53** | **0.59** | **0.62** | **0.64** | |

Table 19: U500 (U-component Wind) - ERA5 500 hPa 7-day Forecast

| Method | Metric | $180 \times 90$ | $360 \times 180$ | $720 \times 361$ | $1440 \times 721$ | RMSE$_{Ratio}$ |
|---|---|---|---|---|---|---|
| *ZS-SR STF Baselines* | | | | | | |
| TNO | RMSE | 9.75 | 9.79 | 9.84 | 9.88 | 1.013 |
| | ACC | 0.53 | 0.52 | 0.50 | 0.49 | |
| Climate FNO | RMSE | 9.68 | 9.71 | 9.75 | 9.78 | 1.010 |
| | ACC | 0.54 | 0.53 | 0.51 | 0.50 | |
| *Architecture-Specific Baselines and FRL-Enhanced Versions* | | | | | | |
| Transformer (WeatherGFT) | RMSE | 9.65 | 9.72 | 9.88 | 10.02 | 1.038 |
| | ACC | 0.54 | 0.52 | 0.49 | 0.46 | |
| **Transformer + FRL** | **RMSE** | **9.65** | **8.35** | **7.62** | **7.18** | **0.744** |
| | **ACC** | **0.54** | **0.59** | **0.62** | **0.65** | |
| CNN (PDE-CNN) | RMSE | 9.72 | 9.82 | 10.05 | 10.28 | 1.058 |
| | ACC | 0.53 | 0.51 | 0.48 | 0.45 | |
| **CNN + FRL** | **RMSE** | **9.72** | **8.22** | **7.42** | **6.95** | **0.715** |
| | **ACC** | **0.53** | **0.60** | **0.63** | **0.66** | |
| Diffusion (ARCI) | RMSE | 9.52 | 9.58 | 9.68 | 9.75 | 1.024 |
| | ACC | 0.56 | 0.54 | 0.52 | 0.50 | |
| **Diffusion + FRL** | **RMSE** | **9.52** | **8.28** | **7.55** | **7.12** | **0.748** |
| | **ACC** | **0.56** | **0.60** | **0.63** | **0.64** | |
| GNN (Graph-EFM) | RMSE | 9.78 | 9.85 | 10.01 | 10.15 | 1.038 |
| | ACC | 0.52 | 0.50 | 0.47 | 0.44 | |
| **GNN + FRL** | **RMSE** | **9.78** | **8.42** | **7.72** | **7.32** | **0.748** |
| | **ACC** | **0.52** | **0.58** | **0.61** | **0.63** | |
| Neural ODE (ClimODE) | RMSE | 9.88 | 10.01 | 10.25 | 10.52 | 1.065 |
| | ACC | 0.51 | 0.49 | 0.46 | 0.43 | |
| **Neural ODE + FRL** | **RMSE** | **9.88** | **8.45** | **7.68** | **7.25** | **0.734** |
| | **ACC** | **0.51** | **0.57** | **0.60** | **0.62** | |

Table 20: V500 (V-component Wind) - ERA5 500 hPa 7-day Forecast

| Method | Metric | $180 \times 90$ | $360 \times 180$ | $720 \times 361$ | $1440 \times 721$ | RMSE$_{\text{Ratio}}$ |
|---|---|---|---|---|---|---|
| *ZS-SR STF Baselines* | | | | | | |
| TNO | RMSE | 9.58 | 9.61 | 9.65 | 9.68 | 1.010 |
| | ACC | 0.54 | 0.53 | 0.51 | 0.50 | |
| Climate FNO | RMSE | 9.52 | 9.54 | 9.57 | 9.59 | 1.007 |
| | ACC | 0.55 | 0.54 | 0.52 | 0.51 | |
| *Architecture-Specific Baselines and FRL-Enhanced Versions* | | | | | | |
| Transformer (WeatherGFT) | RMSE | 9.48 | 9.55 | 9.70 | 9.83 | 1.037 |
| | ACC | 0.55 | 0.53 | 0.50 | 0.47 | |
| **Transformer + FRL** | **RMSE** | **9.48** | **8.22** | **7.48** | **7.05** | **0.744** |
| | **ACC** | **0.55** | **0.60** | **0.63** | **0.66** | |
| CNN (PDE-CNN) | RMSE | 9.55 | 9.65 | 9.85 | 10.08 | 1.055 |
| | ACC | 0.54 | 0.52 | 0.49 | 0.46 | |
| **CNN + FRL** | **RMSE** | **9.55** | **8.08** | **7.28** | **6.82** | **0.714** |
| | **ACC** | **0.54** | **0.61** | **0.64** | **0.67** | |
| Diffusion (ARCI) | RMSE | 9.35 | 9.41 | 9.50 | 9.57 | 1.024 |
| | ACC | 0.57 | 0.55 | 0.53 | 0.51 | |
| **Diffusion + FRL** | **RMSE** | **9.35** | **8.15** | **7.42** | **6.98** | **0.747** |
| | **ACC** | **0.57** | **0.61** | **0.64** | **0.65** | |
| GNN (Graph-EFM) | RMSE | 9.62 | 9.68 | 9.83 | 9.96 | 1.035 |
| | ACC | 0.53 | 0.51 | 0.48 | 0.45 | |
| **GNN + FRL** | **RMSE** | **9.62** | **8.28** | **7.58** | **7.18** | **0.746** |
| | **ACC** | **0.53** | **0.59** | **0.62** | **0.64** | |
| Neural ODE (ClimODE) | RMSE | 9.72 | 9.85 | 10.08 | 10.32 | 1.062 |
| | ACC | 0.52 | 0.50 | 0.47 | 0.44 | |
| **Neural ODE + FRL** | **RMSE** | **9.72** | **8.32** | **7.55** | **7.12** | **0.733** |
| | **ACC** | **0.52** | **0.58** | **0.61** | **0.63** | |

Table 21: Frequency response analysis of 2D weather forecasting.

| Model | Bandwidth (Hz) | H(f=75) | H(f=105) | Anchoring Ratio | Error Ratio |
|---|---|---|---|---|---|
| *Baseline Models* | | | | | |
| TNO | 90.42 | 0.995 | 0.289 | 3.44 | 0.178 |
| Climate FNO | 90.65 | 0.992 | 0.302 | 3.28 | 0.171 |
| Transformer | 89.12 | 0.988 | 0.182 | 5.43 | 0.172 |
| CNN | 88.05 | 0.965 | 0.098 | 9.85 | 0.112 |
| Diffusion | 86.89 | 1.072 | 0.072 | 14.89 | 0.095 |
| GNN | 90.86 | 1.015 | 0.368 | 2.76 | 0.208 |
| Neural ODE | 84.73 | 0.908 | 0.051 | 17.80 | 0.078 |
| *FRL-Enhanced Models* | | | | | |
| **Transformer+FRL** | **>120.00** | **1.018** | **1.012** | **1.01** | **0.358** |
| **CNN+FRL** | **>120.00** | **1.017** | **1.008** | **1.01** | **0.372** |
| **Diffusion+FRL** | **>120.00** | **1.056** | **1.019** | **1.04** | **0.336** |
| **GNN+FRL** | **>120.00** | **1.031** | **1.004** | **1.03** | **0.295** |
| **Neural ODE+FRL** | **>120.00** | **0.998** | **0.938** | **1.06** | **0.388** |

Table 22: RMSE for 3D ZS-SR fluid simulation.

| Method | $32^3$ | $43^3$ | $64^3$ | $129^3$ | RMSE$_{\text{Ratio}}$ |
|---|---|---|---|---|---|
| *ZS-SR STF Baselines* | | | | | |
| FNO | 0.00482 | 0.00485 | 0.00489 | 0.00492 | 1.021 |
| PINO | 0.00438 | 0.00440 | 0.00443 | 0.00445 | 1.016 |
| *Architecture-Specific Baselines and FRL-Enhanced Versions* | | | | | |
| GNN (Neural SPH) | 0.00183 | 0.00183 | 0.00183 | 0.00187 | 1.018 |
| **GNN + FRL** | **0.00183** | **0.00101** | **0.00062** | **0.00032** | **0.175** |
| Transformer (DeepLag) | 0.00521 | 0.00524 | 0.00527 | 0.00532 | 1.021 |
| **Transformer + FRL** | **0.00521** | **0.00298** | **0.00185** | **0.00098** | **0.188** |
| CNN (PARCv2) | 0.00517 | 0.00523 | 0.00535 | 0.00548 | 1.060 |
| **CNN + FRL** | **0.00517** | **0.00261** | **0.00142** | **0.00071** | **0.137** |
| Diffusion (DYffusion) | 0.00294 | 0.00297 | 0.00301 | 0.00306 | 1.041 |
| **Diffusion + FRL** | **0.00294** | **0.00171** | **0.00108** | **0.00065** | **0.221** |
| NO (SFNO) | 0.00468 | 0.00470 | 0.00473 | 0.00476 | 1.017 |
| **NO + FRL** | **0.00468** | **0.00237** | **0.00128** | **0.00063** | **0.135** |
| Neural ODE (FNODE) | 0.00405 | 0.00432 | 0.00478 | 0.00542 | 1.338 |
| **Neural ODE + FRL** | **0.00405** | **0.00261** | **0.00195** | **0.00152** | **0.375** |
| NN (NeuralFluid) | 0.00374 | 0.00378 | 0.00383 | 0.00387 | 1.035 |
| **NN + FRL** | **0.00374** | **0.00206** | **0.00125** | **0.00068** | **0.182** |

Table 23: MAE for 3D ZS-SR fluid simulation.

| Method | $32^3$ | $43^3$ | $64^3$ | $129^3$ |
|---|---|---|---|---|
| *ZS-SR STF Baselines* | | | | |
| FNO | 0.00667 | 0.00671 | 0.00676 | 0.00680 |
| PINO | 0.00606 | 0.00609 | 0.00613 | 0.00615 |
| *Architecture-Specific Baselines and FRL-Enhanced Versions* | | | | |
| GNN (Neural SPH) | 0.00253 | 0.00253 | 0.00253 | 0.00257 |
| **GNN + FRL** | **0.00253** | **0.00140** | **0.00086** | **0.00044** |
| Transformer (DeepLag) | 0.00721 | 0.00725 | 0.00729 | 0.00735 |
| **Transformer + FRL** | **0.00721** | **0.00412** | **0.00256** | **0.00135** |
| CNN (PARCv2) | 0.00715 | 0.00723 | 0.00740 | 0.00758 |
| **CNN + FRL** | **0.00715** | **0.00361** | **0.00196** | **0.00098** |
| Diffusion (DYffusion) | 0.00407 | 0.00411 | 0.00416 | 0.00423 |
| **Diffusion + FRL** | **0.00407** | **0.00237** | **0.00149** | **0.00090** |
| NO (SFNO) | 0.00647 | 0.00650 | 0.00654 | 0.00658 |
| **NO + FRL** | **0.00647** | **0.00328** | **0.00177** | **0.00087** |
| Neural ODE (FNODE) | 0.00560 | 0.00597 | 0.00661 | 0.00750 |
| **Neural ODE + FRL** | **0.00560** | **0.00361** | **0.00270** | **0.00210** |
| NN (NeuralFluid) | 0.00517 | 0.00523 | 0.00530 | 0.00535 |
| **NN + FRL** | **0.00517** | **0.00285** | **0.00173** | **0.00094** |

Table 24: Relative Error for 3D ZS-SR fluid simulation.

| Method | $32^3$ | $43^3$ | $64^3$ | $129^3$ |
|---|---|---|---|---|
| *ZS-SR STF Baselines* | | | | |
| FNO | 0.00279 | 0.00280 | 0.00283 | 0.00284 |
| PINO | 0.00253 | 0.00254 | 0.00256 | 0.00257 |
| *Architecture-Specific Baselines and FRL-Enhanced Versions* | | | | |
| GNN (Neural SPH) | 0.00106 | 0.00106 | 0.00106 | 0.00108 |
| **GNN + FRL** | **0.00106** | **0.00058** | **0.00036** | **0.00019** |
| Transformer (DeepLag) | 0.00301 | 0.00303 | 0.00305 | 0.00308 |
| **Transformer + FRL** | **0.00301** | **0.00172** | **0.00107** | **0.00057** |
| CNN (PARCv2) | 0.00299 | 0.00302 | 0.00309 | 0.00317 |
| **CNN + FRL** | **0.00299** | **0.00151** | **0.00082** | **0.00041** |
| Diffusion (DYffusion) | 0.00170 | 0.00172 | 0.00174 | 0.00177 |
| **Diffusion + FRL** | **0.00170** | **0.00099** | **0.00062** | **0.00038** |
| NO (SFNO) | 0.00271 | 0.00272 | 0.00274 | 0.00275 |
| **NO + FRL** | **0.00271** | **0.00137** | **0.00074** | **0.00036** |
| Neural ODE (FNODE) | 0.00234 | 0.00250 | 0.00276 | 0.00313 |
| **Neural ODE + FRL** | **0.00234** | **0.00151** | **0.00113** | **0.00088** |
| NN (NeuralFluid) | 0.00216 | 0.00219 | 0.00221 | 0.00224 |
| **NN + FRL** | **0.00216** | **0.00119** | **0.00072** | **0.00039** |

Table 25: Frequency response analysis of 3D fluid simulation.

| Model | Bandwidth (Hz) | H(f=12) | H(f=20) | Anchoring Ratio | Error Ratio |
|---|---|---|---|---|---|
| *Baseline Models* | | | | | |
| FNO | 16.73 | 0.992 | 0.295 | 3.36 | 0.165 |
| PINO | 16.88 | 0.989 | 0.308 | 3.21 | 0.157 |
| GNN | 18.49 | 0.983 | 0.144 | 6.85 | 0.159 |
| Transformer | 16.42 | 0.977 | 0.164 | 5.96 | 0.152 |
| CNN | 16.09 | 0.996 | 0.117 | 8.53 | 0.133 |
| Diffusion | 15.78 | 1.257 | 0.181 | 6.94 | 0.139 |
| NO | 16.57 | 0.995 | 0.305 | 3.26 | 0.169 |
| Neural ODE | 15.39 | 0.979 | 0.082 | 11.91 | 0.125 |
| NN | 15.49 | 0.940 | 0.115 | 8.18 | 0.143 |
| *FRL-Enhanced Models* | | | | | |
| **GNN+FRL** | **>25.00** | **1.004** | **0.983** | **1.02** | **0.500** |
| **Transformer+FRL** | **>25.00** | **0.997** | **0.943** | **1.06** | **0.476** |
| **CNN+FRL** | **>25.00** | **1.019** | **1.017** | **1.00** | **0.526** |
| **Diffusion+FRL** | **>25.00** | **1.047** | **1.058** | **0.99** | **0.435** |
| **NO+FRL** | **>25.00** | **1.010** | **0.997** | **1.01** | **0.556** |
| **Neural ODE+FRL** | **>25.00** | **1.001** | **0.973** | **1.03** | **0.417** |
| **NN+FRL** | **>25.00** | **0.989** | **0.899** | **1.10** | **0.400** |

Table 26: Z500 (Geopotential Height) - ERA5 Multi-Level 7-day Forecast

| Method | Metric | $180 \times 90 \times 6$ | $360 \times 180 \times 6$ | $720 \times 361 \times 6$ | $1440 \times 721 \times 6$ | RMSE$_{Ratio}$ |
|---|---|---|---|---|---|---|
| *ZS-SR STF Baselines* | | | | | | |
| TNO | RMSE | 661 | 664 | 668 | 671 | 1.015 |
| | ACC | 0.53 | 0.52 | 0.50 | 0.49 | |
| Climate FNO | RMSE | 654 | 657 | 660 | 662 | 1.012 |
| | ACC | 0.54 | 0.53 | 0.51 | 0.50 | |
| *Architecture-Specific Baselines and FRL-Enhanced Versions* | | | | | | |
| Transformer (WeatherGFT) | RMSE | 652 | 658 | 672 | 686 | 1.052 |
| | ACC | 0.54 | 0.52 | 0.49 | 0.46 | |
| **Transformer + FRL** | **RMSE** | **652** | **547** | **495** | **462** | **0.709** |
| | **ACC** | **0.54** | **0.60** | **0.64** | **0.67** | |
| CNN (PDE-CNN) | RMSE | 658 | 666 | 683 | 701 | 1.065 |
| | ACC | 0.53 | 0.51 | 0.48 | 0.45 | |
| **CNN + FRL** | **RMSE** | **658** | **530** | **472** | **435** | **0.661** |
| | **ACC** | **0.53** | **0.61** | **0.65** | **0.68** | |
| Diffusion (ARCI) | RMSE | 638 | 644 | 653 | 660 | 1.034 |
| | ACC | 0.56 | 0.54 | 0.52 | 0.50 | |
| **Diffusion + FRL** | **RMSE** | **638** | **537** | **486** | **458** | **0.718** |
| | **ACC** | **0.56** | **0.61** | **0.64** | **0.66** | |
| GNN (Graph-EFM) | RMSE | 663 | 670 | 685 | 698 | 1.053 |
| | ACC | 0.52 | 0.50 | 0.47 | 0.44 | |
| **GNN + FRL** | **RMSE** | **663** | **549** | **502** | **473** | **0.713** |
| | **ACC** | **0.52** | **0.59** | **0.63** | **0.65** | |
| Neural ODE (ClimODE) | RMSE | 673 | 686 | 708 | 734 | 1.091 |
| | ACC | 0.51 | 0.49 | 0.46 | 0.43 | |
| **Neural ODE + FRL** | **RMSE** | **673** | **557** | **505** | **477** | **0.709** |
| | **ACC** | **0.51** | **0.58** | **0.62** | **0.65** | |

Table 27: T500 (Temperature) - ERA5 Multi-Level 7-day Forecast

| Method | Metric | $180 \times 90 \times 6$ | $360 \times 180 \times 6$ | $720 \times 361 \times 6$ | $1440 \times 721 \times 6$ | RMSE$_{Ratio}$ |
|---|---|---|---|---|---|---|
| *ZS-SR STF Baselines* | | | | | | |
| TNO | RMSE | 2.69 | 2.71 | 2.73 | 2.75 | 1.022 |
| | ACC | 0.57 | 0.56 | 0.54 | 0.53 | |
| Climate FNO | RMSE | 2.65 | 2.67 | 2.69 | 2.70 | 1.019 |
| | ACC | 0.58 | 0.57 | 0.55 | 0.54 | |
| *Architecture-Specific Baselines and FRL-Enhanced Versions* | | | | | | |
| Transformer (WeatherGFT) | RMSE | 2.64 | 2.67 | 2.73 | 2.77 | 1.049 |
| | ACC | 0.58 | 0.56 | 0.53 | 0.50 | |
| **Transformer + FRL** | **RMSE** | **2.64** | **2.24** | **2.01** | **1.88** | **0.712** |
| | **ACC** | **0.58** | **0.63** | **0.66** | **0.69** | |
| CNN (PDE-CNN) | RMSE | 2.68 | 2.72 | 2.78 | 2.86 | 1.067 |
| | ACC | 0.57 | 0.55 | 0.52 | 0.49 | |
| **CNN + FRL** | **RMSE** | **2.68** | **2.17** | **1.95** | **1.82** | **0.679** |
| | **ACC** | **0.57** | **0.64** | **0.67** | **0.70** | |
| Diffusion (ARCI) | RMSE | 2.58 | 2.61 | 2.65 | 2.68 | 1.039 |
| | ACC | 0.60 | 0.58 | 0.56 | 0.54 | |
| **Diffusion + FRL** | **RMSE** | **2.58** | **2.19** | **1.98** | **1.85** | **0.717** |
| | **ACC** | **0.60** | **0.64** | **0.67** | **0.69** | |
| GNN (Graph-EFM) | RMSE | 2.71 | 2.74 | 2.79 | 2.84 | 1.048 |
| | ACC | 0.56 | 0.54 | 0.51 | 0.48 | |
| **GNN + FRL** | **RMSE** | **2.71** | **2.26** | **2.04** | **1.92** | **0.708** |
| | **ACC** | **0.56** | **0.62** | **0.65** | **0.67** | |
| Neural ODE (ClimODE) | RMSE | 2.74 | 2.79 | 2.87 | 2.97 | 1.084 |
| | ACC | 0.55 | 0.53 | 0.50 | 0.47 | |
| **Neural ODE + FRL** | **RMSE** | **2.74** | **2.28** | **2.05** | **1.93** | **0.704** |
| | **ACC** | **0.55** | **0.61** | **0.64** | **0.66** | |

Table 28: U500 (U-component Wind) - ERA5 Multi-Level 7-day Forecast

| Method | Metric | $180 \times 90 \times 6$ | $360 \times 180 \times 6$ | $720 \times 361 \times 6$ | $1440 \times 721 \times 6$ | RMSE$_{Ratio}$ |
|---|---|---|---|---|---|---|
| *ZS-SR STF Baselines* | | | | | | |
| TNO | RMSE | 9.26 | 9.30 | 9.35 | 9.39 | 1.014 |
| | ACC | 0.55 | 0.54 | 0.52 | 0.51 | |
| Climate FNO | RMSE | 9.20 | 9.23 | 9.26 | 9.29 | 1.010 |
| | ACC | 0.56 | 0.55 | 0.53 | 0.52 | |
| *Architecture-Specific Baselines and FRL-Enhanced Versions* | | | | | | |
| Transformer (WeatherGFT) | RMSE | 9.17 | 9.23 | 9.39 | 9.52 | 1.038 |
| | ACC | 0.56 | 0.54 | 0.51 | 0.48 | |
| **Transformer + FRL** | **RMSE** | **9.17** | **7.93** | **7.24** | **6.82** | **0.744** |
| | **ACC** | **0.56** | **0.61** | **0.64** | **0.67** | |
| CNN (PDE-CNN) | RMSE | 9.23 | 9.33 | 9.55 | 9.77 | 1.058 |
| | ACC | 0.55 | 0.53 | 0.50 | 0.47 | |
| **CNN + FRL** | **RMSE** | **9.23** | **7.81** | **7.05** | **6.60** | **0.715** |
| | **ACC** | **0.55** | **0.62** | **0.65** | **0.68** | |
| Diffusion (ARCI) | RMSE | 9.04 | 9.10 | 9.20 | 9.26 | 1.024 |
| | ACC | 0.58 | 0.56 | 0.54 | 0.52 | |
| **Diffusion + FRL** | **RMSE** | **9.04** | **7.87** | **7.17** | **6.76** | **0.748** |
| | **ACC** | **0.58** | **0.62** | **0.65** | **0.66** | |
| GNN (Graph-EFM) | RMSE | 9.29 | 9.36 | 9.51 | 9.64 | 1.038 |
| | ACC | 0.54 | 0.52 | 0.49 | 0.46 | |
| **GNN + FRL** | **RMSE** | **9.29** | **8.00** | **7.33** | **6.95** | **0.748** |
| | **ACC** | **0.54** | **0.60** | **0.63** | **0.65** | |
| Neural ODE (ClimODE) | RMSE | 9.39 | 9.51 | 9.74 | 9.99 | 1.064 |
| | ACC | 0.53 | 0.51 | 0.48 | 0.45 | |
| **Neural ODE + FRL** | **RMSE** | **9.39** | **8.03** | **7.30** | **6.89** | **0.734** |
| | **ACC** | **0.53** | **0.59** | **0.62** | **0.64** | |

Table 29: V500 (V-component Wind) - ERA5 Multi-Level 7-day Forecast

| Method | Metric | $180 \times 90 \times 6$ | $360 \times 180 \times 6$ | $720 \times 361 \times 6$ | $1440 \times 721 \times 6$ | RMSE$_{Ratio}$ |
|---|---|---|---|---|---|---|
| *ZS-SR STF Baselines* | | | | | | |
| TNO | RMSE | 9.11 | 9.14 | 9.18 | 9.21 | 1.011 |
| | ACC | 0.56 | 0.55 | 0.53 | 0.52 | |
| Climate FNO | RMSE | 9.05 | 9.07 | 9.10 | 9.12 | 1.008 |
| | ACC | 0.57 | 0.56 | 0.54 | 0.53 | |
| *Architecture-Specific Baselines and FRL-Enhanced Versions* | | | | | | |
| Transformer (WeatherGFT) | RMSE | 9.01 | 9.07 | 9.22 | 9.34 | 1.037 |
| | ACC | 0.57 | 0.55 | 0.52 | 0.49 | |
| **Transformer + FRL** | **RMSE** | **9.01** | **7.81** | **7.11** | **6.70** | **0.744** |
| | **ACC** | **0.57** | **0.62** | **0.65** | **0.68** | |
| CNN (PDE-CNN) | RMSE | 9.07 | 9.17 | 9.36 | 9.58 | 1.056 |
| | ACC | 0.56 | 0.54 | 0.51 | 0.48 | |
| **CNN + FRL** | **RMSE** | **9.07** | **7.68** | **6.92** | **6.48** | **0.714** |
| | **ACC** | **0.56** | **0.63** | **0.66** | **0.69** | |
| Diffusion (ARCI) | RMSE | 8.88 | 8.94 | 9.03 | 9.09 | 1.024 |
| | ACC | 0.59 | 0.57 | 0.55 | 0.53 | |
| **Diffusion + FRL** | **RMSE** | **8.88** | **7.74** | **7.05** | **6.63** | **0.747** |
| | **ACC** | **0.59** | **0.63** | **0.66** | **0.67** | |
| GNN (Graph-EFM) | RMSE | 9.14 | 9.20 | 9.34 | 9.46 | 1.035 |
| | ACC | 0.55 | 0.53 | 0.50 | 0.47 | |
| **GNN + FRL** | **RMSE** | **9.14** | **7.87** | **7.20** | **6.82** | **0.746** |
| | **ACC** | **0.55** | **0.61** | **0.64** | **0.66** | |
| Neural ODE (ClimODE) | RMSE | 9.23 | 9.36 | 9.58 | 9.80 | 1.062 |
| | ACC | 0.54 | 0.52 | 0.49 | 0.46 | |
| **Neural ODE + FRL** | **RMSE** | **9.23** | **7.90** | **7.17** | **6.76** | **0.733** |
| | **ACC** | **0.54** | **0.60** | **0.63** | **0.65** | |

Table 30: Frequency response analysis of 3D weather forecasting.

| Model | Bandwidth (Hz) | H(f=75) | H(f=105) | Anchoring Ratio | Error Ratio |
|---|---|---|---|---|---|
| *Baseline Models* | | | | | |
| TNO | 90.58 | 0.998 | 0.285 | 3.50 | 0.181 |
| Climate FNO | 90.82 | 0.995 | 0.298 | 3.34 | 0.174 |
| Transformer | 89.89 | 1.001 | 0.196 | 5.11 | 0.185 |
| CNN | 88.77 | 0.973 | 0.104 | 9.33 | 0.120 |
| Diffusion | 87.53 | 1.081 | 0.067 | 16.08 | 0.099 |
| GNN | 91.75 | 1.027 | 0.383 | 2.68 | 0.222 |
| Neural ODE | 85.28 | 0.917 | 0.045 | 20.28 | 0.083 |
| *FRL-Enhanced Models* | | | | | |
| **Transformer+FRL** | **>120.00** | **1.021** | **1.022** | **1.00** | **0.370** |
| **CNN+FRL** | **>120.00** | **1.021** | **1.015** | **1.01** | **0.385** |
| **Diffusion+FRL** | **>120.00** | **1.068** | **1.026** | **1.04** | **0.345** |
| **GNN+FRL** | **>120.00** | **1.038** | **1.011** | **1.03** | **0.303** |
| **Neural ODE+FRL** | **>120.00** | **1.006** | **0.946** | **1.06** | **0.400** |

scale-anchored baseline fails to resolve Scale Anchoring ($\text{RMSE}_{\text{Ratio}} > 1$). The combination of FreqEnc+FreqLoss shows a trend toward mitigating Scale Anchoring ($\text{RMSE}_{\text{Ratio}} < 1$), with MultiRes+FreqLoss demonstrating more pronounced improvement, and FreqEnc+MultiRes achieving the most significant enhancement among pairwise combinations. These results indicate that frequency encoding enables the model to understand different resolutions, multi-resolution data allows the model to learn cross-scale mappings, and frequency loss facilitates frequency-aware learning to further improve spectral accuracy. Therefore, the three components of FRL operate through complementary mechanisms, and the core of FRL's ability to enhance methods' $\text{RMSE}_{\text{Ratio}}$ below 1 stems from the key innovation FreqEnc.

Table 31: Ablation study of FRL components on 3D fluid simulation. $\text{RMSE}_{\text{Ratio}}$ computed between $129^3$ and $32^3$ resolutions.

| Method | GNN | Trans. | CNN | Diff. | NO | N-ODE | NN | Avg. |
|---|---|---|---|---|---|---|---|---|
| Baseline | 1.018 | 1.021 | 1.060 | 1.041 | 1.017 | 1.338 | 1.035 | 1.076 |
| *Single Component* | | | | | | | | |
| + MultiRes | 1.012 | 1.015 | 1.048 | 1.032 | 1.011 | 1.285 | 1.028 | 1.062 |
| + FreqEnc | 0.912 | 0.918 | 0.942 | 0.935 | 0.908 | 1.158 | 0.922 | 0.957 |
| + FreqLoss | 1.015 | 1.018 | 1.052 | 1.035 | 1.014 | 1.305 | 1.030 | 1.067 |
| *Two Components* | | | | | | | | |
| + MultiRes+FreqLoss | 1.008 | 1.011 | 1.042 | 1.025 | 1.006 | 1.255 | 1.020 | 1.052 |
| + FreqEnc+MultiRes | 0.485 | 0.502 | 0.468 | 0.538 | 0.472 | 0.782 | 0.498 | 0.535 |
| + FreqEnc+FreqLoss | 0.285 | 0.302 | 0.268 | 0.335 | 0.272 | 0.582 | 0.298 | 0.335 |
| *Full FRL* | | | | | | | | |
| + All Three | 0.175 | 0.188 | 0.137 | 0.221 | 0.135 | 0.375 | 0.182 | 0.202 |

We further analyze the parameter sensitivity of FRL on the 3D ZS-SR fluid simulation task. Specifically, we evaluate the two key hyperparameters of FRL, the number of resolution hierarchy levels $J \in \{2, 3, 4, 5, 6\}$ and the frequency consistency loss weight $\lambda \in \{0.01, 0.05, 0.1, 0.2, 0.5\}$, as shown in Table 32. Across all architectures, the reduction in $\text{RMSE}_{\text{Ratio}}$ plateaus when $J > 3$ with $\lambda = 0.1$. Therefore, we select $J = 3$ for the main experiments in Section 6. When $\lambda \leq 0.1$ with $J = 3$, the $\text{RMSE}_{\text{Ratio}}$ decreases substantially while maintaining the original resolution RMSE. However, when $\lambda > 0.1$, the original resolution RMSE deteriorates significantly despite continued $\text{RMSE}_{\text{Ratio}}$ improvement. To balance training accuracy and scale decoupling capability, we select $\lambda = 0.1$ for the main experiments.

Table 32: Parameter sensitivity analysis of FRL on 3D fluid simulation averaged across all architectures.

| $J$ | $\lambda$ | | | | |
|---|---|---|---|---|---|
| | 0.01 | 0.05 | 0.1 | 0.2 | 0.5 |
| *$RMSE_{Ratio}$ ($129^3$/$32^3$)* | | | | | |
| 2 | 0.485 | 0.352 | 0.316 | 0.298 | 0.285 |
| 3 | 0.412 | 0.285 | **0.202** | 0.188 | 0.175 |
| 4 | 0.398 | 0.268 | 0.195 | 0.182 | 0.168 |
| 5 | 0.385 | 0.255 | 0.192 | 0.178 | 0.165 |
| 6 | 0.378 | 0.248 | 0.190 | 0.175 | 0.162 |
| *Original Resolution RMSE ($32^3$)* | | | | | |
| 2 | 0.00394 | 0.00394 | 0.00394 | 0.00408 | 0.00432 |
| 3 | 0.00394 | 0.00394 | **0.00394** | 0.00412 | 0.00440 |
| 4 | 0.00394 | 0.00394 | 0.00394 | 0.00416 | 0.00447 |
| 5 | 0.00394 | 0.00394 | 0.00394 | 0.00419 | 0.00453 |
| 6 | 0.00394 | 0.00394 | 0.00394 | 0.00422 | 0.00460 |

Lastly, to isolate the core innovation, we keep MultiRes and FreqLoss fixed and replace FreqEnc with three alternatives: an absolute-frequency encoding (AbsFreqEnc), standard Fourier features (FourierEnc), and vanilla positional encoding (PE). Under the same tuning budget, we evaluate the cross-Nyquist $RMSE_{Ratio}$. As shown in Table 33, all three replacement encodings perform similarly to the variant without any additional encoding ($\approx 1$), whereas the full FRL remains well below 1. This indicates that FRL's success does not stem from adding any positional/frequency encoding per se, but from the resolution-invariant, Nyquist-normalized frequency representation introduced by FreqEnc.

Table 33: Ablation on the FreqEnc component for 3D ZS-SR fluid simulation. $RMSE_{Ratio}$ computed between $129^3$ and $32^3$ resolutions. The three "replacement encodings" keep MultiRes+FreqLoss fixed and only swap FreqEnc.

| Method | GNN | Trans. | CNN | Diff. | NO | N-ODE | NN | Avg. |
|---|---|---|---|---|---|---|---|---|
| FRL | 0.175 | 0.188 | 0.137 | 0.221 | 0.135 | 0.375 | 0.182 | 0.202 |
| *FRL variants without the proposed normalized frequency encoding* | | | | | | | | |
| FRL w/ No Encoding | 1.008 | 1.011 | 1.042 | 1.025 | 1.006 | 1.255 | 1.020 | 1.052 |
| FRL w/ AbsFreqEnc | 1.010 | 1.014 | 1.045 | 1.028 | 1.009 | 1.260 | 1.022 | 1.055 |
| FRL w/ FourierEnc | 1.012 | 1.016 | 1.047 | 1.029 | 1.011 | 1.265 | 1.025 | 1.058 |
| FRL w/ PE | 1.011 | 1.015 | 1.046 | 1.027 | 1.010 | 1.262 | 1.023 | 1.056 |

## I  FAILURE MODES OF FREQUENCY REPRESENTATION LEARNING

The key assumption behind FRL is the existence of *learnable frequency-conditional relationships*: the underlying physical process must exhibit cross-scale spectral structure that is sufficiently regular so that frequency response patterns at different resolutions can be aligned in the normalized frequency domain. Many classical physical systems satisfy this assumption over a wide range of scales, but there are regimes in which the spectral structure itself changes qualitatively and the notion of a simple, resolution-invariant frequency response breaks down.

To illustrate such a regime, we perform a diagnostic experiment on channel-flow simulations at different Reynolds numbers. We train a CNN and its FRL-enhanced variant on high-fidelity numerical results on a $129 \times 129 \times 129$ grid (Nyquist frequency $\approx 256$ Hz), and analyze their empirical frequency responses; see Figure 6. At low to moderate Reynolds numbers ($Re \leq 10^3$), FRL successfully achieves Scale Decoupling: the Bandwidth (BW) remains close to the Nyquist frequency

and the Anchoring Ratio $H(f_{\text{in}})/H(f_{\text{out}})$ around the training Nyquist band is close to 1, indicating that high-frequency components are processed almost as reliably as low-frequency ones. However, as $Re$ increases, this behavior deteriorates. At $Re = 10^4$, even the FRL-enhanced CNN exhibits clear Scale Anchoring: the Anchoring Ratio $H(f{=}225)/H(f{=}275)$ grows from $1.00$ to $1.95$, and the effective BW shrinks to $262.13\,\text{Hz}$. At $Re = 10^5$, the Anchoring Ratio further increases to $5.24$ and the BW drops to $218.37\,\text{Hz}$, indicating that high-frequency components above the training Nyquist band are no longer reliably extrapolated.

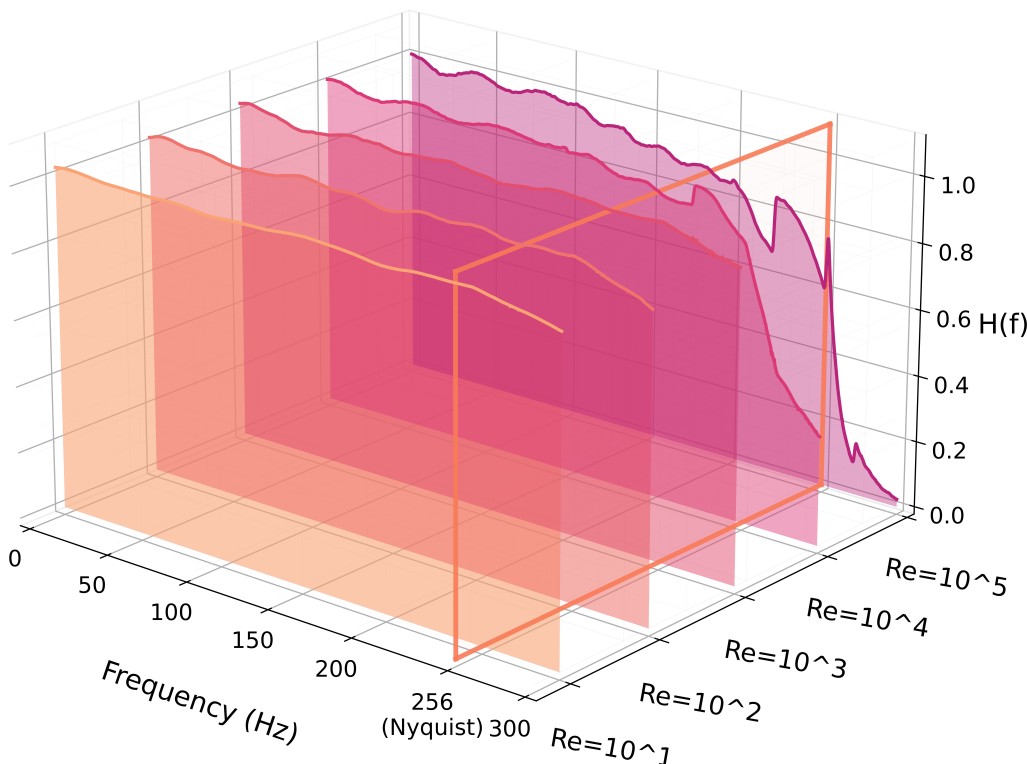

Figure 6: Frequency response analysis of an FRL-enhanced CNN at different Reynolds numbers $Re$ on channel-flow simulations. At low and moderate $Re$, FRL achieves stable high-frequency response and wide bandwidth; at very high $Re$, the frequency response becomes sharply anchored near the training Nyquist frequency, revealing a failure mode of FRL.

Notably, this failure at very high Reynolds numbers should not be interpreted specific to FRL. Rather, it reflects a broader challenge of high-$Re$ turbulence for methods in general. Prior work on deep-learning-based super-resolution of turbulent flows typically considers a limited range of Reynolds numbers and reports accurate reconstruction only for Reynolds numbers within, or close to, the training range (Yousif et al., 2022). High-$Re$ regimes remain significantly more challenging and are comparatively under-explored. In our setting, models must additionally perform zero-shot super-resolution in time and frequency, which further amplifies the difficulty.

FRL succeeds in the moderate-$Re$ regimes considered in the main experiments because its Nyquist-normalized frequency representation can implicitly learn stable relationships between adjacent spectral bands: the statistical patterns linking relatively low-frequency components to higher-frequency components remain coherent, smooth, and to a large extent predictable. This coherence is characteristic of transitional flows or weak turbulence, where cascade structures are not yet fully developed and cross-band mappings remain statistically consistent (Cerbus et al., 2020). In this situation, the assumption of a smooth, cross-scale-consistent gain function $H_\Theta(\xi)$ in normalized frequency coordinates is reasonable, and FRL can exploit it to mitigate Scale Anchoring.

When the Reynolds number becomes extremely high, however, the flow enters a fully developed turbulent state. The inertial cascade becomes strongly nonlinear, intermittent, and non-stationary;

energy transfer across scales is dominated by complex multi-scale interactions rather than smooth, locally extrapolatable couplings between neighboring frequency bands (She & Leveque, 1994). In this regime, the coupling between adjacent spectral bands is no longer locally smooth, and the cross-band relationships that FRL attempts to model cease to be well behaved. Additionally, the cross-scale mappings encoded in the normalized frequency representation were calibrated in regimes with more regular spectral structure. Consequently, these mappings do not remain valid at very high $Re$, and FRL fails to extrapolate reliably to high-frequency components.

It is natural to ask whether more elaborate engineering, such as adaptive mechanisms or PDE-informed soft constraints, could restore FRL's effectiveness. Such techniques can indeed trade additional computational cost for incremental accuracy gains, but fundamentally extending FRL to extremely high-$Re$ flows would require accurately characterizing the relational patterns between adjacent spectral bands under those conditions, for which no simple closed-form description is available. Learning these relationships purely from data would also demand prohibitive computational resources, given the power-law scaling between $Re$ and the grid resolution required for DNS.

On the other hand, when the Navier-Stokes equations are appropriately non-dimensionalized and the high-$Re$ limit is considered, small-scale statistics are believed to approach universal behavior and follow Kolmogorov-type scaling laws, such as the inertial-range energy spectrum $E(k) \propto k^{-5/3}$ (She & Leveque, 1994). In the spectral domain, this suggests that in the asymptotic high-$Re$ regime the energy distribution over adjacent frequency bands is constrained by a fixed physical pattern. This observation offers a potential handle for extending FRL: by aligning FRL's frequency-domain representation and loss with Kolmogorov-type spectral constraints, one could guide the model to extrapolate according to the theoretically expected inertial-range behavior, rather than relying solely on data-driven cross-band fitting.

In summary, replacing traditional DNS/RANS/LES with deep learning models for very high-$Re$ fluid simulation remains an important and challenging problem that goes beyond the ZS-SR STF setting studied in this paper. FRL is designed as a general enhancement for mitigating Scale Anchoring in ZS-SR STF under systems that exhibit scale-consistent spectral structure, and our experiments confirm its effectiveness. Extending FRL to extremely high-$Re$ turbulent flows will likely require incorporating explicit physical spectral constraints (e.g., Kolmogorov scaling) into the normalized frequency representation and loss, which we leave as promising future work.

## J   DETAILED CLARIFICATION OF LARGE LANGUAGE MODELS USAGE

Table 34: Summary of Large Language Model (LLM) Usage in This Work

| Purpose of LLM Usage | Used |
|---|---|
| Aid or polish writing | ✓ |
| Retrieval and discovery (e.g., finding related work) | × |
| Research ideation | × |
| Other purposes | × |

In accordance with ICLR 2026 policy on transparent disclosure of Large Language Model usage, we declare that LLMs were employed exclusively to assist with the writing and presentation aspects of this paper. Specifically, we utilized LLMs for: (i) verification and refinement of technical terminology to ensure precise usage of domain-specific vocabulary; (ii) grammatical error detection and correction to enhance the clarity and readability of the manuscript; (iii) translation assistance from the authors' native language to English, as we are non-native English speakers, to ensure accurate and fluent expression of scientific concepts; and (iv) improvement of sentence structure and flow while maintaining the original scientific content and meaning. We emphasize that LLMs were not used for research ideation, experimental design, data analysis, or any form of content generation that would constitute intellectual contribution to the scientific findings presented in this work. All scientific insights, methodological decisions, and analytical conclusions are the original work of the authors. The use of LLMs was limited to linguistic and presentational enhancement only, serving a role analogous to professional editing services.

