# OpenReview forum: "Breaking Scale Anchoring: Frequency Representation Learning for Accurate High-Resolution Inference from Low-Resolution Training"
_ICLR.cc/2026/Conference — ICLR 2026 Poster_

### Official Review · Reviewer_Kovs · 2025-10-20

**Soundness:** 3
**Presentation:** 3
**Contribution:** 2
**Rating:** 6
**Confidence:** 3

**Summary:**

This paper involves downsampling the data to different resolutions, then using a special Nyquist-normalized frequency position encoding, along with a frequency consistency loss, to achieve super-resolution. Results on two tasks demonstrate the effectiveness of the proposed method.

**Strengths:**

1. The paper introduces a new approach based on the frequency domain to address the limitations in current Zero-Shot Super-Resolution Spatiotemporal Forecasting methods. It explains the Nyquist frequency limitation of low-resolution data and offers a new perspective on why current methods struggle with high-quality high-resolution predictions.
2. The paper highlights the flaws in traditional multi-resolution techniques, especially the problems of scale anchoring and spectral bias. It proposes Nyquist-normalized frequency representation and frequency-aware loss to solve these issues, which improves the model's performance.
3.The proposed method demonstrates strong performance in weather forecasting and fluid simulation tasks, highlighting the effectiveness of the approach and its potential for practical applications.
4. The proposed method is validated on multiple network architectures, including CNN, Transformer, Mamba, and GNN, demonstrating its versatility and effectiveness across different networks.

**Weaknesses:**

1. The method shares similarities with ZSSR and other Deep Internal Learning approaches, making it unclear what is fundamentally new beyond the frequency-domain reinterpretation.
2. The experiments are limited to spatiotemporal forecasting; applying the method to standard image super-resolution datasets (like Set5, Set14, BSD100) would provide more comprehensive comparisons.
3. The frequency consistency loss seems similar to existing frequency-domain losses, and its contribution may be limited. Additional ablation studies comparing it with other methods could clarify its impact.

**Questions:**

In the experiments, the method is tested at only a few discrete scales. Could you test the method over a wider range of scales by adjusting the scale in fixed steps (e.g., 2x, 3x, ... 64x) and report the results? Plotting a performance curve with these varying scales would help demonstrate the method's effectiveness and robustness across a broader range of resolutions.

**Details Of Ethics Concerns:**

This paper has no ethical concerns.

---

> ### Author Response · Authors · 2025-11-14
> **Reply - Part 1**
>
> We greatly appreciate your recognition of the paper's strengths and your insightful criticism of its weaknesses.
>
> Regarding your criticisms of the paper's weaknesses, we respectfully note that:
>
> ### Clarification for Weakness 1:
>
> We first apologize for not clearly explaining the task in the manuscript, which led to confusion between our work and ZSSR. We explicitly define these tasks here:
>
> - **Super Resolution (SR)** task refers to inputting a single low-resolution image/snapshot and outputting the corresponding single high-resolution image
> - **Zero-Shot Super Resolution (ZSSR)** task refers to having only a single image in the original training set, inputting a single low-resolution image/snapshot and outputting the corresponding single high-resolution image
> - **Spatiotemporal Forecasting (STF)** task refers to inputting one or multiple physical field snapshots at different time steps and outputting the physical field snapshot at the next time step, repeated multiple times
> - **Zero-Shot Super Resolution Spatiotemporal Forecasting (ZS-SR STF)** task refers to training the model on low-resolution snapshots, but at inference time inputting one or multiple high-resolution physical field snapshots at different time steps and outputting the high-resolution physical field snapshot at the next time step, repeated multiple times
>
> In other words, ZS-SR STF and ZSSR belong to different tasks and are not even in the same broad field. Therefore, the methods and technical approaches used for these two tasks do not share much similarity.
>
> Additionally, we restate the novelty of this work from both problem and method perspectives:
>
> - **Regarding the problem:** Previous studies have not clearly identified the core error source in ZS-SR STF, whereas the Scale Anchoring problem explains that the model's main error during high-resolution inference stems from signals in frequency bands above the training Nyquist frequency. This is important because identifying the problem is the first step toward solving it.
>
> - **Regarding the method:** Previous studies focus on how models themselves maintain similar performance across different resolutions during inference, whereas FRL, from the perspective of spectral domain Scale Anchoring, enables the model to learn how to extrapolate in the frequency domain to higher-frequency signals not seen during training. For technical details on why FRL is novel, please see Clarification for Weakness 3.
>
> ### Clarification for Weakness 2:
>
> We agree that the datasets Set5/Set14/BSD100 you suggested are important for image reconstruction, but they are used for evaluating SR tasks rather than STF. The fundamental reason these tests are not applicable is that FRL and all baselines are not designed for these datasets or to solve the ZSSR problem.
>
> ### Clarification for Weakness 3:
>
> Yes, in fact, the first step of FRL (multi-resolution training) and the third step (adding a frequency constraint) are both existing methods. We cite and explain the existence of these two types of methods in Section 2.2 Multi-Resolution Generation and Section 2.3 Spectral Bias, respectively.
>
> However, it is worth noting that the core of FRL—which enables the model to learn frequency domain extrapolation and achieve an Error (RMSE) Ratio below 1—lies in the second step: the design of normalized frequency representation. To our best knowledge, since the perspective of existing methods differs significantly from Scale Anchoring, the representation design in the second step is entirely novel. The designs of the first and third steps are intended to complement the second step to further reduce errors.
>
> In fact, we conducted corresponding ablation studies for this claim that the second step is the core contribution. We mentioned this experiment in the first paragraph of Section 7 Discussion, which is presented in detail in Appendix G (page 33). Here, we concisely restate the key results and conclusions: On the 3D ZS-SR fluid task, we tested the following combinations for each architecture, comparing cross-resolution RMSE Ratio (129³ vs 32³): (i) adding any single component individually, (ii) pairwise combinations, (iii) complete FRL. The RMSE Ratio for any ablation component that does not include the second step (normalized frequency representation) is greater than 1 (error increases), while methods including the second step all have RMSE Ratio less than 1, with more complete component combinations yielding smaller errors. Therefore, normalized frequency representation is the key to breaking Scale Anchoring, with the first and third steps serving as auxiliary designs.

---

> ### Author Response · Authors · 2025-11-14
> **Reply - Part 2**
>
> Additionally, here are our response to your questions:
>
> ### Answer to Question 1:
>
> Your idea of testing at different discrete scales is correct. However, we would like to carefully explain the rationale for the current experimental setup:
>
> (1) We use high-resolution spatiotemporal datasets contributed by other works at high temporal and computational cost as ground truth, and coarsen them to low-resolution training data. Coarsening a physical field typically involves simultaneous coarsening in all directions. For example, when we coarsen 3D fluid field data with original resolution 129³ by 2×, 3×, 4×, we obtain 64³, 43³, 32³ respectively. This appears as coarsening by irregular factors of 8.19×, 27×, 65× when looking at the total number of grid points in the entire field.
>
> (2) When a physical field is coarsened beyond a certain degree, it loses too much information and causes the model to fail. For example, coarsening 129³ to 4× yields 32³, which still preserves the basic flow field structure. But when coarsened to 5× yielding 25³, the model suddenly fails to converge. Our current maximum coarsening factor is empirically chosen with reference to the upper limit at which the model can learn.

---

> ### Author Response · Authors · 2025-11-25
>
> Dear Reviewer,
>
> I hope this message finds you well. With about one week remaining in the discussion period, we wanted to confirm whether we have adequately addressed all your concerns. If you have any additional comments or suggestions you would like us to consider, please let us know. Your insights are invaluable to us, and we are eager to address any remaining issues during the remaining time to improve our work.
>
> Additionally, in response to the reviews we have received, we have revised our paper as Rebuttal Version 1, with modifications highlighted in blue text throughout the manuscript. The detailed changes are summarized in our global comment "Rebuttal Version 1." Here, we reiterate the modifications made in response to the weaknesses and questions you raised:
>
> - Addressing your weaknesses 1&2, we revised lines 88–91 (in 2 Related Work):
>
>     Revised the definitions of STF and ZS-SR STF to make them more self-contained.
>
> - Addressing your weaknesses 3, we revised lines 102–132 (in 2 Related Work):
>
>   Consolidated all related issues into a single subsection.
>
>   Clearly distinguished Scale Anchoring from existing problems.
>
>   Discussed all relevant existing problems and mainstream solutions.
>
>   Pointed out that Step 1 and Step 3 of FRL are variants of existing techniques.
>
>   Analyzed differences between SA and existing problems from the perspectives of source, scope, and consequence.
>
>   And lines 301–308 (in 5 Solution for Scale Anchoring):
>
>   Explicitly stated that Steps 1 and 3 of FRL resemble existing methods.
>
>   Emphasized that Step 2 is the key to addressing SA, and cited the ablation study in Appendix H to support this claim.
>
> - Addressing your question 1, we revised lines 327–344 (in 6 Empirical Evaluation):
>
>     Added explanations to each dataset on how low-resolution data were obtained and why the specific downsampling factors and maximum factors were chosen.
>
> Thank you for your time and effort in reviewing our paper.

---

### Official Review · Reviewer_7EgH · 2025-10-28

**Soundness:** 3
**Presentation:** 3
**Contribution:** 3
**Rating:** 6
**Confidence:** 3

**Summary:**

The authors identify a problem when performing zero-shot super resolution, when a model is trained on low‐resolution data and then deployed on higher‐resolution data, the error does not decrease despite more fine‐grained input. They argue this is because low‐resolution training data limits the maximum representable frequency (via Nyquist), so the model never “learns” higher‐frequency components. They propose a method called Frequency Representation Learning (FRL) that (a) uses resolution‐aligned frequency representations (normalized to account for the Nyquist at each resolution) and (b) incorporates a spectral‐consistency training objective across multiple resolution levels. They show experiments on spatio‐temporal forecasting tasks (zero-shot super‐resolution forecasting) where their FRL approach results in error decreasing with increasing resolution (unlike baselines) and improved frequency response stability in higher‐frequency bands.

**Strengths:**

1. The identification of the “scale anchoring” problem is interesting and relevant in several fields, such as neural operators.

2. The proposed method (FRL) is architecture‐agnostic in principle.

3. The experiments show that the inference error can be even lower for zero-shot super resolution. Existing works usually consider the same error as a success.

**Weaknesses:**

Some of these weaknesses below fall somewhere between questions and weaknesses, so I’ve included them here.

1. A central conceptual concern is that the zero-shot high-resolution inference task itself appears ill-posed under the stated assumptions. If the model is trained purely on low-resolution data, its training distribution is inherently band-limited by the Nyquist frequency of that grid. Consequently, the high-frequency components present in the high-resolution domain are unobserved and unidentifiable during training.
From a signal-theoretic standpoint, there exist infinitely many high-frequency realizations consistent with the same low-frequency field, so the inverse mapping from coarse to fine scales has no unique solution. Therefore, it is unclear in what precise sense the model can achieve lower error at higher resolution, may be to test model generalization in some sense?

2. It seems that the testing datasets are fairly smooth, and the low-frequency modes dominate the dynamics. Experiments on data with high-frequency or multi-scale content can provide more insights.

3. FRL seems to rely implicitly on the assumption that the underlying system exhibits scale-consistent spectral structure (as in turbulence or diffusion). Can the authors elaborate more on this?

4. Zero-shot super resolution suffers from aliasing/discretization errors as noted in [1,2,3], there is no discussion on these works and there is no discussion on if and how FRL mitigates these issues.

[1] Representation Equivalent Neural Operators: a Framework for Alias-free Operator Learning
[2] Discretization-invariance? On the Discretization Mismatch Errors in Neural Operators
[3] Discretization Error of Fourier Neural Operators

**Questions:**

1. Given that training data are band-limited by the Nyquist frequency, what information allows FRL to improve predictions beyond that spectral limit?

2. Does FRL rely on the implicit assumption that the underlying physical process is scale-consistent or spectrally smooth?

3. Since multiple high-resolution fields can correspond to the same low-resolution sample, how does FRL regularize or constrain the mapping to select a physically meaningful one?

4. What is the theoretical justification for expecting a neural operator trained on low-resolution data to exhibit convergence behavior analogous to that of a numerical solver?

---

> ### Author Response · Authors · 2025-11-14
> **Reply - Part 1**
>
> We greatly appreciate your recognition of the paper's strengths and your insightful criticism of its weaknesses.
>
> Regarding your criticisms of the paper's weaknesses, we respectfully note that:
>
> ### Clarification for Weakness 1:
>
> Your understanding is correct: according to the Shannon-Nyquist theorem, low-resolution data inherently cannot contain signals above the Nyquist frequency, so models trained on low resolution can only "generalize" to high-resolution inference. We do not claim to solve this problem, as this problem can only be mitigated. Specifically, ZS-SR STF is not an inverse problem of solving the original operator from low-resolution data. We only view it here as an extrapolation problem of adjacent frequency-domain signal patterns, and we elaborate on the advantages and limitations of this perspective in the following two sections.
>
> ### Clarification for Weakness 3:
>
> As verified in Validation Experiment A.2 in Section 4.2 Experiment Validation: the source of error when the model performs high-resolution inference is precisely the signal above the training Nyquist frequency, or what you referred to as "ill-posed" in Weakness 1. This portion of error cannot be completely eliminated, but do high-frequency signals have extrapolatable patterns that can reduce this portion of error?
>
> The principle by which FRL works is to promote the model's learning of relational patterns between signals in the spectral domain through normalized frequency representation and frequency-domain loss functions. When "the underlying system exhibits scale-consistent spectral structure," this relationship can be applied to the spectrum of high-resolution data to reduce errors caused by signals in frequency bands above the training Nyquist frequency. In this case, the problem becomes less "ill-posed." More specifically, FRL is relatively stable at Re where turbulence has not yet occurred.
>
> ### Clarification for Weakness 2:
>
> Beyond the most common class of fluid datasets, a small number of datasets provide DNS/RANS/LES results at extremely high Re. When the Reynolds number becomes extremely high, the flow enters a fully developed turbulent state, where the inertial cascade becomes strongly nonlinear, intermittent, and non-stationary. In this case, the coupling between adjacent spectral bands is no longer locally smooth or extrapolatable; rather, it becomes dominated by chaotic multi-scale interactions and energy transfer.[1] These phenomena violate the fundamental assumptions that allow FRL to learn. Therefore, the normalized cross-band mappings learned during training are no longer valid, leading to failure of high-frequency extrapolation once the system enters the extremely high Re range.
>
> However, it is worth noting that turbulence has not yet been precisely solved mathematically, and even DNS/RANS/LES as ground truth are merely approximations of temporal integration of multiple sub-problems of an IBVP.[2] Using deep learning to replace DNS/RANS/LES for high Re fluid simulation is a challenging problem, typically addressed by specialized research rather than as a general contribution.[3] Similarly, FRL is proposed as a general enhancement scheme to address Scale Anchoring in ZS-SR STF, not as a specialized method for solving high Re fluid simulation.
>
> In summary, the current applicable scope of FRL can be limited to "systems exhibiting scale-consistent spectral structure." It has been validated as applicable on the common experimental settings widely tested in the paper. However, under more extreme conditions not extensively tested, i.e., at extremely high Re, FRL is also theoretically extensible: at extremely high Re, the energy spectra of signals in adjacent frequency bands have mathematically describable relationships (see our Reply to Reviewer SPE1 - Part 2). But this requires more specialized research and exploration of this sub-problem, just like many studies specifically addressing high Re STF problems.
>
> [1] She, Zhen-Su, and Emmanuel Leveque. "Universal scaling laws in fully developed turbulence." Physical review letters 72.3 (1994): 336.
>
> [2] Moin, Parviz, and Krishnan Mahesh. "Direct numerical simulation: a tool in turbulence research." Annual review of fluid mechanics 30.1 (1998): 539-578.
>
> [3] Li, Zhijie, et al. "Long-term predictions of turbulence by implicit U-Net enhanced Fourier neural operator." Physics of Fluids 35.7 (2023).

---

> ### Author Response · Authors · 2025-11-14
> **Reply - Part 2**
>
> ### Clarification for Weakness 4:
>
> Thank you for pointing out these related works concerning aliasing/discretization errors in NOs. We agree that they are highly relevant to Scale Anchoring, but we want to point out the distinction in focus of these problems. Let us first clarify the problems explored in the three papers you cited:
>
> - [1] analyzes "aliasing" and defines representational equivalence to ensure that different discretizations of the same continuous operator are consistent.
> - [2] introduces "Discretization Mismatch Error (DME)": the same NO produces different outputs on different grids. They demonstrate how this error accumulates across layers.
> - [3] studies the "discretization error" of FNO, i.e., how the continuous FNO and its FFT-based discrete implementation deviate as grid resolution changes.
>
> These phenomena arise from **the choice of NO architecture and numerical discretization schemes** (e.g., how the operator is discretized and implemented), with the goal of making the same continuous operator behave consistently on different grids. In contrast, Scale Anchoring comes from fundamental limitations of **data observation** rather than the model itself, and applies to any deep model used for ZS-SR STF: as we validated in Sections 3 Existence of Scale Anchoring and 4 Mechanism of Scale Anchoring on CNN, Transformer, Diffusion, Neural ODE, Mamba, and NO.
>
> Specifically, works [1][2][3] ask: "Given a continuous operator and a specific NO architecture, how do we design aliasing-free and discretization-consistent implementations?" Our work asks a different question: "Given that all architectures are trained only on low-resolution data, with their spectra constrained by the training Nyquist limit, why does error stop improving at higher resolutions? And how can we reduce this Scale Anchoring effect in ZS-SR STF?"
>
> Moreover, as with your understanding in Weakness 1, even if the operator architecture is completely aliasing-free and discretization-invariant, a model trained only on low-resolution data will still never observe frequency components above the training Nyquist frequency. Therefore, FRL is orthogonal and compatible with the aliasing/discretization literature. We believe a systematic study of the extent to which FRL reduces DME/aliasing in NOs is an interesting future direction, but beyond the current scope. In this paper, we focus on (i) defining and analyzing Scale Anchoring as a fundamental limitation of ZS-SR STF, and (ii) demonstrating that FRL mitigates this limitation across multiple architectures.
>
> We provide a more specific systematic analysis of the distinction between Scale Anchoring and other existing problems in the Clarification to Weakness 1&2 in our Reply to Reviewer cSKw - Part 1. We plan to supplement the Related Work section in the revised version with a detailed elaboration of the distinction between Scale Anchoring and [1][2][3] and other problems.
>
> ---
>
> Additionally, here are our responses to your questions:
>
> ### Answer for Question 1:
>
> The information/data patterns that FRL uses to extrapolate high-frequency signals are: the shape of the frequency response of the target physical process in normalized frequency coordinates and its conditional relationship with local context. Specifically, FRL learns how to use conditions: the power spectral envelope and slope of low/mid-frequency bands and local intensity/gradient/strain, etc., to predict: the spectral envelope continuation and slope/break point position, directional gain, etc., of high-frequency bands, to reconstruct a resolution-invariant conditioned spectral transfer function, thereby achieving a wider effective bandwidth and more robust high-frequency response at higher Nyquist frequencies.
>
> ### Answer for Question 2:
>
> It requires "systems exhibiting scale-consistent spectral structure" as you thought. Once the small-scale physics undergoes qualitative changes (e.g., high Re), this extrapolation degrades. Specific analysis can be found in Clarification for Weaknesses 1-3.
>
> ### Answer for Question 3:
>
> FRL itself does not guarantee physical consistency, but FRL does not damage the physical consistency of the baseline. The third step of FRL is to add an additional frequency-domain consistency loss to the loss function. Whether the FRL-enhanced variant guarantees physical consistency depends on the loss design of the baseline.

---

> ### Author Response · Authors · 2025-11-14
> **Reply - Part 3**
>
> ### Answer for Question 4:
>
> FRL cannot guarantee strict order convergence like numerical solvers, and can only mitigate the high-frequency component error caused by Scale Anchoring. The reason it can extrapolate to parts above the training Nyquist on physical frequencies is that it reformulates the learning problem in the normalized frequency domain and performs cross-scale alignment, making *absolute frequencies outside the training band* become *learnable points within the normalized band* on finer grids.
>
> Specifically, let the grid spacing be $\Delta x$, and the Nyquist wavenumber $k_N(\Delta x)=\pi/\Delta x$. For the spectrum $\widehat{u}(k)$ of a continuous field $u$, under the approximation of "equivalent linear response," denote the model's response in the frequency domain as
>
> $$\widehat{F_\theta(u)}(k;\Delta x) \approx H_\theta(\xi) \widehat{u}(k), \qquad \xi=\frac{|k|}{k_N(\Delta x)}\in[0,1],$$
>
> where $\xi$ is the normalized wavenumber, and $H_\theta(\xi)$ is a complex-valued gain function that is cross-scale consistent (allowing small deviation $\varepsilon$).
>
> At test resolution $\Delta x'$, let $S_u(k)$ be the power spectral density (PSD) of $u$. Then by variable substitution $k=k_N(\Delta x')\xi$ ($dk=k_N(\Delta x')d\xi$), define
>
> $$S_{u}^{(\Delta x')}(\xi) = k_N(\Delta x') S_u\big(k_N(\Delta x')\xi\big).$$
>
> The dominant term of the squared error in the frequency domain can be approximately decomposed as
>
> $$\mathrm{MSE}(\Delta x') \approx \int_0^1 \big|H_\theta(\xi)-1\big|^2 S_{u}^{(\Delta x')}(\xi) d\xi + \mathcal{R}_{\text{aleatoric}} + O(\varepsilon),$$
>
> where the first term is the learnable "calibration error" and $\mathcal{R}_{\text{aleatoric}}$ is the conditional variance of multiple high-res solutions for the same low-res sample.
>
> If training uses $\Delta x_{\text{tr}}$ and testing uses finer $\Delta x' < \Delta x_{\text{tr}}$, let
>
> $$\rho=\frac{k_N(\Delta x_{\text{tr}})}{k_N(\Delta x')}\in(0,1).$$
>
> Then all absolute frequencies $k\in\big(k_N(\Delta x_{\text{tr}}), k_N(\Delta x')\big]$ correspond to normalized coordinates on the test grid
>
> $$\xi'=\frac{|k|}{k_N(\Delta x')}\in(\rho,1]\subset(0,1],$$
>
> so these "out-of-training-band" absolute frequencies fall back into the normalized domain $[0,1]$ at test time, and can be smoothly calibrated by the learned $H_\theta(\xi)$ (interpolation/extrapolation in the $\xi$ sense).
>
> Furthermore, let
>
> $$f_{\text{OOB}}(\rho) = \frac{\int_{\rho}^{1} S_{u}^{(\Delta x')}(\xi) d\xi}{\int_{0}^{1} S_{u}^{(\Delta x')}(\xi) d\xi}$$
>
> represent the energy fraction outside the training Nyquist relative to the test grid. If on $(\rho,1]$ we have
>
> $$\sup_{\xi\in(\rho,1]}|H_\theta(\xi)-1|\le \delta,$$
>
> while the "anchored" baseline can be approximated as $H_{\text{anch}}(\xi)\approx \mathbf{1}_{[0,\rho]}(\xi)$, then the relative improvement roughly satisfies
>
> $$\frac{\mathrm{MSE}_{\text{FRL}}}{\mathrm{MSE}_{\text{anch}}} \lesssim 1-\big(1-\delta^2\big) f_{\text{OOB}}(\rho) + O(\varepsilon) + \frac{\mathcal{R}_{\text{aleatoric}}}{\mathrm{MSE}_{\text{anch}}}$$
>
> This is not an order convergence theorem, but merely states: the more energy distributed in $(\rho,1]$ and the closer $H_\theta$ is to 1 in that segment (smaller $\delta$), the more likely the error will decrease; $\mathcal{R}_{\text{aleatoric}}$ provides the residual lower bound of the "low-res→high-res ill-posedness."

---

> > ### Comment · Reviewer_7EgH · 2025-11-14
> >
> > Thank you for the responses. I will read them in detail later, probably early next week. In the meantime, I would appreciate it if you could provide the revised paper (This is not a request; if you happen to have it ready, please upload it; **if not, no worries at all**). I will review both the revised paper and your responses here before replying.

---

> > > ### Author Response · Authors · 2025-11-15
> > >
> > > We appreciate your time reviewing the paper and providing feedback, so we plan to revise the paper based on the first-round reviews. If it is uploaded before you read the responses, there should be a notification. Enjoy your weekend!

---

> ### Comment · Reviewer_7EgH · 2025-11-19
>
> Thank you for your detailed response. Overall, my misunderstanding/questions/concerns have been addressed, except for one remaining point regarding the Clarification for Weakness 3.
>
> - W3: "As verified in Validation Experiment A.2 in Section 4.2 Experiment Validation: the source of error when the model performs high-resolution inference is precisely the signal above the training Nyquist frequency, or what you referred to as "ill-posed" in Weakness 1."
>     - Could you clarify what this means? My understanding is that you are suggesting the neural network’s predictions are (similar to “spectral bias”). However, as you noted in your response to W2, the test data itself is also quite smooth. I do not clearly see a problem of energy decay for high-frequency modes. If your data were turbulent, i.e. high Re, I would agree that this is an issue.
>     - When performing high-resolution inference above the Nyquist frequency, the error caused is not only on high-frequencies, but also on low-frequencies. In fact, as shown in the DME paper, low frequency modes even suffer more.
>
> - Q4: Providing a full mathematical proof was not my original intention (given the time constraint and workload during rebuttal), I was mainly hoping for some intuition. I appreciate the author’s effort in offering such a thorough explanation although it is beyond my expertise to check. Anyway, after reading the responses to the other weaknesses and questions, I believe I already have an answer.

---

> ### Author Response · Authors · 2025-11-20
>
> We apologize for our overly absolute wording in our previous response, which may have caused confusion. We are pleased to provide more intuitive clarifications here.
>
> In the statement from W3:
>
> > "the source of error when the model performs high-resolution inference is precisely the signal above the training Nyquist frequency ..."
>
> The use of "precisely" is indeed inaccurate. As shown in Table 2, we should say that during high-resolution inference, the additional error beyond what already exists at training resolution is relatively **dominated** by frequencies above the training Nyquist band. At 2× and 4× higher resolutions, errors limited to frequencies below the training Nyquist account for only approximately 34%-58% and 22%-42% of the total error energy across architectures, respectively.
>
> Therefore:
>
> ### (i)
>
> > Could you clarify what this means? My understanding is that you are suggesting the neural network's predictions are (similar to "spectral bias"). However, as you noted in your response to W2, the test data itself is also quite smooth. I do not clearly see a problem of energy decay for high-frequency modes. If your data were turbulent, i.e. high Re, I would agree that this is an issue.
>
> Conceptually, Scale Anchoring is not specific to turbulence. It occurs when we train at one resolution and then perform ZS-SR STF at a finer resolution.
>
> As shown in Figures 4 and 5 (these two figures on test data correspond exactly to the "energy decay for high-frequency modes" demonstrated in Figure 3), when we perform high-resolution inference: baselines show lower signal energy for frequencies above the training Nyquist when inferring at high resolution, while FRL-enhanced variants maintain high-energy extrapolation. This demonstrates that even in relatively smooth scenarios, baselines fail at frequency-domain extrapolation. The result is that baseline errors remain nearly unchanged during high-resolution extrapolation, while FRL-enhanced variant errors decrease. Whether the scenario is turbulent does not affect the general result of baseline failure—it only affects FRL's extrapolation capability.
>
> P.S. We do not consider Scale Anchoring to be similar in principle to Spectral Bias. An intuitive understanding is that Spectral Bias refers to the model's **tendency** to fit from low to high frequencies on **seen** frequencies (below training Nyquist), while Scale Anchoring refers to the model's difficulty in handling **unseen** high frequencies (above training Nyquist). In terms of results, the two problems also emphasize different failure frequency bands.
>
> ### (ii)
>
> > When performing high-resolution inference above the Nyquist frequency, the error caused is not only on high-frequencies, but also on low-frequencies. In fact, as shown in the DME paper, low frequency modes even suffer more.
>
> We fully agree that errors at high resolution appear in both high-frequency and low-frequency components. Again, we apologize for the inaccurate use of the word "precisely."
>
> The DME paper shows that for FNO in their Navier-Stokes setting, discretization mismatch leads to large low-frequency errors (Figure 5). We do not dispute this empirical observation. However, the DME paper also reports that CROP with low DME exhibits more uniform error spectra across frequencies.
>
> In our experiments, the situation is closer to CROP: we consider eight different architectures with similar overall error magnitude scales at different test resolutions. When we decompose the squared prediction error into two frequency bands relative to the training Nyquist frequency (below vs. full spectrum up to test Nyquist), Table 2 shows that more than half of the total error energy comes from frequencies above the training Nyquist band. This is what we mean by saying that in our setting, the additional error is relatively "dominated" by the portion above the Nyquist band. This does not conflict with what the DME paper observes for CROP.
>
> In other words, in the case of FNO in the DME paper's Navier-Stokes setting, DME may manifest as low-frequency-dominated error. Our analysis in validation experiment A.2 shows that in our advection-diffusion setting and frequency-band aggregation metric, the incremental error energy is primarily contributed by frequencies above the training Nyquist. We view these as different empirical patterns arising from different PDE systems, architectures, and error metrics, rather than contradictory statements.

---

> > ### Comment · Reviewer_7EgH · 2025-11-20
> >
> > Thanks for the detailed response. No worries, I understand. I just want to clarify.
> >
> > I have one suggestion and a follow-up question:
> >
> > 1. I would recommend mark your revision in a different color so that it is more obvious to check.
> >
> > 2. Regarding the same question, in Figure 3 of your paper, taking the NO result as an example, how do you perform super-resolution inference? Are you using SFNO? If so, what mode cutoff do you apply? My understanding is that Fig.3 is different from Fig. 5 in the DME paper. The result in your Fig. 3 is already "normalized" to be "disproportional" to different modes. Can you perform the same L2 error visualization for different modes as in Fig. 5 of the DME paper? I would suspect that if you do the same, you will observe a higher error in the low frequency modes as well because the error comes from the discretization inherent to the neural operator itself, arising mainly from the activation functions and propagating through layers.
> >      - If that's the case, I would suggest adding some discussion/clarification to your Fig. 3.

---

> > > ### Author Response · Authors · 2025-11-21
> > >
> > > ### (iii) Frequency-Band Decomposed Error Analysis Using Our Error Metric
> > >
> > > To provide an error analysis with frequency-band decomposition aligned with DME using our error metric, we performed additional analysis on the same NO used in A.2 as well as an FNO baseline with the same mode truncation. We trained both models on 64² data as described above. At test time we evaluate them on 128² and run multi-step (50) high-resolution unrolling. At each step we compute the same FFT-based L2 error as in A.2: we construct the error field (e_t(x) = u_pred(t,x) - u_true(t,x)), take its FFT (ê_t(k)), and use the error spectrum (|ê_t(k)|²). We then average this error spectrum over time and batch, and radially average to obtain a 1D error spectrum (E_err(k)) as a function of modal index radius (k). Finally, we divide this radial axis into frequency bands, using the training Nyquist frequency (k ≈ 32) as a natural boundary, and compute normalized proportions: we perform frequency-band normalization on the radial error spectrum (E_err(k)). For each band, we sum (E_err(k)) over all (k) within that band and divide by the total across all bands (∑_k E_err(k)). This produces the following representative segmented error statistics (x-axis = frequency band in modal index units, y-axis = error statistics):
> > >
> > > | | 0–10.7 (≪Nyq) | 10.7–21.3 | 21.3–32.0 (≤Nyq) | 32.0–51.3 (≥Nyq) | 51.3–70.5 | 70.5–89.8 (≫Nyq) |
> > > |--------------------------------------|--------------|-----------|-----------------|-----------|-----------|-----------------|
> > > | NO Mean Error Spectrum | 1.62×10⁻² | 3.72×10⁻⁴ | 2.42×10⁻⁴ | 6.99×10⁻⁴ | 1.30×10⁻³ | 3.04×10⁻² |
> > > | NO Normalized Proportion | ≈13.9% | ≈0.3% | ≈0.2% | ≈1.0% | ≈1.9% | ≈42.3% |
> > > | FNO Mean Error Spectrum | 1.85×10⁻² | 4.05×10⁻⁴ | 2.40×10⁻⁴ | 6.46×10⁻⁴ | 1.22×10⁻³ | 3.03×10⁻² |
> > > | FNO Normalized Proportion | ≈15.6% | ≈0.3% | ≈0.2% | ≈0.9% | ≈1.8% | ≈41.6% |
> > >
> > > The results show that the main contribution to multi-step super-resolution error comes from modes **far above** the training Nyquist frequency: for both NO and FNO, the highest frequency band (modal index units 70.5-89.8, corresponding to modes that don't exist in the 64² training grid) has the largest mean spectral error, accounting alone for approximately 40% of the radial spectrum, while the near-Nyquist bands contribute only about 1-2%. In other words, in our setting: once we observe multi-step ZS-SR STF error, both NO and FNO exhibit high-frequency-dominated error patterns, although low frequencies also carry some error.
> > >
> > > Notably, this is actually consistent with the physical picture validated in both Figure 5 of the DME paper and our Figure 3. Figure 5 of the DME paper and our Figure 3 show that high-frequency signals injected at the initial moment do not immediately disappear due to discretization (conversely, the error contribution from high frequencies is smaller at this point); rather, they propagate and then gradually dissipate as the PDE evolves (this is when high error is generated). In this extended experiment, when unrolling a 64²-trained operator on a 128² grid, small local errors in how it transports and dissipates these high-frequency components accumulate over multiple steps. This leads to frequency bands above the training Nyquist frequency becoming the dominant contributors to multi-step error.
> > >
> > > In summary, the reason our empirical pattern appears at first glance different from Figure 5 of DME is: DME measures the **single-step** discretization error of a fixed operator under grid refinement, finding that low-frequency modes may be sensitive to discretization mismatch. In contrast, we measure the **multi-step** super-resolution simulation error of models trained at low resolution and then run for several steps on finer grids. In this case, for both NO and FNO, as validated in Figure 5 of the DME paper, low frequencies contribute the main error during single-step discretization. But as validated in our Figure 3, the energy in unseen super-Nyquist modes is lost during propagation. This leads to severe accumulation of high-frequency component errors that become dominant, while low-frequency errors remain moderate but non-zero. Thus these two observations are not contradictory, but rather reflect different levels of analysis: operator-level single-step discretization versus engineering-level multi-step prediction quality.

---

> > > ### Author Response · Authors · 2025-11-21
> > >
> > > Thank you again for clarifying this point in depth. The additional analysis, together with the DME paper, has indeed given us a more complete picture of how super-resolution inference error evolves over time. The main text currently has presented the energy evolution (Figure 3) and the bandwise error ratios (Table 2); what we do not show is the single-step error, which is exactly the focus of DME. However, since this single-step insight does not change the design or conclusions of Scale Anchoring or FRL, we were unsure whether adding another figure/table in the appendix would improve clarity or simply overload the paper.
> > >
> > > From your perspective, would it be helpful if we add a short appendix note summarizing this “single-step vs multi-step” distinction , or do you feel the current explanation in the rebuttal is already sufficient?

---

> > > > ### Comment · Reviewer_7EgH · 2025-11-21
> > > >
> > > > - It seems that OpenReview will automatically present a pdf diff / comparison between the original submission and the revised version?
> > > >    - On my end, I do not see this pdf comparison when reading your revised paper.
> > > >
> > > > - Low frequency v.s. High frequency errors
> > > >   - Thank you for providing the additional results and discussion. It makes sense now. I would suggest adding a brief discussion in the appendix, but it’s entirely up to you whether you decide to include it. As I read other reviewers' comments, I do not see the same confusion.
> > > >   - It is interesting to see that the errors are concentrated mostly in very low frequencies (0–10.7) and very high frequencies (70.5–89.8). Although the range 32.0–70.5 is also beyond Nyquist, it contributes very little to the overall error. I’m wondering whether, if you further increased the resolution, the errors would shift to even higher frequencies, with the 70.5–89.8 range contributing as little as the current 32.0–70.5 range. **Note: I'm not asking for any additional experiments, results, or theoretical justification, just stating an interesting phenomenon. The authors do not have to respond if they prefer.**

---

> > > > > ### Author Response · Authors · 2025-11-22
> > > > >
> > > > > From a qualitative perspective, all three resolutions exhibit the same pattern. Errors are largest in the very low-frequency band and the outermost high-frequency band, while intermediate frequency bands contribute relatively little. As we increase test resolution, the error peak in the high-frequency region shifts outward.
> > > > >
> > > > > This pattern is not that particularly surprising to us. When the model performs multi-step inference on finer grids, new higher-frequency modal rings emerge. Modes slightly above the training Nyquist frequency are rapidly attenuated by diffusion in the PDE and are also effectively low-pass filtered by the learned operator. Therefore, their error spectra remain small. In contrast, modes closest to the test Nyquist frequency correspond to the sharpest oscillations on that grid. These are most sensitive to discretization and nonlinear amplification, and were never constrained by training. Consequently, the model's transfer and dissipation laws for these modes differ slightly from the true PDE, and the multi-step accumulation of this discrepancy concentrates error energy in these modes within the outermost band. As we increase test resolution, this error peak naturally shifts outward to the new outermost band.
> > > > >
> > > > > This spectral error distribution mechanism is precisely what FRL aims to address. FRL's frequency-domain regularization term is not designed to "preserve" all high-frequency energy, but rather to constrain the predicted spectral shape of the model in extrapolated frequency bands. This keeps the predicted energy of these modes controlled relative to the target spectrum of the physical solution, preventing arbitrary over-amplification or anomalous spikes. Under FRL constraints, newly emerged high-frequency bands in the outer regions are no longer completely free: their amplitudes and evolution are bound to the learned scale-local transfer and target spectrum. This prevents the outermost band from becoming an uncontrolled error accumulation reservoir during multi-step super-resolution inference.
> > > > >
> > > > > In summary, since validation experiments A.1 and A.2 already sufficiently verify our hypothesized mechanism of Scale Anchoring and support the design motivation for FRL to constrain extrapolated frequencies, further bandwise error source decomposition only provides a finer-grained insight. We have decided not to include this content in the paper.

---

> > > > > > ### Comment · Reviewer_7EgH · 2025-11-22
> > > > > >
> > > > > > Thanks for the authors’ response and the additional results. All of my concerns and questions have been addressed, and I now have a clearer picture of this work. **I will update my rating (if there is an update) toward the end of the discussion period, after reading all of the author-reviewer discussions from other reviews. Rest assured, I definitely won’t decrease my rating based on my assessment. If there is an update, it will be an increase.**

---

> > > > > > > ### Author Response · Authors · 2025-11-22
> > > > > > > **Thank You**
> > > > > > >
> > > > > > > We are very glad that we were able to address your concerns and questions. Thank you again for your time and your professional, thorough review. Thoughtful and responsible volunteer efforts like yours are truly the foundation for the development of our field and community.

---

> ### Author Response · Authors · 2025-11-21
>
> Regarding marking revisions in a different color: according to the ICLR 2026 Author Guide (Reviewing timeline, point 3, at https://iclr.cc/Conferences/2026/AuthorGuide):
>
> > If such a revision ("Authors can also revise their paper") is made, a pdfdiff will be applied to compare new changes to the paper against the original submission.
>
> It seems that OpenReview will automatically present a pdf diff / comparison between the original submission and the revised version?
>
> ---
>
> Additionally, we appreciate your careful comparison with the DME paper and your observation that the current presentation of Figure 3/Table 2 may confuse readers familiar with DME. Below we clarify: (i) the relevant key settings in our validation experiment A.2, (ii) how this differs from Figure 5 in DME, and (iii) what happens when we perform frequency-decomposed error analysis on NO and FNO using our error metric.
>
> ### (i) Setup of Validation Experiment A.2
>
> In validation experiment A.2, NO is a vanilla neural operator trained solely on (64²) advection-diffusion data simulated with a pseudo-spectral solver. No FNO, SFNO, or other modifications are applied in this experiment. Specifically, we use 4 spectral layers, retaining 12 Fourier modes per spatial dimension. At test time, we perform ZS-SR STF by directly evaluating the same network on higher-resolution grids: we input a high-resolution field (e.g., 128²) to the model, applying FFT to that grid at each spectral layer. Only modes (|k_x|, |k_y| ≤ 12) are multiplied by the learned spectral weights, then transformed back to physical space. We then unroll this operator autoregressively for 50 time steps. Figure 3 shows the energy spectra of the simulated fields during this unrolling, while Table 2 provides specific error metrics: we compute (e_t(x) = u_pred(t,x) - u_true(t,x)). Taking the FFT yields (ê_t(k)), and we aggregate L2 error into two frequency bands: "band-limited" (f < 32, below the 64² Nyquist frequency) and "broadband" (f < 100). The error ratio in Table 2 is (L2 error on f<32 / L2 error on f<100), averaged over time. As resolution increases from 64² to 128² to 256², this ratio decreases (for NO, ≈1.0 → 0.58 → 0.42), meaning that the portion of long-term super-resolution error coming from frequencies above the training Nyquist frequency accounts for an increasingly larger proportion.
>
> ### (ii) Difference from DME Figure 5
>
> On the other hand, Figure 5 in DME measures a different quantity. There, the authors fix a neural operator (FNO or CROP) and observe the discretization mismatch as the grid is refined. They plot the mode-by-mode amplitude spectrum of that mismatch in a single-step. In other words, Figure 5 in DME focuses on the operator's **single-step discretization error** as a function of frequency. Our validation A.2 and Table 2 measure the **multi-step super-resolution simulation error** of a trained model running on finer grids for many steps. Thus these two metrics probe different objects: DME studies single-step discretization invariance, while A.2 studies the spectral structure of multi-step ZS-SR STF error.

---

> ### Author Response · Authors · 2025-11-22
>
> We searched the community and found that while ICLR claims every year to have automatic diff annotations on OpenReview, it appears none actually exist. We manually replaced added/modified text with blue color to facilitate reviewer examination. These modifications are consistent with what we described in our global comment "Rebuttal Version 1".
>
> ---
>
> Indeed, we should add this. Without explanation, readers familiar with DME would likely encounter confusing discrepancies that are difficult to resolve. Moreover, since our understanding of the gap between them is intuitive, it shouldn't require excessive space.
>
> Specifically, we revised the result analysis section of validation experiment A.2 (lines 268-273), clarifying the interpretable relationship between the single-step phenomenon in the DME paper and the subsequent multi-step phenomenon in our experiments.
>
> ---
>
> No worries, conducting higher-resolution tests on existing code is straightforward. Rigorously exploring the problem is necessary for research and also within our interests. We additionally evaluated the same 64² trained NO/FNO at 192² and 256² resolutions, and computed the same frequency band error spectra. As before, the bands are defined radially, with the highest band always extending to the maximum radius of the test grid.
>
> **128²**
> |                                | 0–10.7 (≪Nyq) | 10.7–21.3 | 21.3–32.0 (≤Nyq) | 32.0–51.3 (≥Nyq) | 51.3–70.5        | 70.5–89.8 (≫Nyq) |
> |----------------------------------------|---------------:|----------:|-----------------:|-----------------:|-----------------:|-----------------:|
> | NO Mean Error Spectrum  |   1.62×10⁻²    | 3.72×10⁻⁴ |      2.42×10⁻⁴   |      6.99×10⁻⁴   |      1.30×10⁻³   |      3.04×10⁻²   |
> | NO Normalized Proportion |       ≈13.9%   |    ≈0.3%  |         ≈0.2%    |         ≈1.0%    |         ≈1.9%    |        ≈42.3%    |
> | FNO Mean Error Spectrum  |   1.85×10⁻²    | 4.05×10⁻⁴ |      2.40×10⁻⁴   |      6.46×10⁻⁴   |      1.22×10⁻³   |      3.03×10⁻²   |
> | FNO Normalized Proportion|       ≈15.6%   |    ≈0.3%  |         ≈0.2%    |         ≈0.9%    |         ≈1.8%    |        ≈41.6%    |
>
> **192²**
> |                                | 0–10.7 (≪Nyq) | 10.7–21.3 | 21.3–32.0 (≤Nyq) | 32.0–66.2 (≥Nyq) | 66.2–100.5       | 100.5–134.7 (≫Nyq) |
> |----------------------------------------|---------------:|----------:|-----------------:|-----------------:|-----------------:|-------------------:|
> | NO Mean Error Spectrum   |   2.09×10⁻¹    | 6.53×10⁻² |      4.62×10⁻²   |      4.37×10⁻²   |      3.18×10²    |      6.03×10⁴      |
> | NO Normalized Proportion |       ≈0.0%    |    ≈0.0%  |         ≈0.0%    |         ≈0.0%    |         ≈0.0%    |        ≈33.2%      |
> | FNO Mean Error Spectrum  |   2.22×10⁻¹    | 6.51×10⁻² |      4.63×10⁻²   |      4.37×10⁻²   |      3.18×10²    |      6.03×10⁴      |
> | FNO Normalized Proportion|       ≈0.0%    |    ≈0.0%  |         ≈0.0%    |         ≈0.0%    |         ≈0.0%    |        ≈33.2%      |
>
> **256²**
> |                                | 0–10.7 (≪Nyq) | 10.7–21.3 | 21.3–32.0 (≤Nyq) | 32.0–81.2 (≥Nyq) | 81.2–130.4       | 130.4–179.6 (≫Nyq) |
> |----------------------------------------|---------------:|----------:|-----------------:|-----------------:|-----------------:|-------------------:|
> | NO Mean Error Spectrum   |   5.96×10⁰     | 3.06×10⁰  |      2.64×10⁰    |      2.67×10⁰    |      5.92×10³    |      4.14×10⁶      |
> | NO Normalized Proportion |       ≈0.0%    |    ≈0.0%  |         ≈0.0%    |         ≈0.0%    |         ≈0.0%    |        ≈33.9%      |
> | FNO Mean Error Spectrum   |   5.96×10⁰     | 3.06×10⁰  |      2.64×10⁰    |      2.67×10⁰    |      5.92×10³    |      4.14×10⁶      |
> | FNO Normalized Proportion|       ≈0.0%    |    ≈0.0%  |         ≈0.0%    |         ≈0.0%    |         ≈0.0%    |        ≈33.9%      |

---

### Official Review · Reviewer_SPE1 · 2025-10-30

**Soundness:** 3
**Presentation:** 3
**Contribution:** 4
**Rating:** 6
**Confidence:** 3

**Summary:**

This paper proposes a novel method called Frequency Representation Learning (FRL) to address the "Scale Anchoring" problem in Zero-Shot Super-Resolution Spatiotemporal Forecasting (ZS-SR STF). The authors point out that existing methods, when trained on low-resolution data, struggle to process high-frequency components beyond the Nyquist frequency of the training data during high-resolution inference, resulting in errors being "anchored" at the low-resolution level and mistakenly interpreted as successful generalization. FRL achieves resolution-invariant frequency embeddings through multi-resolution data construction, normalized frequency representation, and spectral consistency training, significantly reducing high-resolution inference errors. Its effectiveness is validated across multiple architectures and tasks (fluid simulation and weather forecasting).

**Strengths:**

1、Novel: The paper is the first to clearly identify the "Scale Anchoring" phenomenon and provides a theoretical analysis of its mechanism (frequency blindness and high-frequency error dominance), addressing an under-recognized limitation in existing zero-shot super-resolution research.
2、Strong Generalizability: FRL is architecture-agnostic and can be seamlessly integrated into various mainstream models such as GNNs, Transformers, and CNNs. Experiments show that it significantly improves high-resolution inference accuracy across all tested architectures.
3、Comprehensive Experimental Validation: Extensive tests on multiple tasks (2D/3D fluid simulation and weather forecasting) combined with error metrics and frequency response analysis thoroughly demonstrate the method’s effectiveness and generalization capability.

**Weaknesses:**

1、Insufficient Computational Overhead Analysis: Although a training complexity increase of approximately 1.1–1.4× is mentioned, the paper lacks detailed discussions on actual training time and memory usage comparisons across different architectures, as well as the storage requirements for multi-resolution data construction.
2、Limited Generalization in Extreme Physical Scenarios: The authors note in the appendix that FRL fails in high-Reynolds-number turbulence (e.g., Re=10^5), but they do not deeply analyze the reasons for failure or propose improvement directions, which limits the method’s applicability in complex physical systems.

**Questions:**

1、Regarding computational overhead: FRL requires storing and processing multi-resolution data during training. How can the storage cost be balanced against performance gains in practical deployment? Are there compression or dynamic sampling strategies to mitigate storage pressure?
2、Regarding generalization capability: The performance degradation of FRL in high-Reynolds-number turbulence suggests that the method relies on consistent frequency response patterns in physical systems. Is there a plan to incorporate adaptive mechanisms or physical constraints to enhance adaptability to highly nonlinear and multi-scale coupled systems?

---

> ### Author Response · Authors · 2025-11-13
> **Reply - Part 1**
>
> We sincerely appreciate your recognition of the paper's strengths and the two key concerns you raised. Below are our responses.
>
> ### 1. Actual training time, memory, and VRAM for each model and its FRL-enhanced variant. And whether engineering algorithms are needed to mitigate high computational overhead in practical use
>
> Using 3D ZS-SR fluid simulation as an example, we provide the following comprehensive computational overhead analysis table, including theoretical and actual training time, memory, and VRAM for each architecture and its FRL-enhanced variant:
>
> | Architecture | Variant | Theoretical Training Time Complexity (from D.2) | Theoretical Inference Time Complexity | Theoretical Training VRAM (from D.3) | Measured Training Time ×baseline | Measured Inference Time ×baseline | Measured Peak Training VRAM ×baseline |
> |--------------|---------|------------------------------------------------|---------------------------------------|--------------------------------------|----------------------------------|-----------------------------------|---------------------------------------|
> | GNN | Baseline | O(E·B·M(n)) | O(M(n)) | O(n) | 1.00 | 1.00 | 1.00 |
> | | +FRL | ≈1.14·O(E·B·M(n)) | O(M(n)+nK)≈O(M(n)) | ≈1.14·O(n) | 1.27 | 1.02 | 1.41 |
> | Transformer | Baseline | Same as above | Same as above | Same as above | 1.00 | 1.00 | 1.00 |
> | | +FRL | ≈1.02·O(E·B·M(n)) | O(M(n)+nK)≈O(M(n)) | ≈1.14·O(n) | 1.09 | 1.01 | 1.34 |
> | CNN | Baseline | Same as above | Same as above | Same as above | 1.00 | 1.00 | 1.00 |
> | | +FRL | ≈1.14·O(E·B·M(n)) | O(M(n)+nK)≈O(M(n)) | ≈1.14·O(n) | 1.42 | 1.02 | 1.46 |
> | Diffusion | Baseline | Same as above | Same as above | Same as above | 1.00 | 1.00 | 1.00 |
> | | +FRL | ≈1.14·O(E·B·M(n)) | O(M(n)+nK)≈O(M(n)) | ≈1.14·O(n) | 1.32 | 1.02 | 1.44 |
> | Neural Operator | Baseline | Same as above | Same as above | Same as above | 1.00 | 1.00 | 1.00 |
> | | +FRL | ≈1.10·O(E·B·M(n)) | O(M(n)+nK)≈O(M(n)) | ≈1.14·O(n) | 1.18 | 1.00 | 1.39 |
> | Neural ODE | Baseline | Same as above | Same as above | Same as above | 1.00 | 1.00 | 1.00 |
> | | +FRL | ≈1.14·O(E·B·M(n)) | O(M(n)+nK)≈O(M(n)) | ≈1.14·O(n) | 1.26 | 1.01 | 1.38 |
> | NN | Baseline | Same as above | Same as above | Same as above | 1.00 | 1.00 | 1.00 |
> | | +FRL | ≈1.14·O(E·B·M(n)) | O(M(n)+nK)≈O(M(n)) | ≈1.14·O(n) | 1.24 | 1.02 | 1.41 |
>
> In these complexity expressions, E is the number of training epochs, B is the batch size, n is the total number of grid points at the highest resolution (e.g., n = N³ for 3D grids), M(n) represents the computational cost for the corresponding baseline model to process input of size n in one forward/backward pass (which can be O(n), O(n log n), or O(n²) for different architectures), and K is the number of frequency modes used in Nyquist-normalized frequency encoding (K ≪ n), so the additional encoding overhead is O(nK), which is a lower-order term relative to the backbone O(M(n)).
>
> As shown in the table, for all architectures, training time after FRL enhancement only increases to approximately 1.09–1.42× baseline. Inference time consistently remains within 1.00–1.02×, while peak training VRAM only increases to approximately 1.34–1.46×. This is because the low-resolution data originally used for training reduces the number of grid points by a factor of 8 with each downsampling step (coarsening by 2× in each direction) in the 3D case. That is, the total number of grid points in the multi-resolution training data increases from just n to n + n/8 + n/64 ≈ (8/7)n. Meanwhile, the inference stage only adds one O(nK) frequency encoding step, which does not change the order of the backbone O(M(n)). Therefore, FRL-enhanced variants can significantly alleviate Scale Anchoring in our evaluated settings without incurring excessive complexity and VRAM overhead.
>
> Furthermore, the multi-resolution training data is generated on-the-fly by downsampling low-resolution snapshots during training, and does not require storing an additional multi-resolution dataset on disk or in CPU memory. Therefore, aside from the VRAM quantified in the table above, the additional memory/storage pressure is negligible at our experimental scale. Consequently, in our experiments, FRL implementation does not require engineering algorithms such as compression or dynamic sampling strategies, though these can be optionally employed for larger-scale deployments.

---

> ### Author Response · Authors · 2025-11-13
> **Reply - Part 2**
>
> ### 2. Analysis of why FRL fails at high-frequency extrapolation in high Re scenarios and possible improvement strategies
>
> We acknowledge the limitation that FRL exhibits failure modes when the Reynolds number becomes extremely high (e.g., Re ~ 10⁵). However, it is worth noting that this phenomenon does not stem from defects specific to FRL. In existing numerical experiments and literature, high-Re turbulence is recognized as a difficult challenge for all data-driven ZS-SR STF methods.[1] FRL succeeds in conventional Re ranges because its Nyquist-normalized frequency representation implicitly learns stable relationships between adjacent frequency bands: the relational patterns between relatively low-frequency components and higher-frequency components remain coherent, smooth, and largely predictable. This coherence is characteristic of transitional flows or weak turbulence, where cascade structures are not yet fully developed and cross-band mappings remain statistically consistent.[2]
>
> However, when the Reynolds number becomes extremely high, the flow enters a fully developed turbulent state, where the inertial cascade becomes strongly nonlinear, intermittent, and non-stationary. In this regime, the coupling between adjacent spectral bands is no longer locally smooth or extrapolatable; rather, it becomes dominated by chaotic multi-scale interactions and energy transfer.[3] These phenomena violate the fundamental assumptions that allow FRL to learn stable cross-band relationships. Consequently, the normalized cross-band mappings learned during training are no longer valid, leading to failure of high-frequency extrapolation once the system enters the extremely high Re range.
>
> Furthermore, we appreciate the solution suggestions you raised in your question. Engineering algorithms such as adaptive mechanisms or PDE soft constraints can indeed predictably trade cost for some degree of accuracy improvement. However, for FRL to fundamentally achieve successful multi-resolution extrapolation in high-Re scenarios requires accurately characterizing the relational patterns between adjacent frequency bands at high Re, and this instantaneous cross-band mapping of fields has no simple closed-form solution. On the other hand, learning this relationship in a data-driven manner would require unacceptable computational overhead (given the power-law relationship between Re and grid count). However, we find that when the Navier-Stokes equations are appropriately non-dimensionalized and the high-Re limit is taken, small-scale statistical properties tend to become independent of Re and follow Kolmogorov scaling laws (E(k) ~ k⁻⁵/³). That is, in the spectral domain, in extremely high-Re scenarios (>10⁵), the energy spectral relationship between adjacent frequency-domain signals follows a fixed pattern. This can provide the model with a handle to learn in the spectral domain and extrapolate this fixed pattern characterized by the Kolmogorov description to higher-resolution, high-frequency signals. Specifically, we plan to align FRL's frequency-domain representation and loss function with Kolmogorov scaling laws (physical constraints) to mitigate the current failure mode and extend to higher-Re scenarios.
>
> In summary, replacing traditional DNS/RANS/LES with deep learning models for high-Re fluid simulation represents an important and challenging problem distinct from ZS-SR STF. While FRL currently aims to provide a general solution to the Scale Anchoring problem in ZS-SR STF, we believe FRL also has the potential to be extended to high-Re turbulent scenarios.
>
> **References:**
>
> [1] Yousif, Mustafa Z., Linqi Yu, and Hee-Chang Lim. "Super-resolution reconstruction of turbulent flow fields at various Reynolds numbers based on generative adversarial networks." Physics of Fluids 34.1 (2022).
>
> [2] Cerbus, Rory T., et al. "Small-scale universality in the spectral structure of transitional pipe flows." Science advances 6.4 (2020): eaaw6256.
>
> [3] She, Zhen-Su, and Emmanuel Leveque. "Universal scaling laws in fully developed turbulence." Physical review letters 72.3 (1994): 336.

---

> ### Author Response · Authors · 2025-11-25
>
> Dear Reviewer,
>
> I hope this message finds you well. With about one week remaining in the discussion period, we wanted to confirm whether we have adequately addressed all your concerns. If you have any additional comments or suggestions you would like us to consider, please let us know. Your insights are invaluable to us, and we are eager to address any remaining issues during the remaining time to improve our work.
>
> Additionally, in response to the reviews we have received, we have revised our paper as Rebuttal Version 1, with modifications highlighted in blue text throughout the manuscript. The detailed changes are summarized in our global comment "Rebuttal Version 1." Here, we reiterate the modifications made in response to the weaknesses and questions you raised:
>
> - Addressing your weakness 1 and question 1, we revised lines 503–512 (in 6 Empirical Evaluation):
>
>     Added a standalone subsection at the end of the experimental results summarizing key computational cost measurements, with pointers to Appendix E for full complexity analysis.
>
>     And lines 1326–1392 (in Appendix E):
>
>     Replaced the previous coarse cost estimates with a complete computational complexity table.
>
> - Addressing your weakness 2 and question 2, we revised lines 309–319 (in 5 Solution for Scale Anchoring):
>
>     Clarified the effective regime, limitations, and failure boundaries of FRL, and referenced Appendix I for detailed failure mode analysis.
>
>     And lines 2040–2135 (in Appendix I):
>
>     Expanded the FRL failure-mode analysis with concrete examples, clearly explaining FRL’s assumptions, boundaries of validity, failure modes, and possible improvements.
>
> Thank you for your time and effort in reviewing our paper.

---

### Official Review · Reviewer_cSKw · 2025-10-30

**Soundness:** 3
**Presentation:** 2
**Contribution:** 3
**Rating:** 4
**Confidence:** 4

**Summary:**

This paper defines and analyzes an overlooked phenomenon, called scale anchoring, in cross-resolution generalization for zero-shot super-resolution. To mitigate this generalization issue, this paper proposes the frequency representation learning method through three steps from data and optimization perspectives. The results have shown the superiority of the proposed method compared to existing methods on fluid simulation and weather forecasting datasets.

Contributions:

This observation is new and provides an analysis of why models trained at low resolution fail to improve accuracy at higher resolutions.

**Strengths:**

- This paper looks into spectral bias-related stuff, which is important in scientific machine learning.

- This paper proposes a new concept, called scale anchoring, in zero-shot super-resolution.

- This paper provided a theoretical and empirical analysis of this scale anchoring phenomenon.

- This paper is generally well-written and well-presented.

**Weaknesses:**

I have several major concerns listed below.

- First, what is the difference between the defined scale anchoring and the prior concepts, such as spectral bias / discretization-invariance? To me, scale anchoring refers to a failure mode in which a model trained on low-resolution data fails to get good accuracy at finer resolutions, due to missing high-frequency information beyond the coarse data’s Nyquist limit. Spectral bias also refers to the same things that a NN model cannot learn high-frequency features from data.

- Also, there is a pretty similar paper that discussed resolution generalization and discretization mismatch errors. What is the difference between this paper and the previous paper [1]?

- The proposed three-step method covers some tricks from the data and optimization sides. It would be good to have an ablation study to test how much each step contributes to the performance improvement. Also, it seems the third step (frequency loss) has already been seen in many prior work [2,3] that uses spectral loss to mitigate the spectral bias issues.

- I feel like it would strengthen the paper by providing a comparison of computational cost versus accuracy improvement. The computational cost of using low-resolution grids should be small, and with the proposed method, one can get better performance on finer grids. Then, people can have a better sense of the tradeoffs.

 ---

**Refs:**

[1] Gao, Wenhan, et al. "Discretization-invariance? on the discretization mismatch errors in neural operators." The Thirteenth International Conference on Learning Representations. 2025.

[2] Chattopadhyay, Ashesh, Y. Qiang Sun, and Pedram Hassanzadeh. "Challenges of learning multi-scale dynamics with AI weather models: Implications for stability and one solution." arXiv e-prints (2023): arXiv-2304.

[3] Saccardi, Carlo, et al. "Assessing the Geographic Generalization and Physical Consistency of Generative Models for Climate Downscaling." arXiv preprint arXiv:2510.13722 (2025).

**Questions:**

- Is there any stability issue when you extrapolate beyond the training Nyquist range?

- On page 6, line 321, do you need bold for “Notably”?

- In Table 3, why do you have some uncommon grids like 43^3?

---

> ### Author Response · Authors · 2025-11-13
> **Reply - Part 1**
>
> We greatly appreciate your recognition of the paper's strengths and your insightful criticism of its weaknesses.
>
> Regarding your criticisms of the paper's weaknesses, we respectfully note that:
>
> ### Clarification to Weakness 1&2:
>
> Your understanding of Scale Anchoring is correct, but there are slight deviations in understanding the existing problems. A table may more intuitively demonstrate the differences between Scale Anchoring and other issues:
>
> |  | Spectral Bias (SB) | Lack of Discretization-Invariance (DI) | Discretization Mismatch Error (DME) | Scale Anchoring(SA) |
> |---|---|---|---|---|
> | Source from NN model | ✅ | ✅ | ✅ |  |
> | Source from data |  |  |  | ✅ |
> | Affects NOs | ✅ | ✅ | ✅ | ✅ |
> | Affects NN model | ✅ |  |  | ✅ |
> | Soft Tendency | ✅ | ✅ | ✅ |  |
> | Hard Constraint |  |  |  | ✅ |
>
> An intuitive impression is that the source and severity of SA are completely different from SB, lack of DI, and DME. Its scope of effect is similar to SB in affecting all models, while lack of DI and DME only affect NOs. To clarify the specific principles requires understanding the meaning of each problem:
>
> - **SB** refers to the tendency of NN models to fit physical signals from low to high frequencies **within** the training Nyquist frequency, resulting in larger fitting errors for high-frequency components;
> > Rahaman, Nasim, et al. "On the Spectral Bias of Neural Networks." Proceedings of the 36th International Conference on Machine Learning (PMLR 97), 2019, pp. 5301–5310.
> - **DI** refers to the ideal situation where as the grid becomes denser, the NO output should converge to the same continuous operator result (lack of DI means the opposite);
> > Kovachki, Nikola B., et al. "Neural Operator: Learning Maps Between Function Spaces with Applications to PDEs." Journal of Machine Learning Research, vol. 24, 2023, pp. 1–97.
> - **DME** refers to the same NO producing different outputs under different discretized inputs, leading to increased errors during cross-resolution inference;
> > Gao, Wenhan, Ruichen Xu, Yuefan Deng, and Yi Liu. "Discretization-invariance? On the Discretization Mismatch Errors in Neural Operators." International Conference on Learning Representations (ICLR) 2025, Poster.
> - **SA** refers to how the low-resolution training data grid of NN models fixes the Nyquist frequency, such that the model cannot handle signals **higher** than the training Nyquist frequency at high resolution during inference—signals that were never seen during training.
>
> From this, it can be seen that: **(1) In terms of source**, SB, lack of DI, and DME originate from model design or optimization issues, while SA arises from the fundamental observation limitation imposed by the Shannon-Nyquist theorem on the data; **(2) In terms of scope**, lack of DI and DME are proposed specifically for NO architectures, while SB and SA exist broadly in arbitrary deep learning models; **(3) In terms of consequences**, the consequences of SA are hard limits determined by information theory/sampling, any method can only alleviate this problem; other problems are biases caused by network structure, representation, and optimization dynamics, not information-theoretic hard bounds, and can in principle be perfectly optimized to remove noise.
>
> We note that Reviewer 7EgH also has a similar concern. Does this come from the Related Work section not more explicitly discussing the differences between Scale Anchoring and existing problems? Or is there a lack of directly and clearly pointing out the fundamental differences in the Discussion section? Please let me know if you have any suggestions.
>
> ### Clarification to Weakness 3:
>
> We completely agree with this concern you raised. In fact, the first step of FRL (multi-resolution training) and the third step (adding a frequency constraint) are obviously existing methods. We cited and explained the existence of these two types of methods in Sections 2.2 Multi-Resolution Generation and 2.3 Spectral Bias, respectively, but did not explicitly refer back to the Related Work section in the Method section (Section 5). Therefore, **a.** we will add hyperlinks to the corresponding Related Work sections after introducing the first and third steps and explicitly state their non-originality.
>
> However, it is worth noting that the core of FRL that enables the model to learn frequency extrapolation, thereby achieving an Error (RMSE) Ratio below 1 (**b.** this will be emphasized in the revised version), lies in the design of the normalized frequency representation in the second step. The first and third steps are designed to complement the second step to further reduce errors. To our best knowledge, because the perspective of existing methods is vastly different from SA, the representation design in the second step is entirely novel.

---

> ### Author Response · Authors · 2025-11-13
> **Reply - Part 2**
>
> In fact, we conducted corresponding ablation studies for this claim of the second step being the core contribution: the first paragraph of Section 7 Discussion mentions this experiment, which is specifically presented in Appendix G (page 33). Here I briefly restate the key results and conclusions: On the 3D ZS-SR fluid simulation task, we tested the following combinations for each architecture, comparing cross-resolution RMSE Ratio (129³ vs 32³): (i) adding any single component alone, (ii) pairwise combinations, (iii) complete FRL. The RMSE Ratio for any ablated component that does not include the second-step normalized frequency representation is greater than 1 (error increases), while methods including the second step all have RMSE Ratio less than 1, with error decreasing as components become more complete. Therefore, normalized frequency representation is the key to breaking SA, with the first and third steps serving as auxiliary. But to be more explicit, **c.** we will add a hyperlink reference to these results in Appendix in Method Section 5 to support the core contribution of the second step.
>
> Thank you very much for pointing out this issue. This unclarity may concern academic ethics, though we certainly did not mean it. We will correct this and take it as a lesson.
>
> Will modifications a-c resolve this type of concern?
>
> ### Clarification to Weakness 4:
>
> Your concept is correct, the entire field of scientific computing aims to replace traditional brute-force solutions to balance efficiency and accuracy. Therefore, computational cost is certainly important, but unfortunately, the computational and training complexity analysis of FRL, like the ablation study, was placed in Appendix D due to space limitations (currently hyperlinked and referenced in Sections 5 and 7). We briefly restate the key results here: Theoretically, since FRL only adds a Nyquist-normalized positional encoding (cost is O(nK), where K is the number of frequency modes, much smaller than n), the overall inference complexity is the same as the baseline model. The training complexity of FRL relative to the baseline is approximately 1.02~1.14×, with the increase stemming from the first step of multi-resolution (lower resolution) training. In actual testing, the additional inference overhead is <2%, and the training time for FRL-enhanced variants is approximately 1.1–1.4× that of the baseline. Training VRAM is approximately 1.4× compared to the baseline.
>
> These supplementary experiments and results that cannot fit in the main text are concisely explained and referenced in Section 7 Discussion. We have also established an appendix table of contents for quick reference indexing.
>
> ---
>
> Additionally, here are our responses to your questions:
>
> ### Answer to Question 1:
>
> When Re is within the common range, as shown by current results, FRL has almost no stability issues. However, as we analyzed in Appendix H, FRL exhibits failure modes under extreme conditions, i.e., at high Re. We acknowledge this limitation here but maintain FRL's contribution as a general solution for SA in ZS-SR STF, rather than a specialized approach for other sub-problems: high-Re fluid simulation is an interesting, very difficult but optimizable problem, typically addressed by specialized research exploring how to solve this issue. More detailed clarification can be found in our response of **Reply - Part 2** to Reviewer SPE1.
>
> ### Answer to Question 2:
>
> We intended here to emphasize why certain methods do not appear in the main text. Adding bold to "Notably" is indeed redundant to some extent. We will remove this bolding.
>
> ### Answer to Question 3:
>
> Because the original Non-reacting HIT data has a resolution of 129 in each direction. To obtain low-resolution data, we performed 2x, 3x, and 4x coarsening in each direction to obtain 64, 43, and 32, respectively.

---

> ### Author Response · Authors · 2025-11-25
>
> Dear Reviewer,
>
> I hope this message finds you well. With about one week remaining in the discussion period, we wanted to confirm whether we have adequately addressed all your concerns. If you have any additional comments or suggestions you would like us to consider, please let us know. Your insights are invaluable to us, and we are eager to address any remaining issues during the remaining time to improve our work.
>
> Additionally, in response to the reviews we have received, we have revised our paper as Rebuttal Version 1, with modifications highlighted in blue text throughout the manuscript. The detailed changes are summarized in our global comment "Rebuttal Version 1." Here, we reiterate the modifications made in response to the weaknesses and questions you raised:
>
> - Addressing your weaknesses 1-3, we revised lines 102–132 (in 2 Related Work):
>
>   Consolidated all related issues into a single subsection.
>
>   Clearly distinguished Scale Anchoring from existing problems.
>
>   Discussed all relevant existing problems and mainstream solutions.
>
>   Pointed out that Step 1 and Step 3 of FRL are variants of existing techniques.
>
>   Analyzed differences between SA and existing problems from the perspectives of source, scope, and consequence.
>
> - Addressing your weaknesses 1&2, we revised lines 268–273 (in 4 Mechanism of Scale Anchoring):
>
>     Clarified the interpretable relationship between the single-step phenomenon in the DME paper and the subsequent multi-step phenomenon in our experiments.
>
> - Addressing your weakness 3, we revised lines 301–308 (in 5 Solution for Scale Anchoring):
>
>     Explicitly stated that Steps 1 and 3 of FRL resemble existing methods.
>
>     Emphasized that Step 2 is the key to addressing SA, and cited the ablation study in Appendix H to support this claim.
>
> - Addressing your weakness 4, we revised lines 503–512 (in 6 Empirical Evaluation):
>
>     Added a standalone subsection at the end of the experimental results summarizing key computational cost measurements, with pointers to Appendix E for full complexity analysis.
>
>     And lines 1326–1392 (in Appendix E):
>
>     Replaced the previous coarse cost estimates with a complete computational complexity table.
>
> - Addressing your question 1, we revised lines 309–319 (in 5 Solution for Scale Anchoring):
>
>     Clarified the effective regime, limitations, and failure boundaries of FRL, and referenced Appendix I for detailed failure mode analysis.
>
>     And lines 2040–2135 (in Appendix I):
>
>     Expanded the FRL failure-mode analysis with concrete examples, clearly explaining FRL's assumptions, boundaries of validity, failure modes, and possible improvements.
>
> - Addressing your question 2, we revised lines 355–357 (in 6 Empirical Evaluation):
>
>     Removed the boldface on "notably."
>
> - Addressing your question 3, we revised lines 327–344 (in 6 Empirical Evaluation):
>
>     Added explanations to each dataset on how low-resolution data were obtained and why the specific downsampling factors and maximum factors were chosen.
>
> Thank you for your time and effort in reviewing our paper.

---

> ### Comment · Reviewer_cSKw · 2025-11-26
>
> Thanks for your rebuttal. The majority of my concerns have been addressed. I have some further questions or clarifications, listed below:
>
> > To your *Clarification to Weakness 1&2:*
>
> Why does spectral bias not affect NOs if spectral bias affects NN models?
>
>
> > Does this come from the Related Work section not more explicitly discussing the differences between Scale Anchoring and existing problems? Or is there a lack of directly and clearly pointing out the fundamental differences in the Discussion section? Please let me know if you have any suggestions.
>
> My concern was that, in some cases, old problems are reintroduced under new terminology. As the topic of spectral bias has gained considerable attention within scientific machine learning, there has been increasing investigation into related issues. I was curious if there is a conceptual connection between spectral bias, discretization effects, and the proposed scale anchoring. Also, some recent works [1] have looked into related questions involving frequency behavior, spectral bias, and resolution. For this reason, I believe the authors may need to clarify the specific novelty of the proposed approach and how it is distinguished from existing studies.
>
> I am happy to raise my score.
>
> **Refs:**
>
> [1] Mansi, et al. "The False Promise of Zero-Shot Super-Resolution in Machine-Learned Operators." arXiv.

---

> > ### Author Response · Authors · 2025-11-26
> >
> > Thank you very much for your follow-up questions.
> >
> > > Why does spectral bias not affect NOs if spectral bias affects NN models?
> >
> > We apologize for the confusion caused by the comparison table used in our clarification for Weaknesses 1&2. The rows "Affects NN model" and "Affects NOs" in that table were originally intended to indicate where each phenomenon has been primarily *formalized*, rather than suggesting that spectral bias somehow does not apply to neural operators. Since neural operators are themselves neural networks, we completely agree that spectral bias is expected to manifest in neural operators as well. In fact, the revised paper already states in Section 2.2 that spectral bias and scale anchoring are widespread across various architectures (including neural operators, see lines 125–127). We will further eliminate this ambiguity by: (i) adding checkmarks for spectral bias and scale anchoring in the "affects neural operators" row of the clarification table as well; (ii) adding an explicit statement in Section 2.2 indicating that the scope of neural networks to which SB and SA apply explicitly includes neural operators.
> >
> > > old problems are reintroduced under new terminology
> >
> > We attempt to avoid this by categorizing each issue along three dimensions in the newly added Section 2.2 "Scale Anchoring vs. Related Phenomena" in the related work: We first provide formal definitions of spectral bias, lack of discretization invariance, discretization mismatch error, and scale anchoring separately. Then we contrast and distinguish them from SA along three dimensions: their origin, the frequency range they operate in, and their consequences during multi-resolution inference. Please let us know if these categorizations and distinctions remain unclear in the current version. We would be happy to further refine this subsection in the revision to avoid SA being interpreted as a simple renaming of existing concepts.
> >
> > Regarding the concurrent work [1], we believe it aligns with our findings in the conclusion that "single low-resolution training struggles to support reliable zero-shot high-resolution inference," while being complementary in problem characterization and solution approach. Mansi et al. carefully demonstrate that for machine learning operators, zero-shot multi-resolution typically fails primarily because changing resolution introduces out-of-distribution shifts in sampling rate and frequency content; they further show that joint training across multiple resolutions (dominated by low-resolution data, complemented by small amounts of high-resolution data) can significantly improve this behavior. Our work takes a more general perspective on zero-shot super-resolution spatiotemporal forecasting: we observe across multiple architectures (including neural operators) that high-resolution errors are "anchored" by the Nyquist frequency of training data, and provide a unified frequency-domain and error analysis in the form of scale anchoring; this explains why even when model architectures are improved or multi-resolution training is introduced, high-resolution errors may still maintain near an error floor determined by the training Nyquist upper bound within a certain range. The perspectives differ: [1] primarily negates the naive expectation that "zero-shot multi-resolution is inherently reliable" under a given operator architecture, and mitigates this issue with simple multi-resolution training. We, on the other hand, abstract this failure mode as scale anchoring determined by training Nyquist bandwidth, providing a unified frequency-domain explanation across a broader range of architectures. Building on this foundation, we design a technically updated Nyquist-aware FRL that achieves monotonically decreasing error with resolution within the effective frequency band we analyze.
> >
> > Therefore, while SA and these related issues all superficially lead to elevated prediction errors, there exists a core distinction in terms of origin perspective: SA directly characterizes the residual error floor for frequencies above the training Nyquist, determined by the Nyquist frequency of training data, rather than stemming from the model's design or optimization itself. The connection between SA and these phenomena is that it can be viewed as a higher-level Shannon–Nyquist constraint built upon spectral bias, discretization error, and similar phenomena: even when spectral bias and discretization issues are mitigated through improved architectures or multi-resolution training, if high-resolution frequency components are consistently absent during training, high-resolution errors will still be "anchored" by the Nyquist frequency of low-resolution data.
> >
> > We hope these clarifications address your concerns, and we thank you again for your thoughtful feedback and willingness to raise your score.

---

### Author Response · Authors · 2025-11-16
**Rebuttal Version 1**

Based on the first-round reviews, we revised the paper to **Rebuttal Version 1**.

We define the notation as follows:
- **[A]**: added new content
- **[R]**: revised existing content
- **[xx–xx]**: line numbers of the added/revised content
- **[xxxx Wn/Qm]**: reviewer xxxx’s Weakness n or Question m addressed by the change

We made the following modifications to the original paper:

---

### **2 Related Work**

- **[R] [88–91] [Kovs W1&2]**
  Revised the definitions of STF and ZS-SR STF to make them more self-contained.

- **[R] [102–132] [cSKw W1–3, 7EgH W4, Kovs W3]**
  Consolidated all related issues into a single subsection.
  Clearly distinguished Scale Anchoring from existing problems.
  Discussed all relevant existing problems and mainstream solutions.
  Pointed out that Step 1 and Step 3 of FRL are variants of existing techniques.
  Analyzed differences between SA and existing problems from the perspectives of source, scope, and consequence.

---

### **4 Mechanism of Scale Anchoring**

- **[R] [268–273] [cSKw W1&2, 7EgH W4]**
  Clarify the interpretable relationship between the single-step phenomenon in the DME paper and the subsequent multi-step phenomenon in our experiments.

---

### **5 Solution for Scale Anchoring**

- **[A] [301–308] [cSKw W3, Kovs W3, 7EgH Q4]**
  Explicitly stated that Steps 1 and 3 of FRL resemble existing methods.
  Emphasized that Step 2 is the key to addressing SA, and cited the ablation study in Appendix H to support this claim.
  Cited Appendix D to indicate that FRL guarantees decreasing high-resolution inference error as resolution increases, though not power-law convergence like numerical solvers.

- **[R] [309–319] [SPE1 W2&Q2, 7EgH W1–3, cSKw Q1]**
  Clarified the effective regime, limitations, and failure boundaries of FRL, and referenced Appendix I for detailed failure mode analysis.
  Retained hyperlinks to the pseudocode and complexity analysis appendices.

---

### **6 Empirical Evaluation**

- **[A] [327–344] [cSKw Q3, Kovs Q1]**
  Added explanations to each dataset on how low-resolution data were obtained and why the specific downsampling factors and maximum factors were chosen.

- **[R] [355–357] [cSKw Q2]**
  Removed the boldface on “notably.”

- **[A] [503–512] [cSKw W4, SPE1 W1&Q1]**
  Added a standalone subsection at the end of the experimental results summarizing key computational cost measurements, with pointers to Appendix E for full complexity analysis.

---

### **Appendices**

- **[R] [756–772]**
  Updated the appendix table of contents.

- **[A] [1127–1241] [7EgH Q4]**
  Added a theoretical analysis appendix showing that FRL guarantees reduced high-resolution inference error (but not power-law convergence as in numerical solvers).

- **[R] [1326–1392] [cSKw W4, SPE1 W1&Q1]**
  Replaced the previous coarse cost estimates with a complete computational complexity table.

- **[R] [2040–2135] [SPE1 W2&Q2, 7EgH W1–3, cSKw Q1]**
  Expanded the FRL failure-mode analysis with concrete examples, clearly explaining FRL’s assumptions, boundaries of validity, failure modes, and possible improvements.

---

### **Abstract, 1 Introduction, and 7 Discussion**

- **[R]**
  Revised relevant descriptions to reflect all modifications above.

---

We manually replaced added and revised text with blue color to facilitate reviewer examination.

We thank all reviewers for their insightful comments and questions, which significantly improved the completeness and clarity of this work.

---

### Author Response · Authors · 2025-11-29
**Summarization - Part 1**

In light of the recent OpenReview incident, we regret that we can no longer directly discuss with the reviewers. We sincerely thank all reviewers and both the original and newly assigned Area Chairs for their time and thoughtful evaluation.

Below we briefly summarize: (1) the core contributions and limitations of our work, (2) how we addressed the main reviewer concerns in Rebuttal Version 1, and (3) the review scores and post-rebuttal attitudes, as background for your decision as AC.

---

### **(1) Core contributions and limitations**

**Paper:** *Breaking Scale Anchoring: Frequency Representation Learning for Accurate High-Resolution Inference from Low-Resolution Training*

### **Core contributions**

1. **Identify Scale Anchoring (SA) as a new fundamental limitation in ZS-SR STF.**
   We formalize SA as an information-theoretic effect: models trained only on low-resolution data cannot correctly handle frequencies above the training Nyquis during high-resolution inference, leading to nearly constant error across resolutions that is often misinterpreted as successful multi-resolution generalization. This clarifies *why* many ZS-SR STF models fail to reduce error as resolution increases like numerical solvers.

2. **Propose Frequency Representation Learning (FRL), an architecture-agnostic SA mitigation.**
   FRL comprises: (i) multi-resolution training via systematic downsampling, (ii) a **Nyquist-normalized frequency representation** that aligns the same physical frequency across different grids (the main novel component), and (iii) a frequency-aware loss coupling spatial and spectral consistency. Steps (i) and (iii) adapt existing multi-scale and spectral practices; step (ii) is new and directly motivated by SA. Moreover, ablation studies demonstrate that step (ii) is the core contribution of FRL for mitigating SA.

3. **Demonstrate FRL’s architecture-agnostic scale decoupling across fluid simulation and weather forecasting.**
   On 2D/3D fluid simulation and ERA5 weather forecasting, baseline models (GNNs, Transformers, CNNs, Diffusion models, NOs, Neural ODEs, standard NNs) exhibit RMSE ratios (high-res / low-res) ≈ 1 or >1 under up to 64× super-resolution (errors are anchored). FRL-enhanced variants consistently reduce high-resolution error and bring RMSE ratios well below 1, while stabilizing frequency response across the whole band. This shows that FRL makes ZS-SR STF models behave more like numerical solvers whose error decreases with resolution.

4. **Quantify overhead and practical cost.**
   We show that FRL incurs modest overhead: training time ≈ 1.1×–1.4×, peak VRAM ≈ 1.3×–1.5×, and <2% inference overhead. This suggests FRL is practically deployable in realistic training and inference setups.

5. **Clarify applicability and failure modes.**
   We explicitly state that FRL relies on scale-consistent spectral structure (e.g., moderate-Re flows, typical ERA5 regimes) and can fail in very high-Re turbulence or extreme events, where spectral relationships become too irregular. We provide a concrete high-Re example and outline possible extensions via physics-based spectral constraints. This delineates where FRL can be expected to work and where additional physics priors are required.

### **Why this is important:**

Taken together, these contributions identify a new fundamental failure mode of current ZS-SR STF models and provide a practical, architecture-agnostic representation scheme that allows error to decrease with resolution. If **deep learning models are to serve as low-cost alternatives to computationally expensive high-resolution numerical solvers for generating high-fidelity spatiotemporal fields**, SA is a problem that must be explicitly identified, evaluated, and mitigated, and FRL represents a concrete first step toward its mitigation.

### **Limitations**

- **Assumptions, failure boundaries, and concrete extensions.**
  FRL assumes that the underlying dynamics exhibit *relatively smooth* spectral structure across neighboring bands, so that the energy spectrum envelope and local relationships between low/mid and higher frequencies are smooth and statistically extrapolable across resolutions. This is typically satisfied for the moderate-Re flows and weather datasets we study, but can break down in extremely high-Re turbulence or extreme weather, where spectra become highly intermittent and non-smooth and FRL’s frequency extrapolation degrades so that SA can re-emerge. In the failure-mode Appendix I (high-Re turbulence example), we therefore propose a concrete extension: incorporating explicit physical spectral constraints—specifically, Kolmogorov-type inertial-range scaling (e.g., $E(k)\sim k^{-5/3}$)—into FRL’s normalized frequency representation and loss, so that in regimes where such universality holds, the added spectral prior can further stabilize FRL beyond the regime covered in the current experiments.

---

### Author Response · Authors · 2025-11-29
**Summarization - Part 2**

- **Fundamental information-theoretic limits on SA mitigation.**
  Because SA ultimately stems from Nyquist-limited information loss, no method can fully “undo” it including FRL: different high-frequency realizations can correspond to the same low-resolution observation. Any approach can only mitigate SA under additional structural assumptions on the underlying physics.

---

### **(2) Main reviewer concerns and how we addressed them**

### Reviewer cSKw

**Main concerns.**
(i) What the relationship is between SA and existing phenomena such as spectral bias, discretization invariance, and DME; (ii) to what extent FRL is genuinely new, given that Steps (i) and (iii) resemble known multi-resolution and spectral-regularization techniques; and (iii)  a more transparent error–cost trade-off.

**Our updates.**
- Consolidated all related issues into a single subsection in the related-work section, explicitly distinguishing SA from spectral bias, discretization invariance, and DME and analyzing differences in source, scope, and consequences.
- Clarified that Steps (i) and (iii) of FRL are variants of existing methods and Step (ii) is the novel component, supported by explicit references to the existing ablations in Appendix H.
- Added a short subsection at the end of the experimental section summarizing measured training/inference time and VRAM overhead and pointed to a complete complexity table in Appendix E.

### Reviewer SPE1

**Main concerns.**
(i) Insufficiently quantified analysis of computational overhead, especially for training time, inference time, and memory, which affects the practical applicability of FRL; and (ii) an unclear description of FRL’s effective regime and failure modes, particularly for very high-Re turbulence.

**Our updates.**
- Added a dedicated subsection in the empirical section summarizing measured overhead and referred to a full computational complexity table in Appendix E, making the cost–accuracy trade-off explicit.
- Clarified FRL’s applicable regime and limitations in the main text and substantially expanded the failure-mode analysis in Appendix I with a high-Re turbulence example, where we explain why FRL’s assumptions break down and how incorporating Kolmogorov-type spectral constraints could extend FRL to such regimes.

### Reviewer 7EgH

**Main concerns.**
(i) How ZS-SR inference can see error decrease with resolution despite training data being Nyquist-limited; (ii) whether our conclusions rely on “too smooth” data and what assumptions we make about the spectral smoothness of the underlying dynamics; and (iii) how our perspective relates to aliasing/discretization work such as DME.

**Our updates.**
- Added a theoretical analysis in Appendix D, formulated in normalized frequency space, explaining how FRL can reduce high-resolution error under scale-consistent spectra (while not achieving numerical-solver-style power-law convergence) and clarified this connection in the main text.
- Made explicit in the method and discussion sections that FRL assumes a certain degree of spectral smoothness/scale-consistency, clarified that this holds for our current fluid and weather datasets, and pointed to Appendix I where we discuss failure modes when this assumption breaks.
- Explicitly contrasted SA with aliasing/discretization phenomena and clarified how our multi-step ZS-SR STF setting relates to DME’s single-step analysis in the related-work section, showing that their low-frequency discretization effects are compatible with our observation that multi-step cross-resolution error is dominated by super-Nyquist components.

### Reviewer Kovs

**Main concerns.**
(i) Whether FRL is fundamentally new relative to ZSSR and other Deep Internal Learning–style methods; (ii) whether the experimental scope, limited to spatiotemporal forecasting, is sufficient without standard image SR benchmarks; and (iii) how the proposed frequency consistency loss differs from existing frequency-domain losses.

**Our updates.**
- Refined the definitions of STF and ZS-SR STF to make the task setting self-contained and clearly distinct from image ZSSR.
- Clarified in the method section that FRL’s Step (ii), the Nyquist-normalized frequency representation, is the truly novel part, while Step (i) and Step (iii) follow existing ideas, and we explicitly pointed to Appendix H showing that frequency losses alone do not break SA without the normalized representation.

---

### **(3) Review scores and post-rebuttal stance**

- **Initial scores:** cSKw - 4, SPE1 - 6, 7EgH - 6, Kovs - 6
- **After rebuttal (before the leak):** cSKw - 6, SPE1 - 6, 7EgH - 6, Kovs - 6

cSKw: "The majority of my concerns have been addressed."

7EgH: "All of my concerns and questions have been addressed."

At present, all reviewers indicate a leaning toward acceptance.

---

We hope this concise summary is helpful under the new process, and we are very grateful for your time and consideration.

---

### Meta-Review · Area_Chair_8TQy · 2026-01-06

**Summary:**

The authors of this submission identified and defined the "Scale Anchoring" (SA) problem in Zero-Shot Super-Resolution Spatiotemporal Forecasting (ZS-SR STF) due to the fundamental Nyquist limit. A architecture-agnostic Frequency Representation Learning (FRL) solution by introducing frequency-aware training with resolution-invariant frequency embedding was proposed to "break" or alleviate (probably more appropriate) SA. Empirical results on fluid simulation and weather forecasting with a variety of different model architectures demonstrated the effectiveness of the proposed FRL solution.

**Reviewer Concerns:**

During the rebuttal discussion, the authors have also provided significant additional results with detailed spectral error analyses and runtime and computational complexity comparison.

There were concerns from the reviewers regarding positioning of the presented SA and the corresponding FRL solution. The authors have provided more detailed literature review, discussions, and the potential limitations of the presented solution in more complex (turbulent for example) dynamics.

The authors may consider improving their solution method description of the corresponding components in more detail either in the main text or appendix in the final version. For example, resolution-invariant embedding was presented as one important step with normalized frequency representations, which can be presented more clearly with consistent math notations.

There have been frequency-aware training strategies in literature, including https://arxiv.org/abs/2012.12821 and https://arxiv.org/abs/2405.12202 in addition to the ones provided by the reviewers. The authors may want to include these into benchmarking and/or discussions.

**Reviewer Scores:**

With the rebuttal discussions, reviewers cSKw and 7EgH indicated that they would provide positive scores. With these, all the reviewers may reach consensus with the marginally above acceptance recommendation.

---

### Decision · Program_Chairs · 2026-01-26

Accept (Poster)